# High drug-loaded microspheres enabled by controlled in-droplet precipitation promote functional recovery after spinal cord injury

Wei Li [1,7], Jian Chen [2,7], Shujie Zhao[2], Tianhe Huang [3], Huiyan Ying [1], Claudia Trujillo[1], Giuseppina Molinaro[1], Zheng Zhou[2], Tao Jiang[2], Wei Liu[2], Linwei Li[2], Yuancheng Bai[3], Peng Quan[1,4], Yaping Ding [1], Jouni Hirvonen[1], Guoyong Yin [2✉], Hélder A. Santos [1,5,6✉], Jin Fan [2✉] & Dongfei Liu [3,1,5✉]

Drug delivery systems with high content of drug can minimize excipients administration, reduce side effects, improve therapeutic efficacy and/or promote patient compliance. However, engineering such systems is extremely challenging, as their loading capacity is inherently limited by the compatibility between drug molecules and carrier materials. To mitigate the drug-carrier compatibility limitation towards therapeutics encapsulation, we developed a sequential solidification strategy. In this strategy, the precisely controlled diffusion of solvents from droplets ensures the fast in-droplet precipitation of drug molecules prior to the solidification of polymer materials. After polymer solidification, a mass of drug nanoparticles is embedded in the polymer matrix, forming a nano-in-micro structured microsphere. All the obtained microspheres exhibit long-term storage stability, controlled release of drug molecules, and most importantly, high mass fraction of therapeutics (21.8–63.1 wt%). Benefiting from their high drug loading degree, the nano-in-micro structured acetalated dextran microspheres deliver a high dose of methylprednisolone (400 μg) within the limited administration volume (10 μL) by one single intrathecal injection. The amount of acetalated dextran used was 1/433 of that of low drug-loaded microspheres. Moreover, the controlled release of methylprednisolone from high drug-loaded microspheres contributes to improved therapeutic efficacy and reduced side effects than low drug-loaded microspheres and free drug in spinal cord injury therapy.

[1] Drug Research Program, Division of Pharmaceutical Chemistry and Technology, Faculty of Pharmacy, University of Helsinki, Helsinki 00014, Finland. [2] Department of Orthopaedics, The First Affiliated Hospital of Nanjing Medical University, Nanjing 210029, China. [3] State Key Laboratory of Natural Medicines, Department of Pharmaceutical Science, China Pharmaceutical University, Nanjing 210009, China. [4] Department of Pharmaceutical Science, School of Pharmacy, Shenyang Pharmaceutical University, Shenyang 110016, China. [5] Helsinki Institute of Life Science (HiLIFE), University of Helsinki, Helsinki 00014, Finland. [6] Department of Biomedical Engineering and W.J. Kolff Institute for Biomedical Engineering and Materials Science, University Medical Center Groningen/University of Groningen, Ant. Deusinglaan 1, 9713 AV Groningen, The Netherlands. [7] These authors contributed equally: Wei Li, Jian Chen. ✉email: guoyong_yin@sina.com; h.a.santos@umcg.nl; fanjin@njmu.edu.cn; dongfei.liu@cpu.edu.cn

Encapsulation is a powerful, well-known approach for incorporating active ingredients in another inert material[1]. This inert material isolates active ingredients from external environment by forming a physical barrier[2]. The formed barrier not only protects active ingredients from harsh conditions, but also enables the controlled release of incorporated active ingredients, for example, drug molecules[1]. The controlled drug release is capable to tailor the pharmacokinetic features of encapsulated drug molecules. Biodegradable polymer microspheres, one of the most representative controlled drug delivery systems, offer a way to control the drug release over weeks or months. These microspheres extend the apparent half-life of encapsulated drugs, thus eliminating the need for multiple doses.

By increasing the mass fraction of therapeutics in microspheres, we can increase the dose in one single administration, reduce the frequency of administration, improve the patient compliance, and cut the use of materials/excipients and associated side effects[3]. For some diseases, a high mass fraction of therapeutics in microspheres is a prerequisite for an effective therapy[4,5]. For example, dosing by the intrathecal route is restricted by the small injection volume (about 10 μL per rat, 0.5–5 mL for human beings), resulting in a limited amount of injectable polymer microspheres. Therefore, an approach that can formulate microspheres with high mass fraction of therapeutics is highly desirable.

The engineering of microspheres with high mass fraction of therapeutics is a challenge as the loading capacity is inherently limited by the compatibility, which refers to miscibility, between active ingredients and inert polymer materials[6–8]. Microspheres are composed primarily of nontherapeutic hosting polymers, for example, poly(lactic-co-glycolic acid) (PLGA). To improve the drug loading capacity, a variety of approaches have been developed to increase the drug-carrier material compatibility, such as utilizing electrostatic attraction and donor–receptor coordination between drug molecules and carrier materials[9–11]. These approaches, however, demand specific molecule features simultaneously for both drug and carrier molecules. When drug nanoparticles, instead of drug molecules, are encapsulated by polymer matrix, the demanded molecule features between drug and carrier molecules can be greatly weakened or even avoided. Regarding the nano-in-micro structure, a mass of drug nanoparticles can be embedded in the polymeric microspheres, leading to the high mass fraction of therapeutics. We hypothesize that oversaturation and precipitation of drug molecules prior to the solidification of polymers in droplets can mitigate the compatibility limitation between drug molecules and carrier materials, and enable the production of nano-in-micro structured microspheres with high content of drug nanoparticles (Fig. 1a, b).

In this work, we control the solvent removal process from droplets by a droplet-based microfluidic device and achieve precisely controlled sequential solidification of drug and polymer molecules by introducing a cosolvent for drug molecules to the droplets. In these droplets, the faster diffusion of cosolvent, which is miscible with the outer fluid, ensures the precipitation of drug molecules and the formation of drug nanoparticles inside the droplets. The solidification of polymer molecules ultimately enables the microencapsulation of a large number of drug nanoparticles (Fig. 1b). To further demonstrate the therapeutic efficacy of engineered microspheres, acetalated dextran (AcDX) microspheres with high mass fraction (63.1 wt%) of methylprednisolone (MPS) are evaluated for their efficacy in spinal cord injury (SCI) therapy.

## Results

### Proof of in-droplet precipitation strategy to engineer microspheres with high mass fraction of cargo. Microspheres were prepared using a microfluidic flow-focusing device. A mixture of

primary solvent and cosolvent, i.e., ethyl acetate and dimethyl sulfoxide, respectively, containing polymer and drug molecules comprised the inner fluid. The outer fluid was Poloxamer 407 solution (1% w/v). Tetraphenylethene (TPE), an aggregation-induced emission molecule (Fig. 1c), was used to verify the hypothesis of in-droplet precipitation of drug molecules. TPE is much more emissive in aggregated state than that in dissolved state[12]. This feature makes TPE an ideal tool for real-time observation of the in-droplet precipitation behavior of payloads. The biocompatible and biodegradable PLGA and AcDX served as carrier materials as they are widely used for drug delivery.

Droplets containing AcDX and TPE became emissive under ultraviolet illumination when they were moving forward in the capillary tube, indicating the aggregation of TPE molecules (Fig. 1d, Supplementary Movie 1). When only TPE was in the droplets, we observed a faster emission of blue fluorescence than those droplets containing AcDX and TPE. The emissive difference of TPE between these two types of droplets indicated that the presence of polymer (AcDX) may hinder the in-droplet precipitation of TPE molecules. By contrast, ethyl acetate droplets with AcDX and TPE did not emit blue fluorescence, suggesting that the fast diffusion of dimethyl sulfoxide from the droplets to the outer fluid is essential for triggering the in-droplet precipitation of TPE.

As shown in Fig. 1e, there are numerous nanoparticles on the cross-sections of TPE encapsulated AcDX microspheres (TPE@AcDX). These nanoparticles are expected to be the in-droplet precipitated TPE nanoparticles. X-ray powder diffraction and differential scanning calorimetry results (Fig. 1f, g) further confirmed the presence of crystallized TPE inside the TPE@AcDX microspheres, which can be ascribed to the in-droplet precipitation of TPE. The drug loading degree (mass fraction of drug in the microspheres) and encapsulation efficiency (percentage of drug loaded relative to the total amount of drug) of TPE were $47.1 \pm 0.5$ wt% and $68.4 \pm 1.3\%$, respectively. Such loading degree value (>30 wt%) is much higher than that of most reported polymer microspheres[4,13].

### Versatile engineering of microspheres with high mass fraction of therapeutics. To evaluate the versatility of this strategy for preparing microspheres with high drug loading, therapeutics with different molecular structures and physicochemical properties, including atorvastatin (ATV, 558.6 g/mol, log $P$ 5.7), methylprednisolone (MPS, 374.5 g/mol, log $P$ 1.5) and hydrochlorothiazide (HCT, 297.7 g/mol, log $P$ −0.07) (Supplementary Fig. S1a), served as model drugs. Drug-loaded polymer microspheres were prepared by droplet microfluidics, using ethyl acetate as the solvent for the inner fluid. The obtained drug loading degree (Supplementary Fig. S1b) and encapsulation efficiency (Supplementary Fig. S1c) of these microspheres were in the range of 0.1–0.6 wt% and 1.3–6.8%, respectively. The drug loading capacity is intrinsically determined by the compatibility between drug molecules and carrier materials[6–8,14,15], although the actual drug loading degree in a carrier material might be influenced by a specific preparation method. Such poor drug encapsulation (Supplementary Fig. S1c) can be ascribed to the poor compatibility between the drug (ATV, MPS, or HCT) and polymer (AcDX or PLGA) molecules.

To prepare nano-in-micro structured microspheres with high drug loading, we added a cosolvent dimethyl sulfoxide to increase the solubility of the drug molecules in the primary solvent, i.e., ethyl acetate (Supplementary Fig. S2). An additional benefit of adding cosolvent dimethyl sulfoxide is its rapid diffusion from the droplets to the surrounding outer fluid. This fast diffusion causes oversaturation and in-droplet precipitation of drug molecules

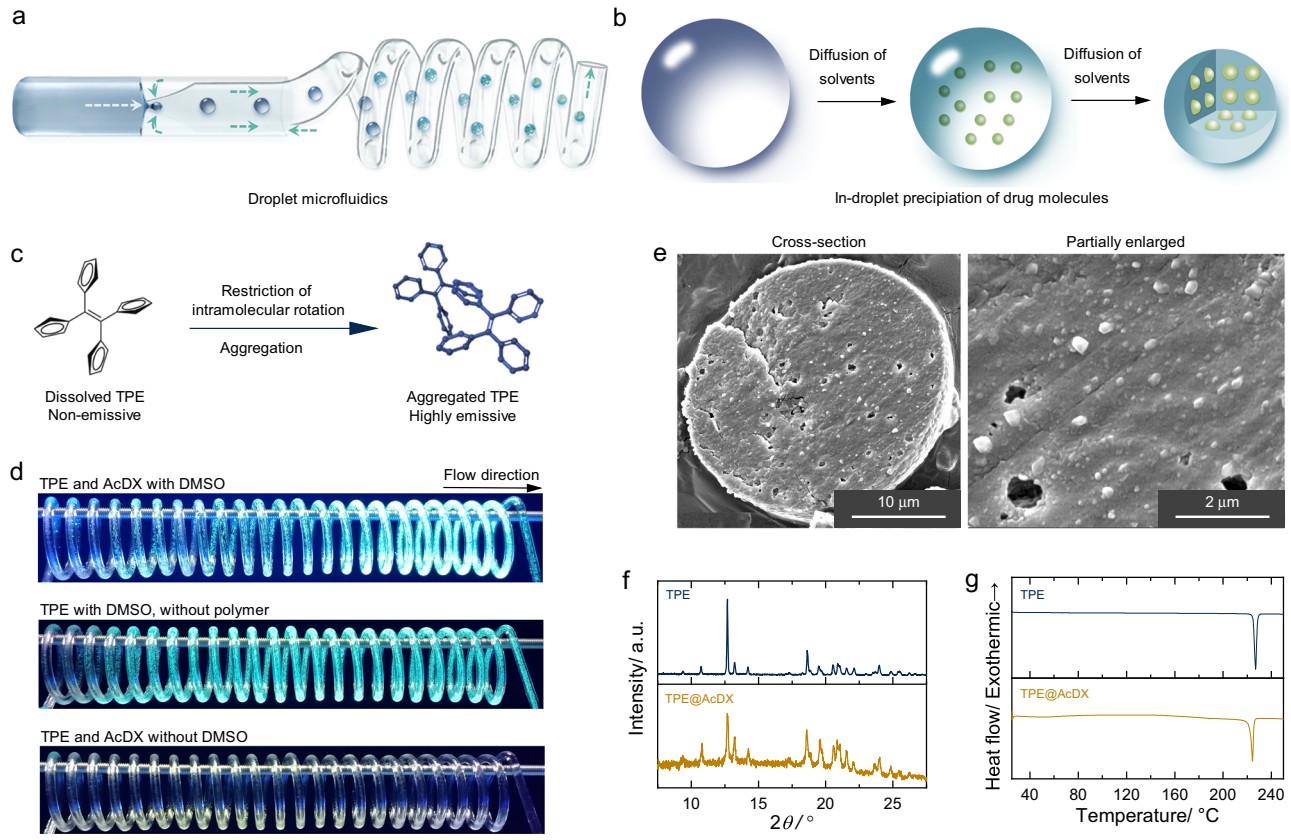

**Fig. 1 Preparation of nano-in-micro structured microspheres. a** The preparation of high drug-loaded microspheres by droplet microfluidics. **b** The structural evolution of the droplets during solidification. **c** Propeller-shaped TPE is non-emissive when dissolved but becomes highly emissive when its molecules are aggregated, due to the restriction of intramolecular rotation of its phenyl rotors against its ethylene stator in the aggregate state. **d** Droplets generated by flow-focusing microfluidic device pass through the spiral-shaped capillary tube under ultraviolet irradiation. **e** Scanning electron microscope images of whole and enlarged cross-section of TPE@AcDX microspheres. X-ray powder diffraction (**f**) and differential scanning calorimetry (**g**) of TPE@AcDX microspheres and TPE.

prior to the solidification of polymer materials. Ultimately, the resulting drug precipitates are proposed to be encapsulated by the polymer matrix upon further solidification of the droplets.

Since ATV, MPS and HCT are not capable of fluorescing, their precipitation process was observed under the phase contrast light microscope. To facilitate the visualization of drug precipitation, large droplets were produced by the microfluidic device with a co-flow pattern. All three drug molecules precipitated inside the droplets, although the occurrence time of visible drug precipitates varied in a manner of MPS < HCT < ATV. Moreover, the shape and size of the precipitates were different for each drug (Fig. 2a). If the precipitated drugs are kept inside the droplets, it is logical that the obtained microspheres would show high drug loading degree.

After confirming the versatility of in-droplet precipitation, drug-loaded polymer microspheres were prepared by droplet microfluidics. All these microspheres showed uniform particle size (Supplementary Fig. S3). There were drug nanoparticles on the cross-sections of microspheres (Fig. 2b, c). Drug loading degree of these microspheres was in the range of 21.8–63.1 wt% (Fig. 2d) and was about 40–450 times higher than those prepared with ethyl acetate only as the inner fluid (Supplementary Fig. S1b). It is interesting that the in-droplet precipitation also significantly increased the encapsulation efficiency of the obtained microspheres (Supplementary Figs. S1c, 2e). This is out of ordinary for microspheres prepared by emulsion method, as the encapsulation efficiency usually decreases when drug loading degree increases[16].

We studied the solid state of all three drugs before and after encapsulation by X-ray powder diffraction and differential scanning calorimetry. As expected, the physical mixtures (bare microspheres and raw drug powders) clearly showed the crystal features of drug molecules in both X-ray powder diffractograms and differential scanning calorimetry curves (Supplementary Fig. S4). MPS-loaded AcDX (MPS@AcDX) and PLGA (MPS@PLGA) microspheres exhibited diffraction peaks of the encapsulated MPS (Fig. 2f), indicating the formation of drug nanocrystals in the microspheres. Regardless of the polymer matrix, the encapsulated ATV and HCT nanoparticles were primarily amorphous, as only broad peaks expressed in the X-ray powder diffractograms. The amorphous state of ATV and HCT in microspheres was also confirmed by the differential scanning calorimetry curves, showing exothermic peaks followed by endothermic peaks (Fig. 2g). This solid-state difference among ATV, MPS, and HCT in microspheres could be ascribed to their in-droplet precipitation behavior. Specifically, MPS precipitated faster than that of ATV and HCT in-droplet under current experimental conditions (Fig. 2a). Faster precipitation of MPS enables the relatively longer period of time for both nucleation and crystal growth of MPS during the droplet solidification process, and ultimately the presence of MPS crystals.

**Numerical simulation of the droplet solidification and drug encapsulation.** The solidification of emulsion droplets is usually driven by solvent diffusion, which is a typical mass transfer-

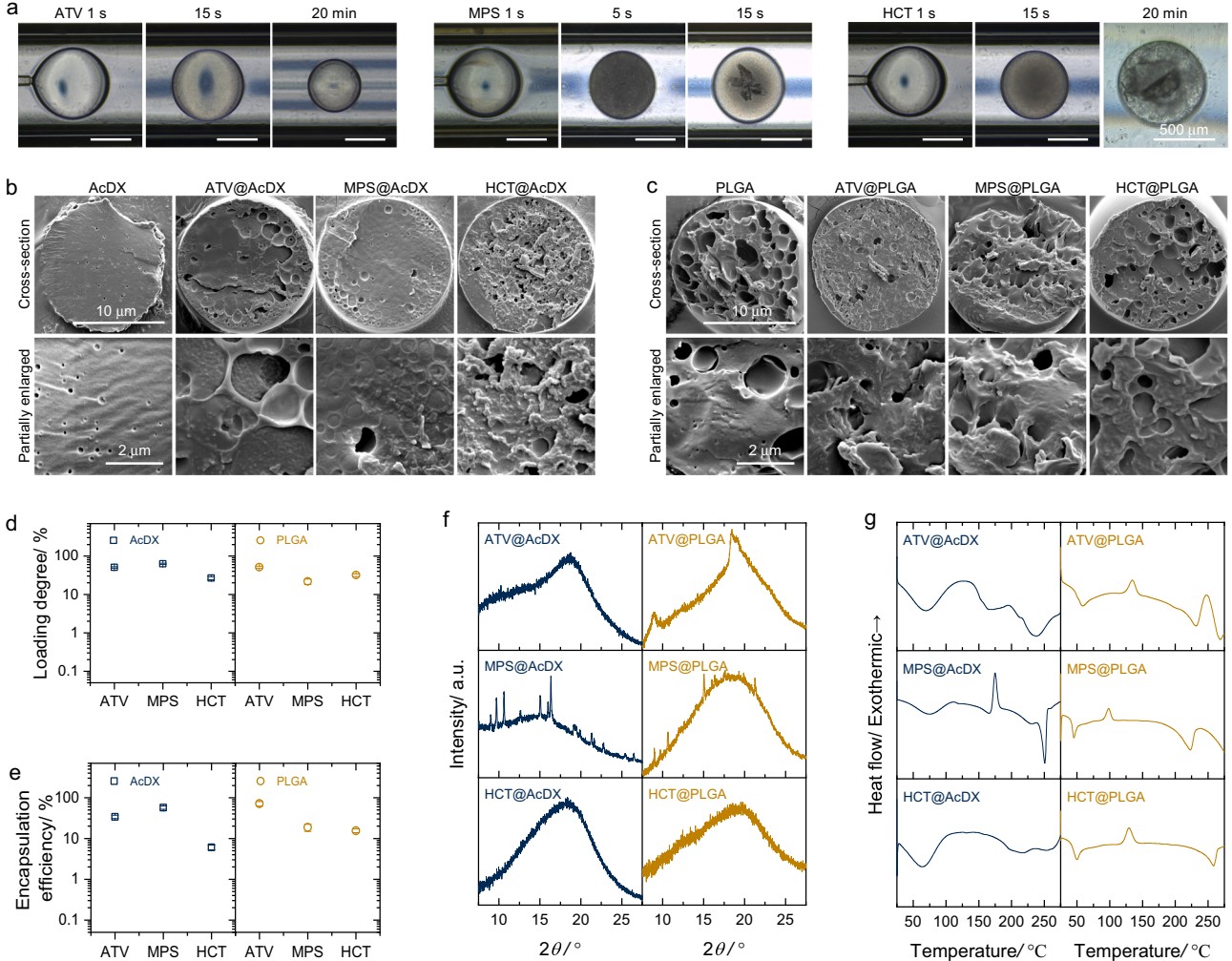

**Fig. 2 The versatility of controlled in-droplet precipitation. a** Light microscope images of drug precipitation process in the droplets generated by co-flow microfluidics. **b, c** Scanning electron microscope images of microspheres prepared by droplet microfluidics, using AcDX (**b**) and PLGA (**c**) as carrier matrix. **d, e** Drug loading degree (**d**) and encapsulation efficiency (**e**) of microspheres prepared with and without dimethyl sulfoxide in the inner fluid ($n = 3$). Data are presented as mean values ± SD. **f, g** X-ray powder diffractogram (**f**) and differential scanning calorimetry curves (**g**) show the solid-state of the encapsulated drug molecules in microspheres.

controlled process and follows the Fick's law. We performed numerical simulations to study the mass diffusion involved in the droplet solidification within the microfluidic channel. In previously developed models[17,18], the droplets contain only one single solvent. Here, we used a mathematical model to study the droplet solidification by taking into account of free drug molecules, cosolvent and drug precipitation in the droplets. The concentration distribution of each component inside the droplet is calculated by Fick's law (Eqs. 1–5). With the calculated concentrations, the mass of each component is obtained by volumetric integration of the droplet, and the volume of each component, volume of droplet and size of the droplet are calculated by Eqs. 6–8, respectively. The variation of droplet size is obtained by iterating volume change via mass conservation. The mass of droplet decreases when solvents diffuse out from the droplet, and consequently the volume and size of the droplet decreases. Simulation was conducted according to the experimental particle size and loading degree of prepared microspheres (Table S1). The droplet size sharply reduced upon solvent diffusion (Fig. 3a, b), meanwhile, the MPS concentration in droplets rapidly increased and reached the saturated state in 38.3 ms (Fig. 3c). The simultaneously decreased MPS solubility and

increased MPS concentration in droplets suggested that the remaining MPS molecules in droplets would be largely precipitated.

The in-droplet precipitation of MPS was further experimentally confirmed by checking the thin sections of the nano-in-micro structured MPS@AcDX microspheres. Compared to bare AcDX microspheres, the matrix of MPS@AcDX microspheres was filled with solid particles that can be washed away by water (Fig. 3d). Therefore, these embedded particles should be in-droplet precipitated MPS rather than water-insoluble AcDX. The quickly solidified microspheres appear to kinetically trap in-situ formed drug particles in polymer matrix, resulting in relatively uniform drug distribution in microspheres[19].

We verified the impact of initial dimethyl sulfoxide ratio in the inner fluid toward the drug encapsulation in MPS@AcDX. When the initial dimethyl sulfoxide ratio increased from 10% v/v to 25% v/v, the time for MPS to reach saturation concentration increased from 38.3 to 168.4 ms (Fig. 3e). With such an increase of dimethyl sulfoxide ratio, MPS loading degree decreased from 62.9 wt% to 60.7 wt% (Fig. 3f). Meanwhile, MPS encapsulation efficiency dropped from 56.6% to 51.4%. These numerical simulation results are in accordance with those experimental ones: the higher

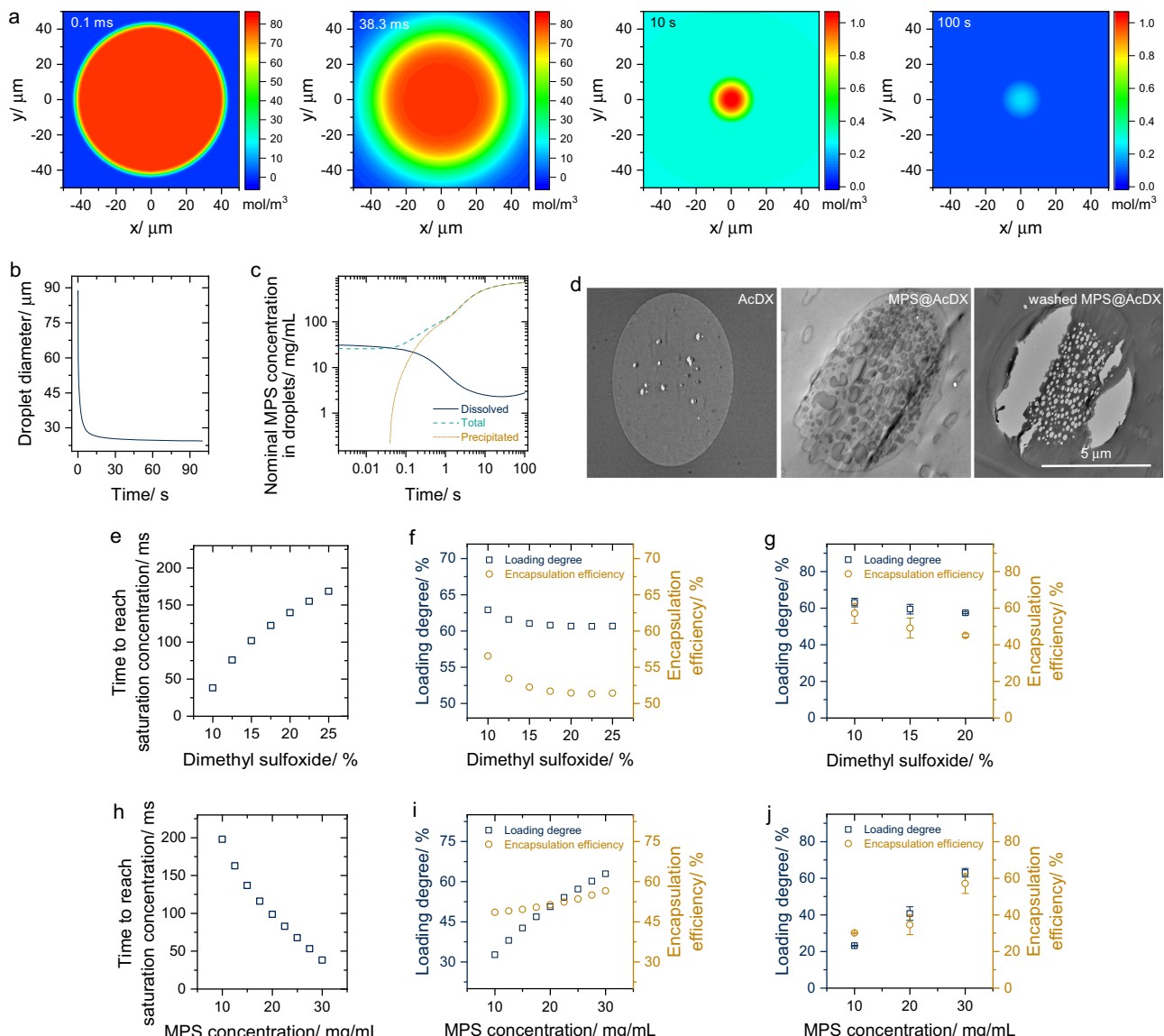

**Fig. 3 Numerical simulation of the droplet solidification and drug encapsulation. a** Concentration fields of MPS on the mid-plane of the droplet. The color bar on the right side represents the concentration magnitude of MPS. **b** The shrinking of a droplet as a function of time during the solidification process. **c** Nominal concentration (mass of drug divided by volume of droplet) of MPS at different states (dissolved and precipitated) in the droplet during the solidification process. **d** Transmission electron microscope images of thin section of bare AcDX, MPS@AcDX, and washed thin section of MPS@AcDX. **e** The impact of initial dimethyl sulfoxide ratio in the inner fluid on the simulated diffusion time that taken for MPS to reach the saturation concentration in droplets. **f, g** Numerically simulated (**f**) and experimental (**g**; $n = 3$) loading degree and encapsulation efficiency of AcDX microspheres prepared by varying the dimethyl sulfoxide ratio in the inner fluid. AcDX concentration and MPS concentration are set at 10 mg/mL and 30 mg/mL, respectively. Data are presented as mean values ± SD. **h** The impact of initial MPS concentration in the inner fluid on the numerically simulated diffusion time that taken for MPS to reach saturation concentration in the droplets. **i, j** Numerically simulated (**i**) and experimental (**j**; $n = 3$) loading degree and encapsulation efficiency of AcDX microspheres. AcDX concentration and dimethyl sulfoxide ratio are set at 10 mg/mL and 10% v/v, respectively. Data are presented as mean values ± SD.

the initial dimethyl sulfoxide ratio of the inner fluid, the lower drug loading degree and encapsulation efficiency for engineered microspheres (Fig. 3g).

Moreover, we evaluated the effect of initial MPS concentration on the drug encapsulation in microspheres. When the initial MPS concentration increased from 10 mg mL$^{-1}$ to 30 mg mL$^{-1}$ in the inner fluid, the diffusion time it took for MPS to reach saturation concentration decreased from 197.8 to 38.3 ms (Fig. 3h). Interestingly, MPS loading degree in MPS@AcDX increased from 32.7 wt% to 62.9 wt%, and MPS encapsulation efficiency climbed from 48.5% to 56.6% (Fig. 3i). We verified these simulated results

by engineering MPS@AcDX microspheres with the initial MPS concentration at 10, 20 and 30 mg/mL, respectively. Regarding these experimental results, the MPS loading degree increased from 23.3 wt% to 63.1 wt%; in the meantime, its encapsulation efficiency boosted from 30.2% to 57.2% (Fig. 3j). Overall, the experimental results of the impact of initial MPS concentration on drug encapsulation are in good agreement with those simulated ones.

From a kinetic point of view, both diffusion time and diffusion driving force determine the number of drug molecules diffused out droplets. In terms of diffusion time, the total time for the

droplet solidification process is at the same level for all the formulations. Therefore, the diffusion driving force is in a dominant position for drug encapsulation in microspheres. Diffusion driving force positively correlates with the drug concentration in droplets. In simulation, let us suppose that the MPS solubility in ethyl acetate is infinite. Under this condition, MPS would not precipitate during droplet solidification. Although high initial MPS concentration increased the driving force for MPS to diffuse out droplets, the fixed droplet solidification time would increase the absolute number of drug molecules stay inside droplets and decrease the in-droplet proportions of corresponding drug molecules (Supplementary Fig. S5). Therefore, higher drug loading degree typically results in lower encapsulation efficiency. This trend actually is widely reported for microspheres prepared by conventional emulsion method[16].

Regarding our strategy, the fast diffusion of dimethyl sulfoxide decreased the saturated concentration of drug molecules and, consequently, generated the in-droplet oversaturation and precipitation of drug molecules. After precipitation, the in-droplets concentration of drug molecules considerably dropped and maintained at the saturated level (Fig. 3c). This saturated drug concentration reached a constant low level after most dimethyl sulfoxide molecules diffused to the outer fluid. As a result of the reduced drug concentration, the driving force for drug molecules to diffuse out of droplets significantly weakened. Moreover, the nearly constant drug concentration inside the droplets would result in a fixed number of drug molecules diffused to the outer fluid. The high initial MPS concentration and low dimethyl sulfoxide ratio in the inner fluid shortened the time for MPS to reach saturation concentration. The earlier the saturation concentration been reached in the inner fluid, the less the number of drug molecules involved in diffusion and, consequently, the less the number of drug molecules diffused out of droplets. Therefore, the higher amount of drug we added, the lower the ratio of drug molecules diffused out, and ultimately, the higher the encapsulation efficiency it would be. This is the reason why in-droplet precipitation strategy can not only enhance the drug loading degree but also significantly increase the encapsulation efficiency of the obtained microspheres (Supplementary Figs. S1c, 2e). Overall, the fast reduction of drug saturation solubility in droplets is contributing to highly efficient drug encapsulation in microspheres.

We also evaluated the drug release profiles for all the engineered microspheres. Complete release was achieved within a few hours for bare drugs (Supplementary Fig. S6). By contrast, these microspheres enabled the sustained release of payloads for 7–12 days depending on the specific combination of drugs and polymers (Fig. 4a, b). Injectable microspheres usually suffer from the burst release of payloads during the first day, which typically accounts for ~10–80% of the total drug loading. This so-called initial burst phenomenon poses a serious toxicity threat and is a major hurdle for the development of microsphere products[20]. For nano-in-micro structured microspheres, drug nanoparticles are fully segregated and isolated from the external environments. Therefore, the release of payloads only needs to cross the polymer matrix and almost no burst release has been observed for all the obtained microspheres. The controlled drug release in such a period potentializes these microspheres in clinical applications.

Amorphous materials are generally unstable, cohesive, potentially hygroscopic, and prone to recrystallization[21]. It is worthy to point out that although with high drug loading, all these microspheres are quite stable over the long-term storage (Fig. 4c–h), as confirmed by X-ray powder diffractograms. This long-term stability is likely to benefit from the nano-in-microstructure of the formed microspheres. The drug

nanoparticles are segregated from each other by the polymeric matrix. The polymer matrix can hinder the diffusion of drug molecules among different drug nanoparticles, which consequently inhibits the crystallization of the encapsulated amorphous drug nanoparticles.

**High MPS-loaded microspheres improve motor function and reduce spinal cord edema.** MPS is potentially beneficial in acute SCI therapy[22], however, high dose intrathecal administration of MPS ($1 \, mg \, kg^{-1}$ bolus dose plus $1 \, mg \, kg^{-1} \, h^{-1}$) is required[23]. In contrast to a high dose, the injection volume was limited (about $10 \, \mu L$ per rat)[24,25]. Therefore, systems with high mass fraction of MPS are highly desirable for SCI therapy. AcDX microspheres can protect neurons from glutamate-induced excitotoxicity[24], and hence we selected AcDX to formulate microspheres with high mass fraction of MPS, HMPS@AcDX, for SCI therapy.

To confirm the benefits of high drug loading in microspheres, vehicle phosphate-buffered saline (PBS, SCI group), low drug-loaded microspheres (LMPS@AcDX) and free MPS served as controls; the corresponding amount of bare AcDX microspheres for LMPS@AcDX, HAcDX ($4000 \, \mu g/10 \, \mu L$) were included. LAcDX ($630 \, \mu g/10 \, \mu L$) equals to the corresponding amount of bare AcDX microspheres for HMPS@AcDX. MPS loading degree for LMPS@AcxDX was only 0.4 wt%, thus, a single injection of LMPS@AcDX ($4000 \, \mu g$) only contained $16 \, \mu g$ MPS, which cannot satisfy the therapeutic needs. In contrast to LMPS@AcDX, the injection of about $630 \, \mu g$ HMPS@AcDX could efficiently deliver the required amount of MPS (about $400 \, \mu g$). Furthermore, when the MPS dose was fixed at $400 \, \mu g$, the amount of AcDX used in HMPS@AcDX was 1/433 of that of LMPS@AcDX.

We performed therapy analysis in rats undergoing a weight-drop injury of the thoracic spinal cord (T10; Fig. 5a). At first, the controlled delivery capability of MPS by AcDX microspheres was verified in vivo. The MPS content in cerebrospinal fluid was analyzed by liquid chromatography and triple quadrupole mass spectrometry. A representative chromatogram of MPS in the cerebrospinal fluid sample has been presented in Supplementary Fig. S7. At day-3 after injury, HMPS@AcDX reached the peak concentration (around $57 \, \mu g/mL$), then gradually decreased to about $3.8 \, \mu g/mL$ at day-7 post trauma (Fig. 5b). In terms of LMPS@AcDX, MPS concentration maintained between 0.3 and $0.6 \, \mu g/mL$ within 7 days after injury. For the MPS group, drug concentration rapidly declined to about $1.9 \, \mu g/mL$ within 1 day, which may ascribe to the high permeability of blood-spinal cord barrier after major trauma[26].

We also calculated the area under the MPS concentration-time curve (Fig. 5c) and the mean residence time of MPS (Fig. 5d) in cerebrospinal fluid. The area under the MPS concentration-time curve for HMPS@AcDX was significantly higher than that of both MPS ($P < 0.001$) and LMPS@AcDX ($P < 0.001$) groups. The mean residence time of MPS for HMPS@AcDX was significantly higher ($P < 0.01$) than that of MPS group but significantly lower ($P < 0.01$) than that of LMPS@AcDX. The highest mean residence time for LMPS@AcDX can be attributed to its slow release of MPS (Supplementary Fig. S8).

After traumatic SCI, motor behavior was assessed in open-field by the 21-point Basso, Beattie, and Bresnahan (BBB) locomotor rating scale (Fig. 5e). All groups exhibited complete hindlimb paralysis (BBB score = 0) at day-1 after major trauma. In comparison with SCI group, the administration of MPS, HAcDX, and LMPS@AcDX had no significant effects on the functional motor recovery, while LAcDX and HMPS@AcDX significantly improved the functional motor recovery from day-7 and day-3 post trauma, respectively. For HMPS@AcDX, the recovery of motor function was observed for day-28 after injury, which was

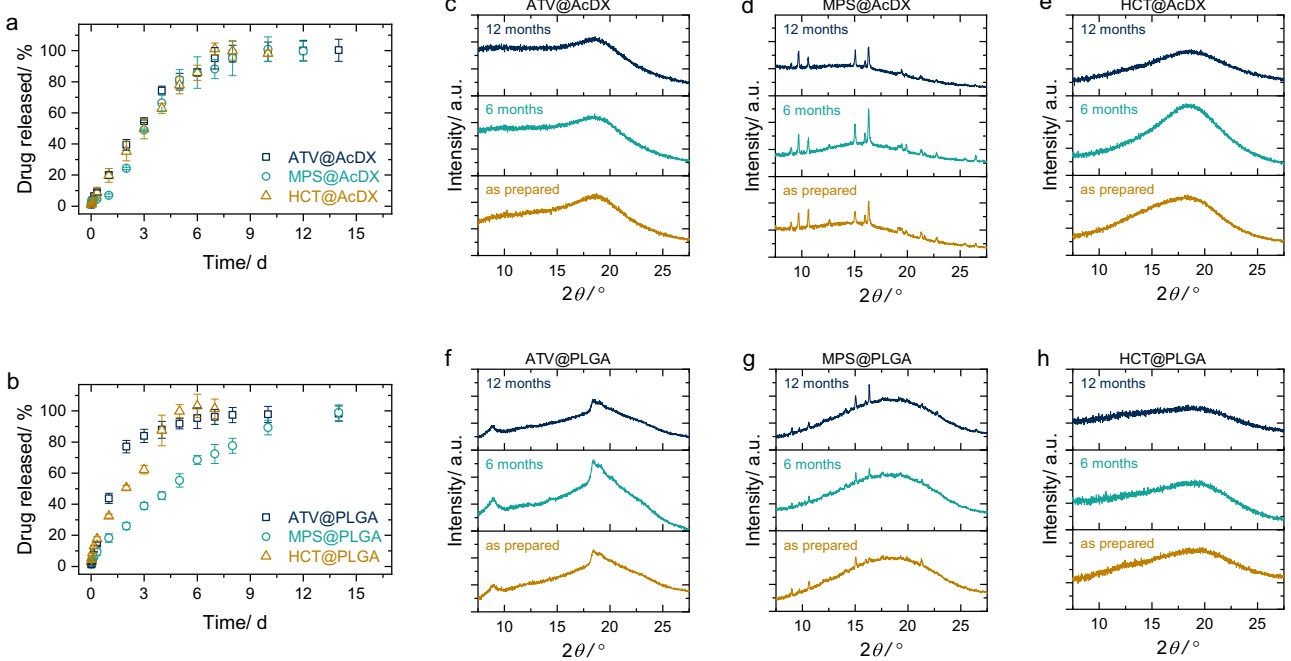

**Fig. 4 Release profiles and stability of the prepared microspheres. a, b** Release profiles of ATV, MPS, or HCT from AcDX (**a**) or PLGA (**b**) microspheres (n = 3). Data are presented as mean values ± SD. **c–e** X-ray powder diffractogram of drug-loaded AcDX microspheres, ATV@AcDX (**c**), MPS@AcDX (**d**) and HCT@AcDX (**e**), stored at 25 °C/40% relative humidity for 6 and 12 months. **f–h** X-ray powder diffractogram of drug-loaded PLGA microspheres, ATV@PLGA (**f**), MPS@PLGA (**g**), and HCT@PLGA (**h**), stored at 25 °C/40% relative humidity for 6 and 12 months.

much faster than that of LMPS@AcDX (from day-7) and LAcDX (from day-14). At day-28 post injury, HMPS@AcDX group scored 13.3, showing frequent to consistent weight supported plantar steps and occasional forelimb-hindlimb coordination. In comparison to the SCI group, LAcDX exhibited a faster rising trend in the BBB scores, but an inverse trend for HAcDX. This phenomenon was in consist with our previous study[24], in which AcDX microspheres with a concentration of 1 mg/mL protected neurons from glutamate-induced excitotoxicity.

The development of a fluid-filled cystic cavity is one of the prominent pathological features for the injured spinal cord. Therefore, we examined the pathological change of the injured spinal cord by Nissl staining (Fig. 5f). Four weeks after injury, the lesion area in the groups treated by LAcDX (P < 0.001) and HMPS@AcDX (P < 0.001) was notably smaller than that of the SCI group (Fig. 5g). In comparison with LMPS@AcDX (P < 0.001) and LAcDX (P < 0.05) groups, HMPS@AcDX also significantly reduced the loss of post-traumatic spinal cord tissue.

We further examined the pathological change of spinal cord at day-1 (Supplementary Fig. S9) and day-28 (Fig. 5h) post injury by a 7.0T magnetic resonance imaging unit. Hemorrhage and edema are recognized by hyperintense signal in T2 weighted images. The degrees of edema and damaged volume are illustrated in Fig. 5i. No significant pathological change was observed in all the groups at day-1 post trauma. When extending to 28 d after injury, the intensity ratio for the signal of spinal cord lesion to far-injury area and lesion volume in LAcDX and HMPS@AcDX groups were significantly lower than those in the SCI group (P < 0.001). Meanwhile, compared to SCI group, there was no statistically significant difference among the MPS, HAcDX, and LMPS@AcDX groups, regarding both the edema signal and damaged volume. While HMPS@AcDX group had significantly lower edema signal and damaged volume than LAcDX group (P < 0.001).

At day-1 after injury, the extent of cell apoptosis in the peritraumatic zone of spinal cord was evaluated by the terminal deoxynucleotidyl transferase-mediated dUTP nick end labeling (TUNEL) assay (Fig. 5j). The number of TUNEL-positive (apoptotic) cells for LAcDX (P < 0.001), LMPS@AcDX (P < 0.05) and HMPS@AcDX (P < 0.001) was significantly smaller than those of SCI group (Fig. 5k). In comparison to LMPS@AcDX (P < 0.001) and LAcDX (P < 0.05) groups, HMPS@AcDX further reduced the number of TUNEL-positive cells.

**High MPS-loaded microspheres inhibit gliosis, attenuates inflammation infiltration, and protects axon and myelin sheath.** To further understand the therapeutic outcomes of HMPS@AcDX microspheres after SCI, we examined the density or status of astrocytes, microglia, neurons and axons. All these cells play roles in the injury of spinal cord after the mechanical disruption[27,28].

The activation of astrocytes and microglia near the lesion site was evaluated by immunostaining of glial fibrillary acidic protein (GFAP; in green)[29] and CD68 (in red)[30], respectively (Fig. 6a). In SCI group, GFAP immunoreactivity in proximity to the injury site was higher than that located further from the damaged area (33.2 ± 13.2%, Fig. 6b), which is consistent with the SCI-associated local astrogliosis. As compared to SCI group, the treatment with MPS, HAcDX or LMPS@AcDX could not significantly inhibit the increase of GFAP immunoreactivity. Regarding the HMPS@AcDX group, the GFAP intensity in proximity to the injury site was only 6.0 ± 3.2% higher than that located further from the traumatic lesion, and it is significant (P < 0.01) lower than that of LMPS@AcDX group. At day-28 after injury, the density of CD68-positive microglia in the lesion area for LAcDX (560 ± 164 per mm²) was significantly (P < 0.01) smaller than that in SCI (1012 ± 350 per mm²) group (Fig. 6b). HMPS@AcDX further significantly (P < 0.01) reduced the number of microglia in the traumatic lesion area compared to rats in LAcDX group (Fig. 6b). Morphologically, regions proximity to the injury site were characterized by hypertrophic

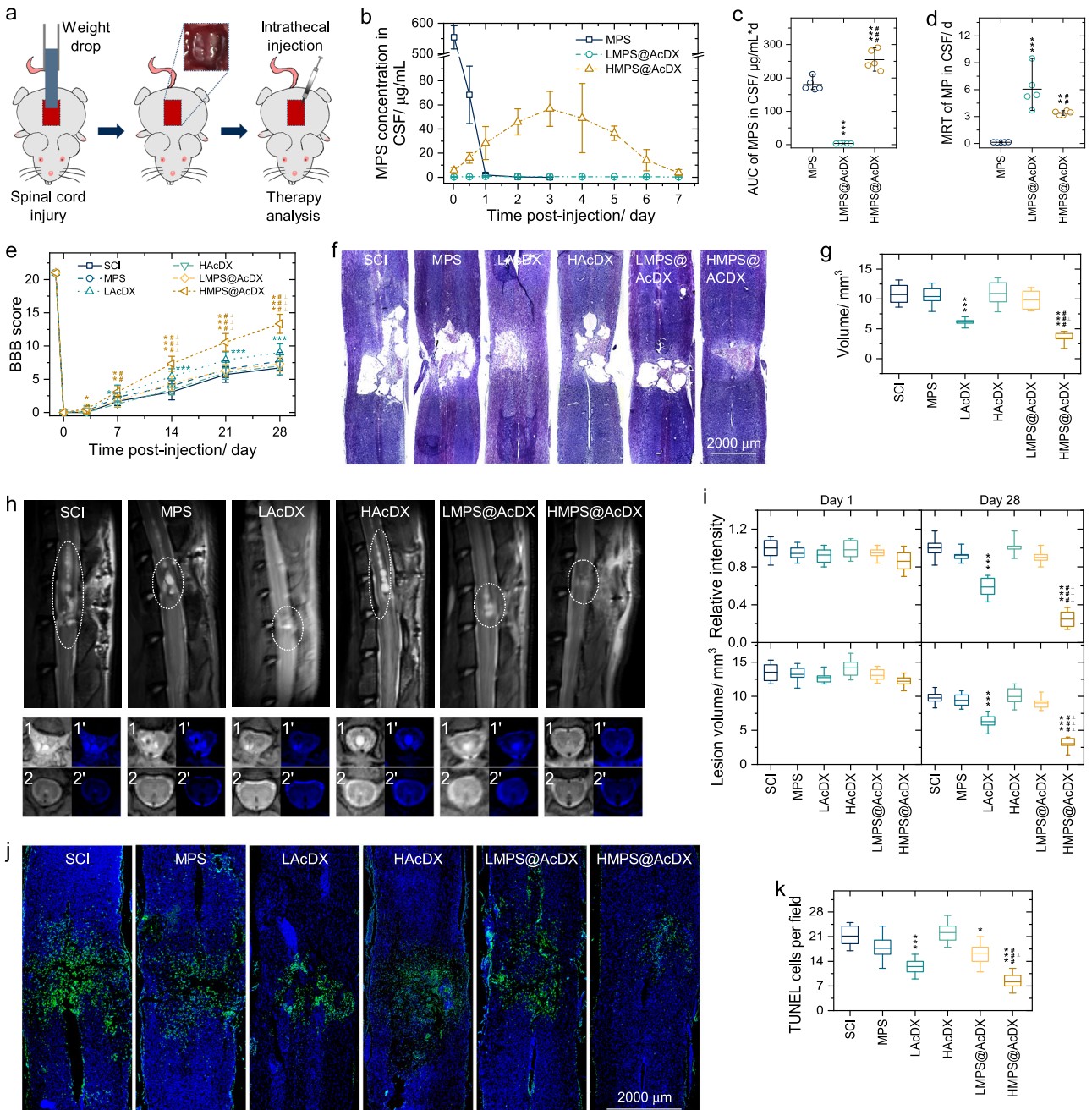

**Fig. 5 HMPS@AcDX microspheres improve motor function and reduce spinal cord edema. a** Microspheres were intrathecally injected within 5 min after weight drop at T10 level. **b–d** MPS concentration in cerebral spinal fluid (CSF) as a function of time (**b**), and the corresponding area under curve (AUC; **c**) and mean residence time (MRT; **d**). $n = 5$ rats per time point; every rat was only sampled once. **e** Rats were functionally graded up to 28 days post-injury by BBB grading scale ($n = 10$, 7, 8, and 9 for SCI/MPS/LAcDX, HAcDX, LMPS@AcDX and HMPS@AcDX, respectively. Data are presented as mean values ± SD for (**b–e**). **f** Sagittal spinal cord sections stained with Nissl display the injury area of spinal cord at day-28 post trauma. **g** Comparison of the lesion volume after treatment ($n = 5$ per group). **h** Sagittal and axial spinal cord T2 weighted images at day-28 after injury. (1) & (1′) are axial images in injury epicenter, (2) & (2′) are axial images in far-injury area, (1′) & (2′) are axial color maps transformed from (1) and (2) normal axial images and the blue color represents the edema signal. **i** Intensity ratio for the signal of lesion to normal spinal cord and cord volume determined for consecutive slices at day-1 and day-28 post injury ($n = 5$ per group). **j** TUNEL-positive apoptotic cells (in green) in sagittal spinal cord sections at day-1 post trauma. The nuclei of all cells were stained with DAPI (in blue). **k** Comparison of the number of TUNEL-positive cells after treatment ($n = 6$ per group). Box plots show the minimum value, the first quartile, the median, the third quartile, and the maximum value. The intervention groups were compared with SCI group (*); HMPS@AcDX group was compared with LMPS@AcDX group (#) and LAcDX group (⊥); *, ⊥$P < 0.05$, **, ##$P < 0.01$, ***, ###, ⊥⊥⊥$P < 0.001$. Statistical significance was analyzed using one-way ANOVA followed by Fisher's post-hoc test. Exact $P$ values are given in the Source data file.

astrocytes (GFAP positive). However, peritraumatic astrocytes in HMPS@AcDX group were more morphologically indistinguishable from astrocytes located distant to the injury site (Fig. 6c).

Chondroitin sulfate proteoglycans (CSPGs) are considered as one of the principal inhibitors for axon regeneration[31]. After SCI, these proteoglycans can be produced by astrocytic scars or diverse cells in SCI lesions including pericytes, fibroblast lineage cells and

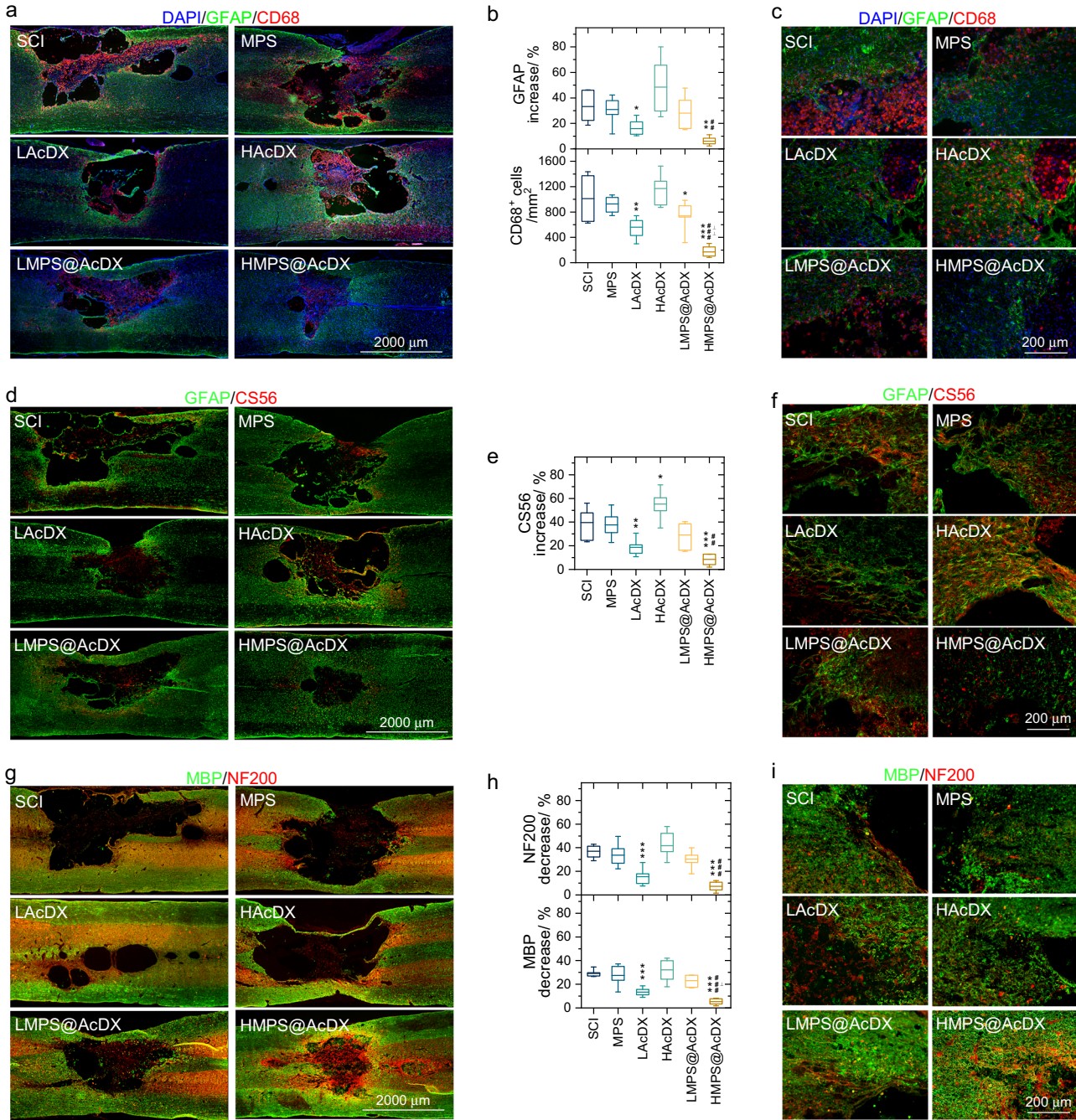

**Fig. 6 HMPS@AcDX inhibits gliosis, attenuates the activation of microglia, suppresses CSGP production, and protects axon and myelin sheath.**
**a** Representative immunohistochemical staining of GFAP (in green) and CD68 (in red) in longitudinal sections of injured spinal cord at day-28 post injury. The nuclei of all cells were stained with DAPI (in blue). **b** Semi-quantification of GFAP intensity and density of microglia in the injured spinal cord. For GFAP intensity, the data are plotted as the relative ratio of the immunoreactivity near the injury site compared with that in distant area. (*n* = 6 per group). **c** Representative immunohistochemical staining of GFAP (in green) and CD68 (in red) adjacent to the lesion area. **d** The deposition of chondroitin sulfate proteoglycans (CSPGs) determined by CS56 antibody (in red) at day-28 after injury. **e** Semi-quantification of CS56 intensity increase in the traumatic lesion area. The data are plotted as the relative ratio of the immunoreactivity near the injury site compared with that in the distant area. (*n* = 6 per group). **f** Enlarged immunohistochemical staining of GFAP (in green) and CS56 (in red) adjacent to the lesion area. **g** Immunohistochemical staining of MBP (in green) and NF200 (in red) in injured spinal cord at day-28 post trauma. **h** Semi-quantification of NF200 and MBP intensity after spinal cord injury therapy. The data are plotted as the relative ratio of the immunoreactivity near the injury site compared with that in distant area. (*n* = 6 per group). Box plots show the minimum value, the first quartile, the median, the third quartile, and the maximum value. **i** Representative immunohistochemical staining of MBP (in green) and NF200 (in red) adjacent to the lesion area. The intervention groups were compared with the SCI group (*); HMPS@AcDX group was compared with the LMPS@AcDX group (#) and LAcDX group (⊥); *, ⊥ $P < 0.05$, **, ##, ⊥⊥ $P < 0.01$, ***, ### $P < 0.001$. Statistical significance was analyzed using one-way ANOVA followed by Fisher's post-hoc test. Exact *P* values are given in the Source data file.

inflammatory cells[32]. We examined cellular production of CSPG (CS56 antibody, in red) and GFAP (in green) (Fig. 6d). Total CSPG levels were, as expected, higher in SCI lesions that those located distant (1 cm) to the injury site (Fig. 6e). After treatment with HMPS@AcDX microspheres, CSPG level was significantly ($P < 0.001$) reduced by ablation or disruption of astrocytic scar formation. Additionally, CSPG deposition was most highly expressed in lesioned tissue of HAcDX group (Fig. 6e), suggesting the adverse effect of high dose bare AcDX microspheres on the regeneration of axons. As shown in Fig. 6f, area adjacent to the lesion site was filled with CSPG-positive, GFAP-negative cells. This finding shows that non-astrocyte cells in SCI lesions produce substantive amounts of CSPGs.

Myelin surrounds nerve cell axons to increase the rate at which electrical impulses is passed along the axon[33]. Neurofilaments are qualified as potential surrogate markers of damage to neuron and axon[34]. To identify the demyelination of residual or regenerated axons, we double-stained 200 kDa subunit of neurofilament (NF200) and the myelin basic-protein marker (MBP; Fig. 6g). The NF200 intensity near the peritraumatic area was $37.1 \pm 5.5\%$ lower than that located further from the traumatic lesion for the SCI group (Fig. 6h). In comparison with SCI group, a significant decrease in NF200 intensity adjacent to the injury site was observed for LAcDX ($P < 0.001$) and HMPS@AcDX ($P < 0.001$) groups. We discovered similar changes toward the intensity of MBP near the lesion area after the therapy with LAcDX and HMPS@AcDX. As shown in the enlarged immunohistochemical staining images, MBP continuously wrapped around the axon (NF200 positive) near traumatic lesion and in injury epicenter (Fig. 6i and Supplementary Fig. S10), suggesting the suppression of demyelination of axons by LAcDX and HMPS@AcDX microspheres. In HMPS@AcDX group, NF200-positive axons and myelinated nerve fibers appeared at the dorsal caudal or the dorsal rostral side, adjacent to the lesion area. As compared to those in HMPS@AcDX group, significantly fewer NF200-positive or myelinated nerve fibers were found in the lesion site of the LAcDX group.

## Discussion

We developed microspheres with high content of therapeutics for drug delivery. The formation of such nano-in-micro structured microspheres relies on the controlled precipitation of drug molecules in droplets prior to the solidification of polymers. Such in-droplet precipitation process is successfully realized by droplet-microfluidics due to its capability in precisely controlling the formation and solidification of droplets[35]. The aggregation-induced emission phenomenon of TPE was employed to visually verify the hypothesis of in-droplet precipitation. The versatility of in-droplet precipitation was demonstrated by engineering microspheres with a variety combination of polymers and therapeutics. All the obtained microspheres showed high mass fraction of therapeutics. Taking MPS@AcDX microspheres for example, a loading degree as high as 63.1 wt% was achieved.

The in-droplet precipitation was triggered by mass diffusion from the droplet to the surrounding outer fluid. Because of the difficulty to experimentally track mass diffusion process at the molecule level[18], we used numerical simulation to perform this task. In accordance with the experimental results, the numerical simulation indicated that drug indeed oversaturated and precipitated inside droplets during their solidification process. Parameters that can lead to earlier drug saturation in the droplets, such as higher initial MPS concentration and lower initial dimethyl sulfoxide ratio in the inner fluid, contributed to simultaneously enhanced loading degree and encapsulation efficiency in prepared microspheres. The fast precipitation of

MPS considerably decreased the duration time of initial high drug concentration in the droplets. Furthermore, in-droplet precipitation strategy greatly reduced the amount of dissolved drug molecules in droplets and, therefore, decreased the MPS concentration gradient between the droplet and outer fluid. The reduced MPS concentration gradient weakened the driving force for MPS diffusion, leading to enhanced drug loading degree and encapsulation efficiency in obtained microspheres. Overall, in-droplet precipitation of drug molecules is a prerequisite to obtaining not only nano-in-micro structured microspheres with high mass fraction of therapeutics, but also highly efficient drug microencapsulation.

The compatibility between drug and polymer, which refers to their miscibility, plays a vital role in determining the drug loading capacity[6–8]. The limited compatibility of a given drug in a specific polymer usually leads to low drug loading in the prepared polymer micro/nano-particles. In other words, high drug loading can only be achieved in very specific combinations of drug and polymer molecules. Several approaches have been developed to enhance the drug-polymer compatibility, such as forming polymer-drug conjugates, and utilizing electrostatic attraction and donor−receptor coordination between drug and carrier molecules[9–11]. All these approaches, however, demanded specific molecule features simultaneously for both drug and polymer molecules, which therefore limits their application in the engineering of drug-loaded microspheres. Our in-droplet precipitation approach is a versatile platform to prepare microspheres with a high mass fraction of therapeutics. After in-droplet precipitation, a mass of drug nanoparticles can be embedded in the polymeric microspheres, leading to the high mass fraction of therapeutics. Most importantly, the demanded molecule features for both drug and carrier molecules can be greatly weakened or even avoided. The only requirement is the low solubility of drug molecules in an emulsified solvent, which can be significantly improved by a cosolvent that is miscible with both the emulsified solvent and water. The commonly used emulsified solvent include ethyl acetate, dichloromethane, dimethyl carbonate, and chloroform. Dimethyl sulfoxide, dimethylformamide, acetonitrile, acetone, methanol, and ethanol are commonly used cosolvents. Due to a wide choice of emulsified solvents and cosolvents, a variety of drugs are expected to meet the solubility requirement, and therefore to be efficiently encapsulated by our in-droplet precipitation approach.

In our study, the encapsulation of high content of drug in the microspheres is ascribed to in-droplet precipitation, which is rather independent of the interaction between drug and polymer molecules. High drug loading was achieved for microspheres with quite free combinations of drug and polymer molecules. Compared to homogeneous mixture of drug and polymer molecules, the formation of drug nanoparticles reduces the interaction between drug molecules and polymer chains. The in-droplet precipitation strategy enables more drug molecules to be encapsulated, without changing the intrinsic compatibility between drug and polymer molecules. Therefore, the developed in-droplet precipitation approach can mitigate the limitation of conventional compatibility-dependent drug loading in microspheres. It should be noted that although our approach increases the drug encapsulation efficiency in microspheres for about 3–26 times (Fig. 2e, Supplementary Fig. S1c), the achieved encapsulation efficiency, in the range of 6.1–71.3% (Fig. 2e), is still significantly less than 100%. This result suggests that the in-droplet precipitation approach can mitigate but not completely eliminate the effects of compatibility between drug and polymer molecules on drug loading.

In principle, such nano-in-micro structured drug-loaded polymer microspheres can be prepared by encapsulating pre-

formed drug nanoparticles in polymer matrix *via* emulsion method or spray drying method. For drug nanoparticles, their engineering approaches can be generally classified into top-down and bottom-up. Bottom-up methods yield drug nanoparticles through nucleation[36] or self-assembly[37]. Top-down approaches, such as milling[38] and high pressure homogenization[39], involve the size reduction of bulk materials to nanoscale, requiring high energy or pressure input. In both top-down and bottom-up approaches, the preparation of drug nanoparticles needs a large amount of work to optimize various parameters, such as the types of solvents and stabilizers[40]. For this pre-prepared drug nanoparticle strategy, one major challenge is to efficiently disperse drug nanoparticles inside emulsified solvents[41].

As a result of the controlled in-droplet precipitation process, the freshly formed drug nanoparticles are immediately embedded within the solidified polymer matrix (Fig. 3b, c). Our approach avoids formulation optimization to stabilize drug nanoparticles in emulsified solvents, without any stabilizers. This makes the in-droplet precipitation approach universally applicable to the production of a variety of drug nanoparticles incorporated nano-in-micro spheres. After encapsulating drug nanoparticles within polymer matrix, the obtained nano-in-micro structure harnesses the controlled drug release feature of polymeric microspheres and the high drug loading degree of drug nanoparticles.

Bulk emulsion approaches (magnetic stirring and homogenization) failed to prepare microspheres with high drug loading degree as illustrated by the formation of many micro-sized free drug crystals outside microspheres (Supplementary Figs. S11–S14). In bulk approaches, the strong and inhomogeneous turbulent mixing leads to fast diffusion of dimethyl sulfoxide from droplets to the external aqueous phase. The fast diffusion of dimethyl sulfoxide results in rapid growth of drug crystals[42], which may break the droplets containing large drug crystals under the high shear force during emulsification. The crystals outside microspheres can hardly be washed away from the obtained microspheres due to their poor solubility and slow dissolution rate. In contrast, droplet-microfluidics can control the in-droplet precipitation process and successfully prepare the high drug-loaded microspheres. The low production rate ($\leq 24$ mg/h in this study) of microspheres, though not the focus of this study, remains a challenge for the clinical translation of engineered microspheres. It is encouraging that the throughput of droplet-microfluidics can be increased by orders of magnitude using parallelized microfluidic devices[43,44] and, for example, the production rate of polycaprolactone microspheres reached 277 g/h[44]. The efficient encapsulation of therapeutics into microspheres can bring a lot of advantages. With a certain number of carriers, high drug loading degree means increased dose of drug per administration, less administration frequency, and improved patient compliance. When the dose is fixed, we can efficiently reduce the amount of excipients used, and ultimately, avoid the potential side effects by increasing drug loading degree. The maximum injection volume is limited for many administration routes. For human beings, the recommended volume for subcutaneous injection is < 1.5 mL[45], and < 5.0 mL for intrathecal administration[46]. In terms of rats, the common volume for intrathecal injection is only about 10 μL[47]. Therefore, in such a small administration volume, only drug delivery systems with high mass fraction of therapeutics can ensure the delivery of required dose of drug.

High drug loading degree could adversely affect the stability and release profiles of the obtained systems. It is a challenge to simultaneously achieve a relatively high drug loading, acceptable stability and controlled release profiles[5]. In this study, all types of microspheres showed no obvious changes in solid-state for 12 months and, meanwhile, successfully enabled the gradual release of drug molecules. For example, MPS@AcDX microspheres with drug loading as high as 63.1 wt% can still sustain the

release of MPS for around 7 days. Regarding the obtained high drug-loaded microspheres, their long-term stability and controlled drug release profiles are likely to come from their nano-in-micro structure. The drug nanoparticles in microspheres are segregated from each other by the polymeric matrix. The polymer matrix acted as a barrier for the drug diffusion, which therefore not only inhibited the crystallization of encapsulated drug nanoparticles, but also avoided burst release of drug molecules.

High dose intrathecal administration of MPS has been clinically used for acute SCI therapy[23]. This treatment is controversial due to no or modest improvements in neurological recovery and the risks of severe side effects[48,49], although the 2017 AOSpine guideline continues to recommend high dose MPS treatment within 8 hours post injury[50,51]. Most of the side effects of MPS therapy are related to the high dose MPS which may be detrimental to tissue protection and recovery, and the relatively modest neurological gains indicate inefficient dosing to the injured site[48]. To demonstrate their therapeutic efficacy, we employed nano-in-micro structured HMPS@AcDX microspheres for local sustained delivery of MPS toward SCI therapy. One single injection of about 630 μg HMPS@AcDX in the limited volume (10 μL) can efficiently deliver the required amount of MPS (400 μg). The amount of AcDX materials in HMPS@AcDX was 432-time less than that of LMPS@AcDX, when MPS dose was fixed at 400 μg.

With such high mass fraction of MPS, HMPS@AcDX reduced the MPS concentration in the first ~12–24 h and maintained a higher and more stable MPS concentration for the next 6 days compared to free MPS (Fig. 5b). The lower MPS concentration provided by HMPS@AcDX in the first ~12–24 h may avoid the detrimental effects of high dose MPS on tissue protection and recovery. Moreover, HMPS@AcDX significantly increased the area under curve and mean residence time of MPS in cerebral spinal fluid (Fig. 5c, d), suggesting an improved bioavailability of MPS at the site of injury compared to free MPS. Our in vivo data demonstrated that the locally sustained release of MPS from HMPS@AcDX conforms the therapeutic need for SCI therapy. Benefiting from the high drug loading degree and controlled release of MPS, HMPS@AcDX inhibited gliosis, attenuated the activation of microglia, suppressed CSGP production, and protected axon and myelin sheath in injured spinal cord. Ultimately, HMPS@AcDX microspheres efficiently improved motor function after SCI. By comparing the therapeutic outcomes of HMPS@AcDX with all the other groups, we can conclude that the high loading and controlled release of MPS are indispensable for SCI therapy. HMPS@AcDX has demonstrated the potential to improve the bioavailability of MPS at the injured site while minimizing side effects and promote the clinical application of MPS.

Moreover, we performed further studies to compare the therapeutic outcomes of high MPS loaded microspheres in immediate treatment (within 5 min after the injury) and delayed treatment (4 h after the injury). HMPS@AcDX was evaluated, and SCI, MPS and LAcDX were used as control groups. We assessed the therapeutic effects in terms of motor function recovery (Supplementary Fig. S15), spinal cord edema reduction (Supplementary Fig. S16), gliosis inhibition and inflammation infiltration attenuation (Supplementary Fig. S17), and axon and myelin sheath protection (Supplementary Figs. S18, S19). Overall, for both immediate treatment and delayed treatment, HMPS@AcDX had significant therapeutic efficacy in SCI therapy and its therapeutic efficacy was significantly higher than that of LAcDX. There was no significant difference in the therapeutic efficacy between immediate treatment (within 5 min after the injury) and delayed treatment (4 h after the injury).

In groups with high dose AcDX (4000 μg), including HAcDX and LMPS@AcDX, 5 out of 20 animals died after the treatment.

By contrast, only 1 out of 40 animals died after the treatment with microspheres in low dose (630 µg; LAcDX and HMPS@AcDX groups). The exact reason for the death of animals treated with microspheres in high dose would need further investigations. It might be due to the too low centration of glutamate and calcium ions in CSF caused by the sequestering capacity of AcDX microspheres toward glutamate and calcium ions[24]. With the administration of high dose AcDX microspheres (4000 µg), the calcium ion concentration in CSF is expected to be much lower than the normal level in CSF. Too low concentration of calcium ions may have adverse influence on the neuronal excitability and consequently adversely affect the function of spinal cord[52] and central respiratory control system[53].

All these findings highlighted the clinical potentials and translational possibilities of HMPS@AcDX, rather than LMPS@AcDX. The therapeutic efficacy of polymer microspheres is mainly determined by the pharmacokinetics of delivered drugs. Human beings and small animals are different in the volume and replenishment rate of cerebrospinal fluid, which are expected to affect the pharmacokinetics of MPS released from the microspheres. Therefore, the drug release kinetics of HMPS@AcDX microspheres need to be further optimized to achieve desired pharmacokinetics of MPS and consequently satisfied clinical therapeutic efficacy in humans.

In summary, we have developed an in-droplet precipitation method to prepare nano-in-micro structured microspheres with high content of payloads. This method is applicable to a wide variety of polymers and drugs, and it mitigates the compatibility limitation toward drug encapsulation for polymeric drug delivery systems. The formation of microspheres with high drug loading lies in sequential solidification of drug and polymer in the droplets, and such process is realized by well controlled generation and solidification of emulsion droplets in a developed microfluidic device. The nano-in-micro structure enables not only the gradual release of payloads from obtained microspheres, but also the long-term solid-state stability of the encapsulated drug molecules. The high drug-loaded AcDX microspheres deliver a high dose of MPS within the limited administration volume by just one single intrathecal injection and remarkably reduce the amount of administered materials/excipients. The controlled release of MPS from high drug-loaded microspheres contributes to improved therapeutic efficacy and reduced side effects in SCI therapy.

## Methods

**Synthesis and characterization of AcDX**. The synthesis procedure of AcDX was the same as described elsewhere[54]. Briefly, dextran (1.00 g, molecular weight ~10,000 Da, Sigma-Aldrich, USA) was dissolved in anhydrous dimethyl sulfoxide (10 mL, Sigma-Aldrich, USA) and purged with dry N₂. Afterward, pyridinium p-toluenesulfonate (15.6 mg, Sigma-Aldrich, USA) was added, followed by 2-methoxypropene (3.4 mL, Sigma-Aldrich, USA). The reaction was maintained at a slightly positive pressure of dry N₂ for 4 h, during which the whole system was sealed to prevent the evaporation of 2-methoxypropene. The reaction was quenched with triethylamine (0.5 mL, Sigma-Aldrich, USA), and the reaction product was precipitated in water (100 mL, pH 8). After centrifugation (10 min, 20,000 × g), the resulting pellet was washed twice thoroughly with water (100 mL). Residual water was removed by freeze dryer for 48 h, yielding AcDX. The ¹H NMR spectra (Supplementary Fig. S20) of bare dextran and AcDX were recorded at room temperature using Bruker Ascend spectrometer (400 MHz).

**Fabrication of flow-focusing microfluidic devices**. Borosilicate glass cylindrical capillaries and microscope slides were used to assemble the microfluidic flow-focusing devices[55]. Firstly, the inner capillary (outer diameter 1.00 mm; 1B100-6, World Precision Instruments, USA) was tapered using a micropipette puller (P-97, Sutter Instrument, USA), and then enlarged to 0.20 mm. Subsequently, the tapered capillary was inserted into the right end of the outer capillary (inner diameter 1.12 mm; TW150-6, World Precision Instruments, USA). A transparent epoxy resin (5 minute® Epoxi, Devcon, USA) was used to seal the microfluidic devices. Syringes were connected to the microfluidic device using polyethylene tubes to allow the independent control of the inner and outer fluids by pumps (PHD Ultra, Harvard Apparatus, USA).

**Engineering of bare polymeric microspheres**. Bare AcDX and PLGA microspheres were prepared by a flow-focusing microfluidic device. AcDX (10 mg mL⁻¹) and PLGA (20 mg mL⁻¹; 5002A, PURAC, Netherlands) were dissolved in a mixture of ethyl acetate-dimethyl sulfoxide and served as the inner fluid. Poloxamer 407 (P-407) (BASF, Germany) solution (10 mg mL⁻¹, pH 7.4) was used as the outer fluid. In the flow–focusing geometry[56,57], the inner fluid is flowed inside the outer capillary and the outer fluid is flowed between the inner and outer capillary in the opposite direction. The inner fluid (0.8 mL h⁻¹) was focused by the outer fluid (10 mL h⁻¹) through the narrow orifice of the tapered inner capillary. The outer fluid exerts pressure and shear stress that force the inner fluid into a narrow thread, and generates droplets inside the inner capillary. The formed oil-in-water emulsion droplets were solidified via the diffusion of solvents from droplets to the surrounding outer fluid. At last, the collected microspheres were washed triple thoroughly with water and then freeze dried (CoolSafe 110, LaboGene APS, Denmark).

**Engineering of cargo-loaded microspheres**. Toward the engineering of cargo-loaded microspheres by microfluidics, AcDX (10 mg mL⁻¹) or PLGA (20 mg mL⁻¹) was dissolved in a mixed solvent of ethyl acetate-dimethyl sulfoxide, serving as the inner fluid. ATV (30 mg mL⁻¹), MPS (30 mg mL⁻¹), HCT (60 mg mL⁻¹) and TPE (10 mg mL⁻¹) was dissolved in the inner fluid. The volume ratio of dimethyl sulfoxide in inner fluid was at 2%, 10%, 5% and 10% (v/v) for ATV, MPS, HCT and TPE, respectively. The rest fabrication procedure was the same as for the bare AcDX and PLGA microspheres. To prepare cargo-loaded microspheres without using dimethyl sulfoxide, excess amount of ATV, MPS, HCT, or TPE was added into the inner fluid. This solution was stirred for overnight at room temperature. After centrifugation (10 min, 16,100 × g), the obtained supernatant was used as the inner fluid. The rest fabrication procedure was the same as for the bare polymeric microspheres. Drug-loaded microspheres were also prepared by bulk method. For bulk method, the oil and water phases were the same as the inner and outer fluids used for microfluidics, respectively. The oil phase (0.8 mL) was added below the liquid surface of the water phase (10 mL) under magnetic stirring (500 rpm; LLG-uniSTIRRER 3, LLG-Labware) or homogenization (5000 rpm; Proxxon FBS 240/E with dispersing tool).

**Real-time observation of in-droplet precipitation process**. The droplets containing TPE were prepared by flow-focusing microfluidics as described above. Spiral quartz glass capillary tube with an inner diameter of 0.8 mm was connected to the flow-focusing microfluidic device. The whole device was placed under ultraviolet irradiation (365 nm, about 80 mW cm⁻²). Images and videos of the solidification process of TPE-containing droplets were taken by a digital camera. The precipitation process of ATV, MPS and HCT was observed under phase contrast light microscope (Leica DMi1, Leica Microsystems, Germany). Droplets were produced using a microfluidic co-flow device, which was the same as the microfluidic flow-focusing device. The inner capillary for microfluidic co-flow device was only enlarged to 0.08 mm. The flow rates of inner and outer fluid were 0.2 and 5 mL h⁻¹, respectively. After the formation of droplets, both inner and outer flows were stopped. The droplets were observed in static outer fluid under phase contrast mode of the light microscope.

**Loading degree, encapsulation efficiency, and drug release studies**. The drug loading degree of microspheres is defined as the mass ratio of the encapsulated drug to the drug-loaded microspheres. The drug encapsulation efficiency of microspheres is defined as the mass ratio of the encapsulated drug to the initial drug added. The microspheres were dissolved in acetonitrile to release all the encapsulated drug molecules. The drug release profiles from obtained microspheres were studied in PBS (pH 7.4) at 37 °C with shaking at 100 rpm. Bare ATV, MPS and HCT powders were used as control groups. The release studies were performed under the sink condition for all the drug molecules. The mass of samples and volume of release media are listed in Table S2. At predetermined time intervals, 200 µL media were collected from each sample and the release media was replaced with 200 µL fresh media. The collected media were firstly centrifuged (16,100 × g, 3 min), and then the drug concentrations in the supernatant were measured. The amount of drug was quantified by Agilent 1100 high performance liquid chromatography. The mobile phase was composed of acetic acid (0.5% v/v, pH 3) and acetonitrile with a volume ratio of 35:65 for ATV and a volume ratio of 58:42 for MPS. A volume ratio of 80:20 and 10:90 between phosphate buffer (0.02 M NaH₂PO₄, pH 3) and acetonitrile was used for HCT and TPE, respectively. The flow rate was 1 mL min⁻¹, and temperature of mobile phase was set at 25 °C. The wavelength used for ATV, MPS, HCT and TPE quantification was 244, 243, 275, and 238 nm, respectively. A Discovery® C18 column (4.6 × 150 mm, 5 µm, Supelco Analytical, USA) was used as stationary phase, and the injection volume of all the samples was 20 µL.

**Internal structure of the obtained microspheres**. The surface and cross-section of microspheres were observed by scanning electron microscopy (Quanta 250 FEG,

FEI, USA). The cross-section of microspheres was obtained by cryomicrotome. The particles were platinum sputtered before imaging. The particle size of microspheres was determined based on the scanning electron microscopy images, and at least 100 particles were analyzed. AcDX and MPS@AcDX microspheres were embedded in epoxy resin. Ultrathin sections (60 nm) were cut and examined with transmission electron microscopy (JEOL 1400, JEOL, Japan) using an acceleration voltage of 80 kV. The ultrathin sections of MPS@AcDX were immersed in water (pH 7.4) for few hours and then left to dry at room temperature. After drying, these washed ultrathin sections of MPS@AcDX were examined again with transmission electron microscopy.

**Physicochemical characterization of the obtained microspheres.** The crystalline state of drugs was studied by X-ray powder diffraction (Malvern Panalytical X'Pert, Cu $K\alpha$) and differential scanning calorimetry (AG-DSC823e, Mettler Toledo, Switzerland). For X-ray powder diffraction, data were collected over the $2\theta$ range from 7.5° to 27.5° using a step size of 0.013° with 200 seconds per step, and each sample was scanned for 8 times. For differential scanning calorimetry, the samples were heated from 25 to 275 °C under nitrogen flow, and the heating rate was 10 °C min$^{-1}$. STARe software was used for collecting the measured data. The long-term stability of microspheres was performed at 25 °C and 40% relative humidity. After storage for 6 and 12 months, the samples were examined by X-ray powder diffraction. The physical mixtures (PM) were prepared by mixing bare ATV, MPS or HCT powder and bare AcDX or PLGA microspheres according to the loading degree of each drug-loaded microspheres.

**Numerical simulation of droplet solidification process.** The droplets are consisted of organic solvents (ethyl acetate and dimethyl sulfoxide), polymers, drug molecules, and water. As a macromolecular, the diffusion of polymer molecules is neglected.

The concentration distribution of each component inside the droplet is calculated by Fick's law Eq. 1:

$$\frac{\partial c_i}{\partial t} = \frac{\partial}{\partial x_i}\left(D_i \frac{\partial c}{\partial x_j}\right) + I_i \tag{1}$$

where $C_i$ (mol m$^{-3}$) is the molar volume concentration of each component in the droplets; $D_i$ (m$^2$ s$^{-1}$) represents the diffusion coefficient of each component in water or polymer solution; $I_i$ (mol m$^{-3}$ s) is the source-sink term of each component.

$D_i$ of each component in the droplet is highly concentration dependent[17]. The solutes, i.e., polymer and drug hinder the diffusion of the components inside the droplet. The volume fraction of solutes in the droplet ($phi\_p$) is applied to correct the $D_i$, and the relationship between $D_i$ and $phi\_p$ is suggested in Eq. 2[58]:

$$D_i = a \times 10^{b+c \times phi\_p} \tag{2}$$

where parameters $a$, $b$, and $c$ are constants. $a$ is calculated by method proposed by Evans et al.[59], and $b$ and $c$ are calculated to be –9 and –4 in this study according to the principle proposed by Li et al.[17].

Solutes in the droplet are polymer and drug. Drug in the droplet exists in either dissolved state or precipitated state. Only the dissolved drug is considered to hinder the diffusion of solvents. The relationship between $D_i$ and the concentration of dissolved drug ($C_{d\_drug}$) is considered to be as the same as the relationship between $D_i$ and the polymer concentration ($C_{polymer}$). Therefore, $phi\_p$ can be calculated by Eq. 3:

$$phi\_p = \frac{V_{d\_drug} \times C_{d\_drug}/rho_{drug} + V_{polymer} \times C_{polymer}/rho_{polymer}}{V_{total}} \tag{3}$$

$I_{water}$ is mainly due to the convective mass transfer between droplet and surrounding water, therefore it is necessary to set convection conditions at the boundary of droplet, for which the convective mass transfer coefficient ($k_c$) is a key parameter.

$k_c$ is formulated using an empirical equation for spherical particles suspended in an agitated vessel[17], and simplied as Eq. 4:

$$k_c = k_c \times step1(State) \tag{4}$$

where step1 is default step function in COMSOL, and State is a while statement for judging if the water is saturated in the droplet.

$I_i$ of other components can be calculated by mass transfer equation. Taking drug MPS for example, the outflow flux of MPS caused by its diffusion at the boundary of droplet is bndFlux_C_MPS in COMSOL. The outflow flux of MPS at the boundary of droplet is intop2(-bndFlux_C_MPS*M_MPS*theta), where intop2 is integral operator at the boundary, M_MPS is the molar mass of MPS. theta is the amount of dissolved drug divided by the total amount of drug, which is determied in the following subsection.

The solubility of MPS (t_MPS) in ethyl acetate-dimethyl sulfoxide with different volume ratio of dimethyl sulfoxide (w_dimethyl sulfoxide) was experimentally measured (Supplementary Fig. S1), and the relationship between t_MPS and w_dimethyl sulfoxide is fitted to be in Eq. 5:

$$t\_MPS = 2.57092 + 1775.56388 \times w\_dimethyl\ sulfoxide^{1.4705} \tag{5}$$

where w_dimethyl sulfoxide is in the range of 0.00–0.10. When $m_{MPS}$ is more than

t_MPS in the droplet, MPS will precipitate. The mass of precipitated drug is defined as crystal = $m_{MPS}$-t_MPS. Then, the percentage of dissovled drug is theta = 1-crystal/$m_{MPS}$.

With the calculated $C_i$, the mass of each component ($m_i$) is obtained by volumetric integration of the droplet. The volume of each component ($V_i$) is calculated by Eq. 6:

$$V_i = m_i/rho_i \tag{6}$$

where rho$_i$ is the density of each component.

Radius variations are directionless, assuming that the diffusion is isotropic in the radial direction and the droplet shrinks uniformly in the radial direction. We calculated the total volume of droplet ($V_{total}$) at each time by Eq. 7:

$$V_{total} = V_{ethyl}\ acetate + V_{dimethyl}\ sulfoxide + V_{polymer} + V_{drug} + V_{water} \tag{7}$$

The radius of the droplet ($R_{drop}$) can be obtained by Eq. 8:

$$R_{drop} = \sqrt[3]{\frac{0.75 V_{total}}{\pi}} \tag{8}$$

All the parameters and their values used for numerical simulation are listed in Table S3.

**The rat model of spinal cord injury and administration of formulations.** All rat experiments were approved by the Animal Care and Use Committee of Nanjing Medical University. Sprague-Dawley male rats were provided by the Animal Experimental Center of Nanjing Medical University. The procedures were the same as described elsewhere[24]. Briefly, under general anesthetic rats (chloral hydrate, 350 mg kg$^{-1}$ body weight) and sterile conditions, a T10 laminectomy was performed to expose the underlying thoracic spinal cord segment. Next, spinal cord contusion injury was produced using a weight-drop device (RWD Life Science Corp., C4P01-001, China)[60,61]. The exposed dorsal surface of the cord was subjected to weight-drop impact using a 10 g rod (2.5 mm in diameter) dropped from a height of 12.5 mm. The muscles were sutured immediately after formulation administration and the skin was then closed. An antimicrobial agent, sodium ampicillin (80 mg kg$^{-1}$ body weight), was administered daily for 1 week after SCI. The bladders of animals were manually voided three times per day until the reflexive control of bladder function was restored.

The formulations administered in this study included saline (10 µL), free MPS (400 µg in 10 µL), LAcDX microspheres (630 µg in 10 µL), HAcDX microspheres (4000 µg in 10 µL), LMPS@AcDX microspheres (4000 µg in 10 µL) and HMPS@AcDX microspheres (630 µg in 10 µL). After loading into a sterilized 26G Hamilton syringe, all formulations were intrathecally injected within 5 min post-trauma in approximately 1 mm rostral and caudal to the lesion epicenter[62]. After each injection, the needle was maintained in the spinal cord for an additional 2 min to minimize the leakage of the injected formulations.

To further compare the therapeutic effects of immediate treatment (within 5 min after the injury) and delayed treatment (4 hours after the injury). The formulations saline (10 µL), free MPS (400 µg in 10 µL), LAcDX microspheres (630 µg in 10 µL), and HMPS@AcDX microspheres (630 µg in 10 µL) were used. All formulations were intrathecally injected within 5 min post-trauma and at 4 hours post-trauma, respectively, using the method described above.

**Pharmacokinetics of MPS in cerebrospinal fluid.** The cerebrospinal fluid was collected by a 1 mL syringe equipped with a 25 G disposable needle. For every sampling time point, the volume of withdrawed cerebrospinal fluid ranged between 100 and 150 µL for each rat. Each rat was only sampled once. The MPS content in cerebrospinal fluid was analyzed by a liquid chromatography and a triple quadrupole mass spectrometry. An aliquot of 20 µL sample was added with 20 µL mixture of methanol and water (1:1, v/v); then 60 µL internal standard (glipzide in acetonitrile, 100 ng/mL) was added. The mixture was vortexed for 10 min and centrifuged at 3100 × $g$ for 10 min. The liquid chromatography system consisted of a Shimadzu (Shimadzu Co., Japan) liquid chromatography equipped with a binary pump (LC-30AD), an autosampler (SIL-30AC), a column oven (CTO-20A), a system controller (CBM-20A), and a degasser (DGU-20A). The mass spectrometric analysis was performed using a triple-quadrupole (AB SCIEX API-4000; Canada) instrument with an electrospray ionization (ESI) interface. Chromatographic separation was performed on a BEH column (Waters ACQUITY C18; 2.1 × 50 mm, 1.7 µm) at 60 °C. The mobile phase A and B consisted of formic acid (0.1%, v/v) in water and acetonitrile (LC-MS grade), respectively. The column was eluted at a flow rate of 0.6 mL min$^{-1}$ in a gradient program consisting of phase B: 15% (0.00–0.20 min), from 15 to 95% (0.20–0.40 min), 95% (0.40–1.30 min), from 95 to 15% (1.30–1.31 min), 15% (1.31–1.80 min). For MPS, the retention time for the analyte and internal standard (glipzide) were 0.92 and 0.91 min, respectively. The injection volume was 3 µL. The precursor ion and product ion were 373.1 and 343.0 $m/z$, respectively, for MPS, and 444.2 and 319.1 $m/z$ for glipzide.

**Magnetic resonance imaging (MRI).** Each animal was anesthetized with halothane (3–4% induction, 1.5–2% maintenance) in oxygen (0.4 L min$^{-1}$) and nitrogen (0.6 L min$^{-1}$). After anesthesia, each rat was placed on the fixation system in prone position. The experiments were performed on a small animal MRI system

(Bruker BioSpec 7T/20 USR, Germany). The sequence protocol included T2-weighted, 256 × 256 matrix, slice thickness 1 mm, intersection gap 1 mm, echo time (TE)/repetition time (TR) 27/3000 ms, RARE factor 16, and flip angle 90°. T2-weighted images were acquired in the sagittal and axial planes by the ParaVision 6.0.1 (Bruker BioSpec, Germany). The area of the lesioned spinal cord containing hyperintense signal was first manually traced by a blinded observer. A computer-aided software (FireVoxel; CAI2R, New York University, NY) was used for axial images to assess and compare the evolution of hyperintense signal and lesion volume obtained by adding the individual slice areas and multiplying by 1.0 mm slice plus gap thickness. For quantitative analysis, the results were calculated by the intensity ratio for the signal of spinal cord lesion to normal cord far from injury area.

**Assessment of the locomotor capacity**. After spinal cord surgery, the recovery of locomotor capacity was measured by the BBB open field locomotor test procedure[63]. Rats were placed in an open field and were allowed to explore freely for 4 min. Two examiners, who were blinded to the treatment regimen, participated in all open field tests and were positioned across from each other to observe both sides of the rats[25,63]. The animal behaviors of the trunk, tail and hindlimbs were evaluated at day-1 after injury and weekly thereafter for 28 days.

**Histopathological analyses**. For morphometry, the images of every spinal cord tissues were captured by a digital camera before the tissue sections. The spinal cord sections of each group at day-28 post trauma were stained the Nissl substance in the cytoplasm of neurons with Cresyl Violet (FD Neuro Technologies, USA). The spinal cord sections were rinsed in distilled water, and subsequently stained in Cresyl Violet solution for 10 min. After rinsing by distilled water, the sections were differentiated in 95% ethyl alcohol, cleared with xylene, prior to mounting with neutral balsam. Severe disruption of tissue organization and/or the loss of staining were identified as traumatic lesion area. The identified areas in each section was measured by ImageJ software (National Institutes of Health). The lesion volume was obtained by the sum of total lesion area multiplied by distance (about 200 μm) between the sections.

**Tissue processing**. Animals were anesthetized with a lethal dose of chloral hydrate at day-1 and day-28 post surgery. The physiological saline was perfused trans-cardially followed by ice-cold paraformaldehyde (4%, w/v). The T6-L1 segments of the spinal cord containing the lesion site (around 2 cm) were removed, fixed in paraformaldehyde (4%, w/v) for 24 h at 4 °C, and then transferred to sucrose (30%, w/v) PBS solution. The spinal cord tissues were embedded in optimal cutting temperature (OCT) compound and cut into serial longitudinal sections with a thickness of about 18 μm. All slides were stored at −70 °C until further use.

**In situ apoptotic assay-TUNEL staining**. The apoptotic cells in the spinal cord at day 1 post-trauma were identified and quantified by the DeadEnd® fluorometric terminal deoxynucleotidyl transferase-mediated dUTP nick end labeling (TUNEL) system (Promega, USA). After rehydration through a series of graded ethanol, the spinal cord specimens (18 μm) were incubated with protease K at room temperature for 20 min. After immersing in PBS for 5 min, the specimens were re-fixed in formaldehyde (4.0%, w/v; 5 min). Next, the spinal cord specimens were washed triple with PBS and equilibrate in equilibration buffer at room temperature for 10 min. The spinal cord specimens were firstly incubated with the TdT-reaction mixture at 37 °C for 60 min, and then immersed in 2 × SSC for 15 min to stop the reaction. The nuclei of all cells were stained with DAPI. Images were captured and apoptotic cells were characterized by nucleus with green fluorescence. Counts of the TUNEL-positive cells were performed in five different slices.

**Spinal cord tissue immunofluorescence**. Primary antibodies used in this study included anti-200 kDa subunit of neurofilament (NF200; 1:50, mouse IgG1; Abcam, UK), anti-glial fibrillary acidic protein (GFAP; 1:5000, rabbit IgG; Abcam, UK), anti-myelin basic-protein marker (MBP; 1:50, rabbit IgG; CST, USA), anti-chondroitin sulfate (CS56; 1:300, mouse IgG1; Abcam, UK) and anti-CD68 (1:300, mouse IgG1; EMD Millipore Corp., USA). The secondary antibody used was Cyanine3- or fluorescein isothiocyanate-conjugated secondary antibody (1:200, Jackson ImmunoResearch, USA). At first, the spinal cord sections at day-28 after injury were permeabilized for 30 min in Triton X-100 PBS solution (0.3%, w/v) and then blocked by natural goat serum PBS solution (10%, v/v). After that, the specimens were incubated with primary antibodies overnight at 4 °C. The specimens were triple washed by PBS, and then incubated with secondary antibody for 2 h at room temperature. After triple washing by PBS, the nuclei were stained with DAPI in specimens double stained with GFAP and CD68. Images were taken by a fluorescence microscope (AXIO Vert.A1&Imager A2, Carl Zeiss Microscopy GmbH, Germany). For each slide, the lesion was identified as the area lacking staining. We selected 4 fields (720 × 540 μm² for each), including the rostral left and right, and caudal left and right sides of gray matter near the traumatic lesion, as the near injury fields. In contrast, 4 fields at least 10 mm distance from the traumatic lesion were chosen as distant fields[30]. The average intensity of GFAP, NF200, MBP and CS56 for each field was measured by ZEN lite software. The data were expressed as the percentage of intensity increase or decrease in near-injury area in respect to the distant area. In terms of CD68-positive cell counting, the immunostained images were captured at lesion interface and traumatic lesion site. The number of CD68-positive cells/mm² was counted by the ImageJ software.

**Statistics and reproducibility**. Results are expressed as the mean values ± standard deviation (SD) for at least three independent experiments (Microsoft Excel). Images are representative of at least three independent samples. For the determination of statistical significance, one-way analysis of variance (ANOVA) followed by Fisher's post-hoc test or pair-sample $t$-test (Origin 2020, OriginLab, Northampton, USA) were used, as applicable. The levels of significance were set at probabilities of *, #, $\perp P < 0.05$, **, ##, $\perp\perp P < 0.01$, ***, ###, $\perp\perp\perp P < 0.001$.

**Reporting summary**. Further information on research design is available in the Nature Research Reporting Summary linked to this article.

## Data availability

The data generated in this study are available within the article and provided in the Supplementary Information/Source Data file. Source data is available for Figs. 1f–g, 2d–g, 3a–c, e, j, 4, 5b–e, g, i, k, 6b, e, and h and Supplementary Figs. S1b, c, S2, S4, S5, S6, S8, S12, S14, S15, S16b, S17b, S18b, S19b, and S20 in the source data file. Source data are provided with this paper.

## Code availability

All codes used in generating results are available from the corresponding authors upon reasonable request.

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

## Acknowledgements

We acknowledge the financial support from the Academy of Finland (No. 322093 (W.L.), 308742 (D.L.), 317042 (H.A.S.), and 331151 (H.A.S.)), Sigrid Jusélius Foundation (W.L. and H.A.S.), Natural Science Foundation of China (No. 81973266 (D.L.), 51903251 (D.L.), 81772352 (J.F.), 82072437 (J.F.), 81902211 (J.C.), 82030069 (G.Y.), and 81772351 (G.Y.)), Natural Science Foundation of Jiangsu Province (No. BK20190554 (D.L.), BK20190033 (J.F.), and BK20191061 (J.C.)), Natural Science Foundation for Distinguished Young Scholars of Jiangsu Province (No. BK20190033 (J.F.)), Finnish Cultural Foundation (No. 00190634 (W.L. and Y.D.)), Mary and Georg C. Ehrnrooth Foundation (W.L.), China Scholarship Council (P.Q.), HiLIFE Research Funds (D.L. and H.A.S.), and the European Research Council Proof-of-Concept Grant (No. 825020 (H.A.S.)) for financial support. The authors thank Reeta Huhtala (Tissue preparation and histochemistry unit, Department of Anatomy, University of Helsinki) for the microsphere section support, thank Electron Microscopy Unit (Institute of Biotechnology, University of Helsinki) for providing laboratory facilities, and acknowledge CSC-IT Center for Science, Finland, for computational resources.

## Author contributions

W.L., H.A.S., J.F., and D.L. conceived and designed the experiments. W.L., J.C., S.Z., T.H., H.Y., C.T., G.M., Z.Z., T.J., W.L., L.L., Y.B., P.Q., and Y.D. performed the experiments. W.L., J.C., G.Y., H.A.S., J.F., and D.L. analyzed and interpreted the data. W.L. and D.L. wrote the paper. All authors discussed the results, revised, and commented on the manuscript.

## Competing interests

The authors declare no competing interests.

**Additional information**

