## [Peer Review File · Nature Communications]

REVIEWER COMMENTS

Reviewer #1 (Remarks to the Author):

This is a good manuscript that deserves to be published in any of highly respected journals. But this reviewer suggests the following revision before publication.

1. Title

It has been a fashion to use “--- inspired” like “nature-inspired”. But do good scientists really have to be inspired by anything? Microparticles and nanoparticles have been with us for several decades, and does it make any sense that someone has a nanoparticle-loaded microparticle only after inspired by pomegranate? I think scientists should behave like scientists, and avoid at all cost to be like a salesman. If there is anything to blame, it is Nature Journals themselves. They are the ones who promote all those fancy terms and figures for their money-making enterprise. Remove such a non-essential word. You are better scientists than most who try to flatter Nature solely to be published.

2. The preparation method is certainly new and the figures look wonderful. But the authors need to be honest in describing the benefits which are limited, and more importantly, the limitations of the technique. No technique is perfect and this is also. There is no need to promote the method any more than what it actually is. For example, there is a section on “Pomegranate-like microspheres overcome the compatibility limit in microencapsulation”. It may be, but it presents its own problems. Honestly describe its limitations.

3. The contents and figures are all look great, as Nature likes to have. Nature usually does not understand the contents, and they have a tendency to publish as long as figures looks great. This manuscript has all ingredients.

4. Release profiles.

According to Fig 5, drug release lasts for 5-10 days, and it appears that the microparticles cumulated at the cerebral spinal fluid. All good. But the authors need to comment on the possibility that it may be useful in humans. So many experiments have shown some efficacy in small animal studies, but almost all failed in clinical studies. The authors should provide caution in extrapolating this to clinical efficacy.

5. Once the above comments are reflected in revision, this manuscript is ready for publication in Nature. Actually, this manuscript deserves to be published in Science.

Reviewer #2 (Remarks to the Author):

The authors present a microfluidic approach to forming polymer particles that contain drugs and assert that the drug is present in nanoparticle form within the particles.

The connection to biological materials, biomimetics and pomegranates is highly speculative and appears to be an attempt to exaggerate the research. The comparison to a pomegranate is just not needed and is a case of drawing similarities to nature purely for the sake of finding an analogy. If a research study is high quality, there is no need or necessity to draw some unfounded connection to the natural world. The reference to pomegranates should be removed.

Many statements within the submission are purely speculative, untrue or unfounded. For example:

- "biological materials outperform most artificial ones" - there is no relevance here and a generalism like this is unsubstantiated. Clearly, there are no hard wearing alloys formed in nature, no biological equivalent to carbon-fibre, no biological thin plastic films (to use meaningless examples). Such a statement just should not be repeatedly used in a high quality research article.

- The introduction has a large section on biological inspiration that should be removed. I would expect that the analogy to fruit with seeds has come about after the work was conducted and was not an inspiration for the work as much of the work is described in numerous other reports and technologies (see below)

- The introduction refers to a report (REF 8) after the statement

"Among the developed particulate systems, biodegradable polymer microspheres have been exploited as a successful drug delivery platform and received regulatory approvals"

Referring to a review when claiming "regulatory approvals" is poor practice and original source material should be cited that show clear evidence of regulatory approval. The actual paper - Expert Opin. Drug Deliv. 9, 1161-1176 (2012) - states as its Expert Opinion:

"The broad clinical use of microspheres for delivery of therapeutic agents in particular biologics such as proteins has not been realized commercially. The limited availability of biodegradable polymers with a long history of regulatory approval and the challenges in gaining regulatory approval of a new polymer have hindered the development of microspheres for parenteral drug delivery."

Note - "The broad clinical use of microspheres...has not been realized commercially" and "The limited availability of biodegradable polymers with...regulatory approval have hindered the development of microspheres for parenteral drug delivery."

- "superfast in-droplet precipitation" - the term superfast has no basis for comparison or the superlative

- I have no idea of what a "high-performance human architecture" actually is

- "ultrahigh drug loading" - This terminology is unclear and has no comparative basis. For example, solid dispersions for solid-drug nanoparticle formation have been reported with drug loading as high as 80wt % in the solid state (drug relative to [drug+polymer+surfactant] excipients)

- Stability The authors state - "It is worthy to point out that although with high drug loading, all these microspheres are quite stable over the long-term storage (Fig. 5c-h), as confirmed by Xray powder diffractograms. This long-term stability is likely to benefit from the pomegranate like structure of the formed microspheres."

To be clear, 15 days stability is not "long term". Any medicine that is of value would need to be studied under accelerated conditions for many months (one standard would be 6 months at 40 deg C and 75% RH). 15 days is clearly not enough to state that these are stable and there is no attempt to quantify crystallinity changes over this very short period.

The approaches and concepts reported are relatively well known

- Firstly, using the slow (or indeed rapid) diffusion of a solvent with water miscibility to create particles is very well known and I refer the authors to *Angew Chem Int Ed Engl.* 2001 3;40(23):4330-4361. doi: 10.1002/1521-3773 Organic Nanoparticles in the Aqueous Phase-Theory, Experiment, and Use - Horn and Rieger.

- Nanoprecipitation is also well known in the literature, however, the authors do not discuss this

- Microfluid approaches to microspheres is also well know

- The authors appear to have formed what is widely described as a "solid dispersion" - again, a well known approach that has been achieved by spray drying and freeze drying of dispersions, emulsions and solutions for many years. The difference here is that this has been achieved in a microfluidic manner (to be clear, I have never seen anyone feel the need to describe solid dispersions using any fruit analogy...)

- The structures described and shown figuratively in, for example, Figure 1, do not bare any real relationship to pomegranates. They are very similar to numerous reports of solid dispersions however. If the authors wish to see images that are more "pomegranate-like" they need to review the literature - for example *Nanoscale*, 2016,8, 7224-7231 - doi.org/10.1039/C6NR00788K

- The reliance on "like dissolves like", Flory-Huggins and Hansen's parameters is very old thinking.

The authors state

"Based on the general principle of 'like dissolves like', Flory-Huggins interaction parameter (χ_{dc}) is commonly used to estimate drug-carrier material compatibility¹⁸. The Flory-Huggins interaction parameter can be calculated from Hansen's solubility parameter (Table S1) by Eq. 1"

This approach has been quite widely shown to not be representative of the effects of drug and polymer interactions and have been considered to be poorly predictive. For example, I would like to direct the authors to "Like Dissolves Like? A Comprehensive Evaluation of Partial Solubility Parameters to Predict Polymer–Drug Compatibility in Ultrahigh Drug-Loaded Polymer Micelles" *Biomacromolecules* 2019, 20, 8, 3041–3056 doi.org/10.1021/acs.biomac.9b00618. The abstract for this paper concludes

"In general, the Flory–Huggins interaction parameters were less suited to correctly estimate the experimental drug solubilization compared to the Hansen solubility parameters. The latter were able to correctly predict some trend regarding good and poor solubilizers, yet the overall predictive strength of Hansen solubility parameters is clearly unsatisfactory." - note "clearly unsatisfactory". This reference also cites several polyoxazoline polymer particles with drug loadings >50 wt%

Additionally:

- "%" should always state what basis the % is taken (eg w/w %)
- it is not possible to have "432-time less". 432-times is a multiplier. This should be 1/432nd or a fraction of some form.
- The paper seems to switch randomly between reporting crystallinity and describing the process as precipitation. It makes it very unclear what exactly is being reported
- Figure 3 appears to have no identifiable drug particles within the microparticles
- The description of a "polymer pulp" in the following statement is just meaningless and unnecessary
- "Therefore, the release of payloads only needs to cross the polymer "pulp membrane" and almost no burst release has been observed for all the obtained microspheres."

I would strongly encourage the authors to rewrite their manuscript without hype and hyperbole and concentrate on the relatively interesting "controlled in-droplet solidification" - their words - which is a clever combination of microfluidics, solid dispersions and nanoprecipitation. If they had not over hyped and created an needless connection to biomimetics and pomegranates, this would have been an interesting report but as it stands, it is impossible to see through the hyperbole, superlative and distracting comparisons.

I recommend rejection from Nature Communications in its current form but urge the editors to provide the authors with an opportunity to re-write and submit as a quality research article that references the literature and places this work properly in context and without exaggeration.

RESPONSE TO THE REVIEWER'S COMMENTS

Reviewer #1:

This is a good manuscript that deserves to be published in any of highly respected journals. But this reviewer suggests the following revision before publication.

Answer: We thank the reviewer for the recommendation and time spent in reviewing our work. The insightful comments will undoubtedly help us to improve the overall quality of the manuscript. The response to the reviewer's comments is addressed below point-by-point. All the modifications/changes were also introduced in the revised version of the manuscript.

1. Title

It has been a fashion to use “--- inspired” like “nature-inspired”. But do good scientists really have to be inspired by anything? Microparticles and nanoparticles have been with us for several decades, and does it make any sense that someone has a nanoparticle-loaded microparticle only after inspired by pomegranate? I think scientists should behave like scientists, and avoid at all cost to be like a salesman. If there is anything to blame, it is Nature Journals themselves. They are the ones who promote all those fancy terms and figures for their money-making enterprise. Remove such a non-essential word. You are better scientists than most who try to flatter Nature solely to be published.

Answer: We thank the reviewer for the suggestion. We agree that researchers should focus on the scientific contents when presenting their results. To focus more on the scientific contents, we have changed the title to “*An ultrahigh drug loaded microsphere enabled by controlled in-droplet precipitation promotes functional recovery after spinal cord injury*”. In addition, we removed the pomegranate and biological inspiration related statements from the manuscript and rewrote the introduction, which now focuses more on the scientific questions of the work.

2. The preparation method is certainly new and the figures look wonderful. But the authors need to be honest in describing the benefits which are limited, and more importantly, the limitations of the technique. No technique is perfect and this is also. There is no need to promote the method any more than what it actually is. For example, there is a section on “Pomegranate-like microspheres overcome the compatibility limit in microencapsulation”. It may be, but it presents its own problems. Honestly describe its limitations.

Answer: We thank the reviewer for acknowledging the novelty of our method. For the subsection “*Pomegranate-like microspheres overcome the compatibility limit in microencapsulation*” in the original manuscript, we changed the title of this subsection to “*Versatile engineering of microspheres with ultrahigh mass fraction of therapeutics*” in the revised manuscript.

The benefits brought by our new method are limited if we emphasize on the encapsulation efficiency of the prepared microspheres. Although the in-droplet precipitation approach increases the drug encapsulation efficiency for about 3–26 times (Fig. 2e, Fig. S1c), the absolute value of encapsulation efficiency, in the range of 6.1–71.3 wt% (Fig. 2e), is still significantly less than 100%. This result suggests that the in-droplet precipitation approach can significantly mitigate but not completely eliminate the restriction of compatibility between drug and polymer molecules on drug loading.

In general, our new method can be used to prepare a wide range of drug loaded polymer microspheres with high drug loading and nano-in-micro structure, which has been demonstrated by using model drugs (ATV, MPS and HCT) and polymers (AcDX and PLGA) with different molecular structures and physicochemical properties. Nonetheless, our new method has a requirement on drug solubility, *i.e.*, the solubility of the drug should be low in an emulsified solvent (*e.g.*, ethyl acetate, dichloromethane, dimethyl carbonate, and chloroform), but can be significantly improved by a co-solvent (*e.g.*, dimethyl sulfoxide, dimethylformamide, acetonitrile, acetone, methanol, and ethanol) that is miscible with both the emulsified solvent and water. Therefore, our new method is not expected to be applicable to all drugs. However, due to a wide choice of emulsified solvents and co-solvents, a wide

variety of drugs are expected to meet the solubility requirement of our new method, and therefore have the potential to be prepared into microspheres with high drug loading and nano-in-micro structure.

Another limitation of our new method is related to the technique (droplet-microfluidics) that we used. Droplet-microfluidics can control the in-droplet precipitation process, and successfully prepare the polymer microspheres with high drug loading and nano-in-micro structure. However, the production rate of droplet-microfluidics (≤ 24 mg/h in this study), though not the focus of this study, is low, and remains a challenge for the clinical translation of prepared polymer microspheres. It is encouraging that the throughput of droplet-microfluidics can be increased by orders of magnitude using parallelized microfluidic devices^{1,2}, and for example, the production rate of polycaprolactone microspheres reached 277 g/h².

We have added the abovementioned contents in the “Discussion” section.

3. The contents and figures are all look great, as Nature likes to have. Nature usually does not understand the contents, and they have a tendency to publish as long as figures looks great. This manuscript has all ingredients.

Answer: We thank the reviewer for the positive comments on the contents and figures of our manuscript.

4. Release profiles.

According to Fig 5, drug release lasts for 5-10 days, and it appears that the microparticles cumulated at the cerebral spinal fluid. All good. But the authors need to comment on the possibility that it may be useful in humans. So many experiments have shown some efficacy in small animal studies, but almost all failed in clinical studies. The authors should provide caution in extrapolating this to clinical efficacy.

Answer: We thank the reviewer for the comments on the potential therapeutic efficacy of our microspheres in humans.

The therapeutic efficacy of polymer microspheres is mainly determined by the

pharmacokinetics of delivered drugs. Human beings and small animals are different in the volume and replenishment rate of cerebrospinal fluid, which are expected to affect the pharmacokinetics of MPS released from the microspheres. Therefore, the drug release kinetics of MPS@AcDX microspheres need to be further optimized in order to achieve desired pharmacokinetics of MPS and consequently satisfied clinical therapeutic efficacy in humans.

We have added the abovementioned contents in the “Discussion” section.

5. Once the above comments are reflected in revision, this manuscript is ready for publication in Nature. Actually, this manuscript deserves to be published in Science.

Answer: We thank again the reviewer for the insightful comments and suggestions, which help us to further strengthen the manuscript.

References:

- 1 Amstad, E. *et al.* Parallelization of microfluidic flow-focusing devices. *Phys. Rev. E* **95**, 043105 (2017).
- 2 Yadavali, S., Jeong, H.-H., Lee, D. & Issadore, D. Silicon and glass very large scale microfluidic droplet integration for terascale generation of polymer microparticles. *Nat. Commun.* **9**, 1222 (2018).

RESPONSE TO THE REVIEWER'S COMMENTS

Reviewer #2:

The authors present a microfluidic approach to forming polymer particles that contain drugs and assert that the drug is present in nanoparticle form within the particles.

The connection to biological materials, biomimetics and pomegranates is highly speculative and appears to be an attempt to exaggerate the research. The comparison to a pomegranate is just not needed and is a case of drawing similarities to nature purely for the sake of finding an analogy. If a research study is high quality, there is no need or necessity to draw some unfounded connection to the natural world. The reference to pomegranates should be removed.

Answer: We thank the reviewer for the time spent in reviewing our work. All the insightful comments and suggestions will undoubtedly help us to improve the overall quality of the manuscript. The response to the reviewer's comments is addressed below point-by-point. All the modifications/changes were also introduced in the revised version of the manuscript.

According to the reviewer's suggestion and in order to avoid distracting readers from the scientific contents of the manuscript, we removed the pomegranate related contents.

Many statements within the submission are purely speculative, untrue or unfounded. For example:

- "biological materials outperform most artificial ones" - there is no relevance here and a generalism like this is unsubstantiated. Clearly, there are no hard wearing alloys formed in nature, no biological equivalent to carbon-fibre, no biological thin plastic films (to use meaningless examples). Such a statement just should not be repeatedly used in a high quality research article.

- The introduction has a large section on biological inspiration that should be removed. I would expect that the analogy to fruit with seeds has come about after the work was conducted and was not an inspiration for the work as much of the work is described in

numerous other reports and technologies (see below)

Answer: We thank the reviewer for the suggestion. In order to focus on the scientific contents of the manuscript, we removed the pomegranate and biological inspiration related statements and rewrote the introduction. The changes have been highlighted in the "Introduction" section.

- The introduction refers to a report (REF 8) after the statement

"Among the developed particulate systems, biodegradable polymer microspheres have been exploited as a successful drug delivery platform and received regulatory approvals"

Referring to a review when claiming "regulatory approvals" is poor practice and original source material should be cited that show clear evidence of regulatory approval. The actual paper - Expert Opin. Drug Deliv. 9, 1161-1176 (2012) - states as its Expert Opinion:

"The broad clinical use of microspheres for delivery of therapeutic agents in particular biologics such as proteins has not been realized commercially. The limited availability of biodegradable polymers with a long history of regulatory approval and the challenges in gaining regulatory approval of a new polymer have hindered the development of microspheres for parenteral drug delivery."

Note - "The broad clinical use of microspheres...has not been realized commercially" and "The limited availability of biodegradable polymers with...regulatory approval have hindered the development of microspheres for parenteral drug delivery."

Answer: We thank the reviewer for this comment. To date, more than 15 (PLA/PLGA-based) polymer microspheres have been approved and marketed worldwide. Some of these polymer microspheres products received regulatory approvals from different agencies, such as U.S. Food and Drug Administration (FDA) and European Medicines Agency (EMA). In principle, we could find the original source material (such as approval letter) from each approval agency for each approved microsphere product. When we wrote "biodegradable polymer

microspheres...regulatory approvals", it applies to all those approved microsphere products in general rather than a specific microsphere product. It would not be practical to cite all the original source materials in this manuscript. The "Table 1" in Mao *et al.* review article¹ summarized the key microsphere products approved and marketed in the U.S., we cited this review article to support our general description that indeed biodegradable polymer microspheres have received regulatory approvals. If we are talking about a specific microsphere product, then definitely we should cite the exact original source material for that specific microsphere product.

We thank the reviewer for pointing out the "Expert Opinion" from Mao *et al.* review article¹: (1) "The broad clinical use of microspheres for delivery of therapeutic agents in particular biologics such as proteins has not been realized commercially.", and (2) "The limited availability of biodegradable polymers with a long history of regulatory approval and the challenges in gaining regulatory approval of a new polymer have hindered the development of microspheres for parenteral drug delivery."

We agree with the first expert opinion. Clinical use of microspheres for drug delivery has been realized commercially. However, the number of polymer microsphere products is still no more than 20, and the number of drugs that been delivered and types of diseases that been treated by polymer microsphere products are still limited. From these perspectives, indeed, the broad clinical use of microspheres for drug delivery has not been realized commercially. In order to avoid giving the readers an impression that microspheres have already been broadly used in clinics, we removed the statement "*biodegradable polymer microspheres have been exploited as a successful drug delivery platform*".

We also agree with the second expert opinion. Without a doubt, the limited availability of biodegradable polymers with regulatory approval is one of the main restriction factors that hinder the development of polymer microspheres for clinical use. For this reason, it reflects the necessity and importance of new methods that enable more types of drugs to be efficiently encapsulated by already existing polymers, such as PLGA, and formulated into microspheres for controlled drug release. As shown in this study, our in-droplet precipitation method tremendously enhanced the loading degree of model drugs (ATV, MPS and HCT) in both

PLGA and AcDX to a useful level, and the obtained MPS@AcDX microspheres with a loading degree of 63.1 wt% contribute to superior therapeutic efficacy and reduced side effects than conventional formulations in spinal cord injury therapy.

- "superfast in-droplet precipitation" - the term superfast has no basis for comparison or the superlative

Answer: We thank the reviewer for this comment. We used the word “superfast” three times in the manuscript (once in the Title, once in the Abstract and once in the Discussion) when we described the “in-droplet precipitation”. According to numerical simulation, the droplet size sharply reduced upon solvent diffusion (Fig. 3b), meanwhile, the MPS concentration in droplets rapidly increased and reached the saturated state in 38.3 ms (Fig. 3c), which indicates the in-droplet precipitation may start in 38.3 ms upon droplet formation. This time-scale is considered to be superfast, if compared with the solidification process of the droplet which is in seconds to tens of seconds (Fig. 3b–c). The significance of superfast in-droplet precipitation is that it rapidly reduces the drug saturation solubility in droplets and consequently contributes to highly efficient drug encapsulation in microspheres, as described more in detail in the section “*Numerical simulation of the droplet solidification and drug encapsulation*”.

We changed “*superfast*” to “*controlled*” in the revised manuscript. We believe “controlled” is a more comprehensive and precise word to describe the whole in-droplet precipitation process of our method. A “superfast” but uncontrolled in-droplet precipitation is likely to fail in the preparation of high drug loaded microspheres, as demonstrated by the bulk emulsion approach (Fig. S10). In addition, “superfast” is also part of the controlled process. “controlled in-droplet precipitation” is the key to our method for the preparation of high drug loaded microspheres.

- I have no idea of what a "high-performance human architecture" actually is

Answer: We removed this statement from the revised Introduction section.

- "ultrahigh drug loading" - This terminology is unclear and has no comparative basis. For example, solid dispersions for solid-drug nanoparticle formation have been reported with drug loading as high as 80wt % in the solid state (drug relative to [drug+polymer+surfactant] excipients)

Answer: In our study, the drug loading degree is defined as the mass fraction of drug in the microspheres. The drug loading degree in polymer microspheres has been reported in many studies and discussed/summarized in some review articles^{1,2}. In most studies, the loading degree is just a few percents or less than one percent, and the very high ones usually do not exceed 30 wt%. Therefore, drug loading higher than 30 wt% is considered as ultrahigh drug loading. We have added a description in the subsection – “*Proof of in-droplet precipitation strategy to engineer microspheres with high mass fraction of cargo*” of the Results part.

We should also note that the difficulty in achieving a certain drug loading degree in polymer microspheres is influenced by the chosen drug and polymer. If the loading degree of a drug in polymer microspheres is low but has been considerably increased by a new method/technique, the achieved loading degree, though may not as high as 30 wt%, may also be considered to be ultrahigh.

In our study, the level of drug loading degree is discussed and compared in the context of polymer microspheres used for controlled drug release. Indeed, some other drug loaded particles may also have ultrahigh drug loading, but they are not used for controlled drug release and they are not able to control the drug release, for example, solid dispersion and surfactant stabilized amorphous/crystallized drug nanoparticles (nearly 100 wt% drug loading degree) which are used for enhancing the solubility of poorly water-soluble drugs.

- Stability The authors state - "It is worthy to point out that although with high drug loading, all these microspheres are quite stable over the long-term storage (Fig. 5c–h), as confirmed by Xray powder diffractograms. This long-term stability is likely to benefit from the pomegranate like structure of the formed microspheres."

To be clear, 15 days stability is not "long term". Any medicine that is of value would need to be studied under accelerated conditions for many months (one standard would be 6 months at 40 deg C and 75% RH). 15 days is clearly not enough to state that these are stable and there is no attempt to quantify crystallinity changes over this very short period.

Answer: We thank the reviewer for raising the issue regarding long term stability. The stability testing of the prepared drug (ATV, MPS and HCT) loaded polymer (AcDX and PLGA) microspheres was until 12 months (as shown in Fig. 4c–e and f–h) rather than 15 days. We agree with the reviewer that 40 °C/75% RH/6 months are used as conditions for accelerated stability studies. More specifically, 40 °C/75% RH/6 months apply to the general case of accelerated stability studies for both active pharmaceutical ingredients and finished pharmaceutical products, as indicated in Annex 2, WHO Technical Report Series, No. 953, 2009 (https://www.who.int/medicines/publications/pharmprep/pdf_trs953.pdf?ua=1). From the point of view of stability test, our drug loaded microspheres should be considered as finished pharmaceutical products rather than active pharmaceutical ingredients. In addition, marketed drug loaded microspheres are usually packaged in impermeable containers (such as closed syringe, please see https://www.accessdata.fda.gov/drugsatfda_docs/label/2014/020517s036_019732s041lbl.pdf), and usually stored at 25 °C and 2–8 °C if the microspheres contain small molecule drug (please see https://www.accessdata.fda.gov/drugsatfda_docs/label/2014/020517s036_019732s041lbl.pdf) and macromolecule drug (please see https://www.accessdata.fda.gov/drugsatfda_docs/label/2004/21075s008lbl.pdf), respectively. Therefore, to study the stability of our drug loaded microspheres, we used the conditions (25 °C/40% RH/12 months) of long-term stability study for finished pharmaceutical products packaged in impermeable containers, as given in the above-mentioned WHO Technical Report. For the stability test, we only tested the samples by X-ray powder diffractogram after different time points, and these data are not enough for precisely quantifying the crystallinity

changes.

According to the X-ray powder diffractogram (Fig. 4c–e and f–h), ATV@AcDX, HCT@AcDX and HCT@PLGA microspheres did not show any crystallization peak of drugs at all time points, indicating no major changes in crystallinity of drug inside microspheres. The X-ray powder diffractogram of MPS@AcDX, ATV@PLGA and MPS@PLGA showed crystallization peaks of drugs at all time points, and these peaks showed slight intensity fluctuation/increasement over 12 months. Such slight intensity fluctuation/increasement indicates that there is no obvious change in the crystallinity of encapsulated drugs.

The approaches and concepts reported are relatively well known

- Firstly, using the slow (or indeed rapid) diffusion of a solvent with water miscibility to create particles is very well known and I refer the authors to Angew Chem Int Ed Engl. 2001 3;40(23):4330-4361. doi: 10.1002/1521-3773 Organic Nanoparticles in the Aqueous Phase-Theory, Experiment, and Use - Horn and Rieger.

Answer: We thank the reviewer for providing this comprehensive review article³, in which various preparation methods of organic nanoparticles in aqueous phase have been summarized and discussed. Although this review article does not cover the microparticles, we believe some of the mentioned preparation methods for nanoparticles also apply to microparticles.

The reviewer's statement "*using the slow (or indeed rapid) diffusion of a solvent with water miscibility to create particles*" describes a basic principle of particle formation, and this basic principle is involved in many specific methods/techniques for preparing particles. We believe our controlled in-droplet precipitation is an innovative method that applies the basic principle of "*using diffusion of a solvent with water miscibility to create particles*".

In general, the basic principle of "*using diffusion of a solvent with water miscibility to create particles*" is involved in the emulsification diffusion method and solvent-displacement method, as named in Horn and Rieger's review article³. Broadly speaking, emulsification diffusion method contains both emulsification solvent evaporation method and emulsification

solvent extraction method, as in both methods the solvents need to diffuse from the emulsions to the surrounding aqueous phase. Solvent-displacement method is also known as nanoprecipitation method. Emulsification method and nanoprecipitation method have been commonly used for preparing microparticles and nanoparticles. However, as far as we know, neither of these two methods nor their combination has been used to prepare polymer microspheres with high drug loading and/or nano-in-micro structure. Our approach, which combines the emulsification method and nanoprecipitation method, can prepare a wide range of polymer microspheres with high drug loading and nano-in-micro structure. More importantly, the prepared microspheres can control the drug release which makes them useful for therapeutic applications.

Our approach not only enables the precipitation of drug molecules to form drug nanoparticles inside the emulsions (droplets), but also reduces the loss (diffusion) of drug molecules from the emulsions (droplets). Subsequent solidification of emulsions (droplets) enables the precipitated drug nanoparticles to be encapsulated inside the matrix of the polymer microspheres. We believe such a delicate combination of emulsification method and nanoprecipitation method creates a novel method that can prepare polymer microspheres with favorable features for therapeutic applications.

- Nanoprecipitation is also well known in the literature, however, the authors do not discuss this

Answer: In the response to the last comment, we discussed the nanoprecipitation. The formation of drug nanoparticles in the droplets indeed could be classified as nanoprecipitation. Therefore, we added relevant statement/discussion in the “Discussion” section.

As mentioned in the response to the last comment, nanoprecipitation method alone has not been used to and it in fact cannot prepare polymer microspheres with high drug loading and/or nano-in-micro structure. In our method, the nanoprecipitation method, together with the emulsification method, can contribute to the formation of polymer microspheres with high drug loading and nano-in-micro structure.

For the typical nanoprecipitation method (solvent-displacement method), nanoprecipitation

takes place when solvents containing polymer/drug are added to the miscible antisolvent. However, in our method, the nanoprecipitation takes place in a controlled manner inside the emulsions (droplets). Such a controlled in-droplet (nano)precipitation is believed to be novel.

- Microfluid approaches to microspheres is also well know

Answer: We agree that microfluidics, which are very general approaches, have already been used to prepare microspheres before. However, we believe it does not prevent the further innovation of microfluidic approaches for preparing microspheres.

For previous studies, the drug loading degree of prepared polymer microspheres (not including hydrogel/liposome microparticles) is usually limited and no more than 20 wt% as summarized in Table 2 (also shown as below) of our publication². Especially, the loading degree of atorvastatin (ATV) in polymer microspheres is only ~2 wt% in Li et al's study⁴ and ~15 wt% in Liu et al's study⁵. In this study, by using our new microfluidic approach, we increased the loading degree of ATV in polymer microspheres to 50.5 wt%/51.6 wt% (Fig. 2d).

Table 2 Droplet based microfluidics for drug delivery applications

Microfluidic devices	Emulsion types	Particle precursors	Drug loading
Polydimethylsiloxane	O/W	PLGA	Bupivacaine, ca. 20%
Polydimethylsiloxane	O/W	AcDX	Sorafenib (SFN) and celecoxib, ca. 0.5–4.2%
Polydimethylsiloxane	O/W	HPMCAS	Celecoxib, SFN, ca. 2.6%
Glass capillaries	O/W	Sodium alginate	Iopamidol, ca. 17%
Glass capillaries	W/O/W	Thermal sensitive lipid	Doxorubicin (DOX) and (paclitaxel) PTX
Glass capillaries	W/O/W	Polyethylene glycol (PEG)/poly- ϵ -caprolactone	Bovine serum albumin
Polydimethylsiloxane	O/W/O	Polyacrylamide/PNIPA	Rhodamine B isothiocyanate–dextran, —
Glass capillaries	O/W	PSi in HPMCAS mixtures	Atorvastatin and celecoxib, ca. 15% for each
Glass capillaries	O/W	PSi in HPMCAS	Fluorouracil and celecoxib, ca. 7%
Glass capillaries	O/W	PSi in HPMCAS	Glucagon-like peptide-1, 15% in PSi
Glass capillaries	O/W	PSi in HPMCAS	Dipeptidyl peptidase 4, —
Glass capillaries	O/W	Halloysite nanotubes in HPMCAS	Atorvastatin, 2%; celecoxib, 6%
Glass capillaries	W/O/W	PSi in phospholipid vesicle	Piroxicam, 19%
Glass capillaries	W/O/W	PSi, DNA, gold nanorods in lipid	DOX, 30%; erlotinib, 10%; 17-AAG, 15%
Glass capillaries	Janus particles	Poly(acrylamide)/poly(methyl acrylate)	Ketoprofen and sodium fluorescein
Polydimethylsiloxane	W/O	Alginate	No drugs loaded
Polydimethylsiloxane	O/W or O/O	PLGA	No drugs loaded
Polydimethylsiloxane	Water/air; ethanol/air	Amorphous drug nanoparticles	Fenofibrate, clotrimazole, danazol or estradiol
Polymethylmethacrylate	Nitrogen gas/ethanol/water	Ethyl cellulose	Amoxicillin, ca. 13%
Glass capillaries	Fiber breakup to droplets	—	No drugs loaded

The innovation of our microfluidic approach is that it delicately combines the emulsification method and nanoprecipitation method and enables the controlled in-droplet precipitation of drug molecules followed by efficient encapsulation of the precipitated drug nanoparticles in the matrix of polymer microspheres upon solidification of the droplets. This microfluidic approach has not been reported before for preparing polymer microspheres with high drug loading and nano-in-micro structure.

- The authors appear to have formed what is widely described as a "solid dispersion" - again,

a well known approach that has been achieved by spray drying and freeze drying of dispersions, emulsions and solutions for many years. The difference here is that this has been achieved in a microfluidic manner (to be clear, I have never seen anyone feel the need to describe solid dispersions using any fruit analogy...)

Answer: The term “solid dispersion” is narrowly defined as a dispersion of drug in a solid matrix where the drug is in a molecularly dispersed state, and broadly defined as a dispersion of drug in a solid matrix where the drug dispersed state includes many forms such as drug molecules and amorphous/crystallized drug domains⁶. Amorphous/crystallized drug domains could be considered as drug aggregates/particles. Obviously, according to the broad definition of solid dispersion, any drug loaded polymer microsphere is a solid dispersion, no matter how much drug molecules are encapsulated in and how the drug molecules are dispersed in the matrix of the polymer microspheres.

We believe the novelty of our polymer microspheres is not affected by how they are called and should be determined by: **(1)** the problems solved by their high drug loading degree and nano-in-micro structure; **(2)** the method for preparing them; and **(3)** the versatility/complexity of the used method.

(1) The high drug loaded polymer microspheres can increase the dose in a single administration, reduce the frequency of administration and improve patient compliance, and cut the use of materials/excipients and associated side effects. For some diseases, a high drug loading is a prerequisite for an effective therapy^{1,7}. For example, the maximum dose volume is limited by the injection location, such as intrathecally administration, resulting in a limited amount of injectable drug carriers such as polymer microspheres. Therefore, in order to deliver a required dose for effective treatment, a high drug loading must be achieved in the microspheres. Taking our MPS@AcDX microspheres for example, benefiting from their high drug loading degree (63.1 wt%), they delivered a high dose of methylprednisolone (400 µg) within the limited administration volume (10 µL) by one single intrathecal injection. The amount of acetalated dextran used was 1/433 of that of conventional microspheres. It should be pointed out that high drug loading degree could adversely affect the release profiles of polymer microspheres. It is a challenge to simultaneously achieve a relatively high drug

loading and acceptable controlled release profiles⁷. In this study, the prepared ultrahigh MPS-loaded microspheres successfully enabled the gradual release of MPS, and contributes to superior therapeutic efficacy and reduced side effects than conventional formulations in spinal cord injury therapy.

(2) The novelty of our method, compared to previous microfluidic approaches, emulsification diffusion methods and nanoprecipitation methods, has been discussed in the response to several previous comments.

(3) The versatility of a method is also very important. Our method has been demonstrated to be applicable for a rather wide range of model drugs and polymers with different molecular structures and physicochemical properties, although it is not expected to be suitable for all drugs and polymers. Also, our method has a very simple preparation process, as the same as conventional drug loaded microspheres prepared by common single emulsion droplet microfluidics. The versatility and simple preparation process of our method will contribute to its popularization.

Indeed, solid dispersions have been achieved by spray drying and freeze drying of dispersions, emulsions and solutions. Usually, these techniques are used to prepare solid dispersion, in which drugs are molecularly dispersed in the polymer matrix, for the solubility enhancement of poorly water-soluble drugs. These solid dispersions clearly have different structures as compared with our nano-in-micro microspheres.

In principle, such nano-in-micro structured drug-loaded polymer microspheres can be prepared by encapsulating pre-formed drug nanoparticles in polymer matrix *via* emulsion method or spray drying method. For drug nanoparticles, their engineering approaches can be generally classified into top-down and bottom-up. Bottom-up methods yield drug nanoparticles through nucleation⁸ or self-assembly⁹. Top-down approaches, such as milling¹⁰ and high pressure homogenization¹¹, involve the size reduction of bulk materials to nanoscale, requiring high energy or pressure input. In both top-down and bottom-up approaches, the preparation of drug nanoparticles needs a large amount of work to optimize various parameters, such as the types of solvents and stabilizers¹². For this pre-prepared drug nanoparticle strategy, one major challenge is to efficiently disperse drug nanoparticles inside

emulsified solvents¹³.

As a result of the controlled in-droplet precipitation process, the freshly formed drug nanoparticles are immediately embedded within the solidified polymer matrix (**Fig. 3b–c**). Our approach avoids formulation optimization to stabilize drug nanoparticles in emulsified solvents, without any stabilizers. This makes the in-droplet precipitation approach universally applicable to the production of a variety of drug nanoparticles incorporated nano-in-micro spheres.

We have added most of the abovementioned contents in the “Discussion” section.

- The structures described and shown figuratively in, for example, Figure 1, do not bare any real relationship to pomegranates. They are very similar to numerous reports of solid dispersions however. If the authors wish to see images that are more "pomegranate-like" they need to review the literature - for example Nanoscale, 2016,8, 7224-7231 - doi.org/10.1039/C6NR00788K

Answer: Based on the reviewer’s comments, we removed the pomegranates related contents from Figure 1 in order to avoid distracting readers from the scientific contents of the manuscript. A new version of Figure 1 has been updated in the revised manuscript.

- The reliance on "like dissolves like", Flory-Huggins and Hansen's parameters is very old thinking.

The authors state

"Based on the general principle of ‘like dissolves like’, Flory-Huggins interaction parameter

(χ_{dc}) is commonly used to estimate drug-carrier material compatibility¹⁸. The Flory-Huggins interaction parameter can be calculated from Hansen's solubility parameter (Table S1) by Eq. 1"

This approach has been quite widely shown to not be representative of the effects of drug and polymer interactions and have been considered to be poorly predictive. For example, I would like to direct the authors to "Like Dissolves Like? A Comprehensive Evaluation of Partial Solubility Parameters to Predict Polymer-Drug Compatibility in Ultrahigh Drug-Loaded Polymer Micelles" *Biomacromolecules* 2019, 20, 8, 3041-3056 doi.org/10.1021/acs.biomac.9b00618. The abstract for this paper concludes

"In general, the Flory-Huggins interaction parameters were less suited to correctly estimate the experimental drug solubilization compared to the Hansen solubility parameters. The latter were able to correctly predict some trend regarding good and poor solubilizers, yet the overall predictive strength of Hansen solubility parameters is clearly unsatisfactory." - note "clearly unsatisfactory". This reference also cites several polyoxazoline polymer particles with drug loadings >50 wt%

Answer: We thank the reviewer for providing Lübtow *et al.* recent research article¹⁴. In our study, the drug-polymer compatibility, which was estimated by the general principle of 'like dissolves like' and Flory-Huggins interaction parameter and Hansen's solubility parameter¹⁵⁻²⁰, very well predicted the poor encapsulation of drug (ATV, MPS or HCT) in polymer (AcDX or PLGA). Nevertheless, we carefully checked Lübtow *et al.*¹⁴ and Turpin *et al.*²¹ articles. A consensus among all the studies (including our study) is that drug-carrier material compatibility, which refers to miscibility, strongly affects drug loading. However, Lübtow *et al.* and Turpin *et al.* studies show Flory-Huggins interaction parameter and Hansen's solubility parameter cannot sufficiently precisely predict the compatibility, and new methods, that consider all parts of the free energy of mixing and use more detailed partial solubility parameters, need to be developed in order to usefully predict drug-polymer compatibility. In summary, so far, no method can precisely predict compatibility between all drugs and

polymers. Each method has its limitations and is only applicable to certain drugs and polymers.

The original intention of using and calculating Flory-Huggins interaction parameter is to quantify the drug-polymer compatibility, which was qualitatively indicated by the experimental results (Fig. S1c) in our study. Since our experimental results, poor encapsulation of drug (ATV, MPS or HCT) in polymer (AcDX or PLGA), already indicated the poor compatibility between drug (ATV, MPS or HCT) and polymer (AcDX or PLGA), it would not be essential to quantify the drug-polymer compatibility by Flory-Huggins interaction parameter.

Taking into consideration that currently there is no perfect/uncontroversial method for quantifying drug-polymer compatibility, we removed the contents related to the general principle of 'like dissolves like' and Flory-Huggins interaction parameter and Hansen's solubility parameter, and quantitatively described the combability.

Additionally:

- "%" should always state what basis the % is taken (eg w/w %)

Answer: We thank the reviewer for this suggestion. We have made the necessary changes to the manuscript.

- it is not possible to have "432-time less". 432-times is a multiplier. This should be 1/432nd or a fraction of some form.

Answer: We thank the reviewer for this suggestion. We have changed "432-time less than" into "1/433 of".

- The paper seems to switch randomly between reporting crystallinity and describing the process as precipitation. It makes it very unclear what exactly is being reported

Answer: We thank the reviewer for this comment. The diffusion of solvents from the droplets to the surrounding aqueous phase causes oversaturation and precipitation of drug molecules in the droplets. We used precipitation for the description of this process. We would like to point

out that the precipitated drug molecules could be in either crystallized state or amorphous state.

We carefully checked our manuscript and confirmed that we did not report the crystallinity (refers to the degree of structural order of a solid). We indeed used the words “crystals” and “crystallized” in our manuscript, however, we only used these words when the X-ray powder diffractograms and/or differential scanning calorimetry curves confirmed the existence of crystallized drugs.

- Figure 3 appears to have no identifiable drug particles within the microparticles

Answer: The size of the SEM images in “Fig. 3” (*i.e.*, Fig. 2 in the revised manuscript) is rather small, which makes it difficult to identify the drug particles within the microspheres. If we zoom in on these SEM images (see below), it will help to identify the drug particles. We would like to point out that the TPE particles are more visible within the microspheres (Fig. 1e). The reason is that TPE particles are more crystallized in the microspheres (Fig. 1f–g), and therefore exhibits obvious edge angles of crystals. ATV, MPS and HCT are amorphous or weakly crystallized in the microspheres (Fig. 2f–g), and therefore the precipitated drug nanoparticles do not exhibit obvious edge angles of crystals, which increases the difficulty in distinguishing the drug nanoparticles from the polymer matrix. If compared with the SEM images of bare microspheres, the existence of drug nanoparticles in microspheres makes the appearance of their cross-sections different (Fig. 2b–c).

MPS@AcDX

HCT@AcDX

ATV@PLGA

MPS@PLGA

HCT@PLGA

- The description of a "polymer pulp" in the following statement is just meaningless and

unnecessary

- "Therefore, the release of payloads only needs to cross the polymer “pulp membrane” and almost no burst release has been observed for all the obtained microspheres."

Answer: We changed “polymer pulp” into “polymer matrix” in the revised manuscript.

I would strongly encourage the authors to rewrite their manuscript without hype and hyperbole and concentrate on the relatively interesting "controlled in-droplet solidification" - their words - which is a clever combination of microfluidics, solid dispersions and nanoprecipitation. If they had not over hyped and created an needless connection to biomimetics and pomegranates, this would have been an interesting report but as it stands, it is impossible to see through the hyperbole, superlative and distracting comparisons.

I recommend rejection from Nature Communications in its current form but urge the editors to provide the authors with an opportunity to re-write and submit as a quality research article that references the literature and places this work properly in context and without exaggeration.

Answer: We thank the reviewer again for the insightful comments and suggestions, and we have made many changes based on the reviewer’s comments and suggestions. We have removed the biomimetics and pomegranates related contents from the manuscript in order to avoid distracting readers from the scientific contents of the manuscript, and now we are focus on the “controlled in-droplet precipitation” in the revised manuscript. Also, we have discussed and compared with other methods/techniques in the revised manuscript. We thank the reviewer for providing us an opportunity to revise the manuscript.

References:

- 1 Mao, S. R., Guo, C. Q., Shi, Y. & Li, L. C. Recent advances in polymeric microspheres for parenteral drug delivery - part 1. *Expert Opin. Drug Deliv.* **9**, 1161-1176 (2012).
- 2 Liu, D., Zhang, H., Fontana, F., Hirvonen, J. T. & Santos, H. A. Microfluidic-assisted fabrication of carriers for controlled drug delivery. *Lab Chip* **17**, 1856-1883 (2017).
- 3 Horn, D. & Rieger, J. Organic nanoparticles in the aqueous phase—theory, experiment, and use. *Angew. Chem. Int. Ed.* **40**, 4330-4361 (2001).
- 4 Li, W. *et al.* Microfluidic assembly of a nano-in-micro dual drug delivery platform composed of halloysite nanotubes and a pH-responsive polymer for colon cancer therapy. *Acta. Biomater.* **48**, 238-246 (2017).
- 5 Liu, D. *et al.* Microfluidic assembly of monodisperse multistage pH-responsive polymer/porous silicon composites for precisely controlled multi-drug delivery. *Small* **10**, 2029-2038 (2014).
- 6 Huang, Y. & Dai, W.-G. Fundamental aspects of solid dispersion technology for poorly soluble drugs. *Acta Pharm. Sin. B* **4**, 18-25 (2014).
- 7 Chen, W., Zhou, S., Ge, L., Wu, W. & Jiang, X. Translatable high drug loading drug delivery systems based on biocompatible polymer nanocarriers. *Biomacromolecules* **19**, 1732-1745 (2018).
- 8 Liu, D. *et al.* A versatile and robust microfluidic platform toward high throughput synthesis of homogeneous nanoparticles with tunable properties. *Adv. Mater.* **27**, 2298-2304 (2015).
- 9 Kalsin, A. M. *et al.* Electrostatic self-assembly of binary nanoparticle crystals with a diamond-like lattice. *Science* **312**, 420 (2006).
- 10 Van Eerdenbrugh, B., Van den Mooter, G. & Augustijns, P. Top-down production of drug nanocrystals: Nanosuspension stabilization, miniaturization and transformation into solid products. *Int. J. Pharm.* **364**, 64-75 (2008).
- 11 Keck, C. M. & Müller, R. H. Drug nanocrystals of poorly soluble drugs produced by high pressure homogenisation. *Eur. J. Pharm. Biopharm.* **62**, 3-16 (2006).

- 12 Maudens, P., Seemayer, C. A., Pfefferlé, F., Jordan, O. & Allémann, E. Nanocrystals of a potent p38 MAPK inhibitor embedded in microparticles: Therapeutic effects in inflammatory and mechanistic murine models of osteoarthritis. *J. Control. Release* **276**, 102-112 (2018).
- 13 Van Der Hoeven, P. C. & Lyklema, J. Electrostatic stabilization in non-aqueous media. *Adv. Colloid Interface Sci.* **42**, 205-277 (1992).
- 14 Lübtow, M. M., Haider, M. S., Kirsch, M., Klisch, S. & Luxenhofer, R. Like dissolves like? A comprehensive evaluation of partial solubility parameters to predict polymer–drug compatibility in ultrahigh drug-loaded polymer micelles. *Biomacromolecules* **20**, 3041-3056 (2019).
- 15 Liu, J. B., Xiao, Y. H. & Allen, C. Polymer-drug compatibility: A guide to the development of delivery systems for the anticancer agent, Ellipticine. *J. Pharm. Sci.* **93**, 132-143 (2004).
- 16 Letchford, K., Liggins, R. & Burt, H. Solubilization of hydrophobic drugs by methoxy poly(ethylene glycol)-block-polycaprolactone diblock copolymer micelles: Theoretical and experimental data and correlations. *J. Pharm. Sci.* **97**, 1179-1190 (2008).
- 17 Zhang, Y. *et al.* Strategies for improving the payload of small molecular drugs in polymeric micelles. *J. Control. Release* **261**, 352-366 (2017).
- 18 Mahmud, A. *et al.* Self-associating poly(ethylene oxide)-b-poly(alpha-cholesteryl carboxylate-epsilon-caprolactone) block copolymer for the solubilization of STAT-3 inhibitor cucurbitacin I. *Biomacromolecules* **10**, 471-478 (2009).
- 19 Hansen, C. M. *Hansen solubility parameters: a user's handbook*. (CRC press, 2002).
- 20 Shi, C. J. *et al.* Exploring the effect of hydrophilic and hydrophobic structure of grafted polymeric micelles on drug loading. *Int. J. Pharm.* **512**, 282-291 (2016).
- 21 Turpin, E. R. *et al.* In silico screening for solid dispersions: the trouble with solubility parameters and χ_{FH} . *Mol. Pharm.* **15**, 4654-4667 (2018).

REVIEWER COMMENTS

Reviewer #2 (Remarks to the Author):

The authors appear to have generally covered most of the points that I raised during the initial review.

I would encourage them to not choose words like "ultrahigh" as it really has no scale or reference (I note that they have attempted one by saying >30wt% but when does ultra become super and then hyper? "High loadings" would be sufficient I would think)

Also - one note: all pharmaceutical finished products are subject to accelerated stability testing, not just APIs. The authors need to understand the clinical manufacturing and regulatory approval processes and the data requested for compiling into IMP dossiers. I agree with them that ambient conditions (eg 25 deg C for 12 months) would be an acceptable set of conditions BUT 40/75 for several months is also used for finished products. You would only resort to ambient conditions if your finished product cannot show stability at 40 deg C.

Finally, I would be grateful if the authors review the definitions of solid dispersions and solid solutions. These are not the same thing and do not fall into the solid dispersion concept

This is a nice piece of work that just didn't need to be hyped. I would strongly encourage the authors to be 100% careful with hype, hyperbole and definitions.

Reviewer #3 (Remarks to the Author):

The study describes a method to formulate microparticles with high drug content using a co-solvent in the polymer-drug phase that upon diffusion forms drug precipitate within the polymer matrix. The key for high drug loading is attributed to precipitation of the drug before solidification of the polymer phase. The method has been validated with two different polymers and three different drugs. The microparticles loaded with methylprednisolone were evaluated via intrathecal injection in rat spinal cord injury model. The results also show high and sustained drug levels in cerebrospinal

fluid with microparticles as compared to that with drug alone, improvement in functional recovery and reduced edema and cavitation as compared to controls. Below are few major issues:

- Solvent-diffusion method to achieve high drug loaded microparticles is not new. For examples, Richey et al. (US patent 9,293,211 B2, July 9, 2016) have used similar co-solvent system to precipitate drug into the polymer phase to achieve 50% drug load and 97% encapsulation efficiency which is closed to what is described in this paper. In Richey et al. study, buprenorphine was used in PLGA polymer phase with co-solvent benzyl alcohol (also described the use of DMSO, acetonitrile, etc.). Co-solvents selected are such that they increase the solubility of the drug in the polymer phase, but when it diffuses out, drug precipitates out in the polymer phase. In this study, the same basic principle was used — the only difference appears is the use of microfluidics whereas the patent describes homogenization and dilution to facilitate the diffusion of the co-solvent. Indeed, the method described in the patent produced microparticles similar to those described in this paper in terms of particle size, drug load, release profiles and encapsulation efficiency. Therefore, the authors' claim that somehow this is a new method or resolved a major issue of drug encapsulation is not quite convincing.
- The claim that this method forms nanoparticles inside the microspheres does not seem correct— these are just drug precipitates/crystals that are in the nanometer is size range.
- In terms of experimental design in this study, for example the effect of co-solvent amount on encapsulation is similar to what is described in the above patent.
- In spinal cord injury model, the treatment was given within 5 min after inducing the injury. Although the data look quite good, giving treatment right after the injury is clinically irrelevant. The victims of spinal cord injury are not going to get treatment right after the injury. The treatment during the acute phase considered at least 3 hrs after the injury. The biology of injury response changes significantly with time, it is the secondary injury, and hence what is effective right after the injury may not work when given several hrs after the injury. In fact, this is the limitations of methylprednisolone. It was not effective when given several hrs after the injury.
- Drug levels in cerebrospinal fluid were analyzed over 7 days after the treatment. There are total 9 data points and for each time point analysis, 20 μl of cerebrospinal fluid (CSF) was used. Assuming that minimum 30 μl of CSF withdrawn for each time point—so for 9 time point, it would 270 μl . Total volume of CSF in 3-month old rat is 156 μl . Is it possible to withdraw such a large volume of CSF at such a high frequency within a one week? Can such data be reliable? More importantly, is it ethical?
- There are no details of animal age/weight. Only male animals were used but one should use equal number of M/F since the response in male differs than in female.
- The BBB score looks quite “clean” which is rare in SCI model, as it is very well known that some animals without any treatment spontaneously recover whereas there are few which do not respond to the treatment.
- Few other important details are missing—e.g., conditions used for the release study? What buffer? Whether the study was done under sink conditions? These details are necessary as it could affect drug release profile.

- As pointed out by Reviewer #1, there are several unnecessary/sensational sentences/words. e.g., “superior” therapeutic efficacy

RESPONSE TO THE REVIEWER'S COMMENTS

Reviewer #2:

The authors appear to have generally covered most of the points that I raised during the initial review.

Answer: We thank the reviewer for the time spent in reviewing our work, and we also thank the reviewer for acknowledging the efforts we made in the last round of revision.

The response to the reviewer's comments is addressed below point-by-point. All the modifications/changes were also introduced in the revised version of the manuscript and highlighted in yellow.

I would encourage them to not choose words like "ultrahigh" as it really has no scale or reference (I note that they have attempted one by saying >30wt% but when does ultra become super and then hyper? "High loadings" would be sufficient I would think)

Answer: We thank the reviewer for the suggestion. We changed "ultrahigh" to "high" in the revised manuscript.

Also - one note: all pharmaceutical finished products are subject to accelerated stability testing, not just APIs. The authors need to understand the clinical manufacturing and regulatory approval processes and the data requested for compiling into IMP dossiers. I agree with them that ambient conditions (eg 25 deg C for 12 months) would be an acceptable set of conditions BUT 40/75 for several months is also used for finished products. You would only resort to ambient conditions if your finished product cannot show stability at 40 deg C.

Answer: We thank the reviewer for this comment. We agree that both finished pharmaceutical products and active pharmaceutical ingredients should receive accelerated stability testing (such as 40 °C/75% RH/6 months) whenever possible. Also, we agree that ambient conditions (25 °C/40% RH/12 months) are used for stability testing if the finished pharmaceutical products are not stable at 40 °C/75%. This is the reason why we choose ambient conditions (25 °C/40% RH/12 months) for the stability testing of our polymer microspheres, especially

the PLGA microspheres. As shown in the differential scanning calorimetry curve (see below), 40 °C is higher than the onset temperature of glass transition for PLGA (50:50). The PLGA microspheres are not stable at this temperature, therefore we did not conduct the stability at the accelerated conditions.

Finally, I would be grateful if the authors review the definitions of solid dispersions and solid solutions. These are not the same thing and do not fall into the solid dispersion concept

Answer: We thank the reviewer for this comment. We agree solid dispersion and solid solution are different. Briefly, when the drug is molecularly dispersed in a solid matrix, it is a solid solution, and when the drug is dispersed as crystalline or amorphous particles in a solid matrix, it is solid dispersion.

This is a nice piece of work that just didn't need to be hyped. I would strongly encourage the authors to be 100% careful with hype, hyperbole and definitions.

Answer: We thank the reviewer for the suggestion, and we made necessary changes in the revised manuscript.

RESPONSE TO THE REVIEWER'S COMMENTS

Reviewer #3:

The study describes a method to formulate microparticles with high drug content using a co-solvent in the polymer-drug phase that upon diffusion forms drug precipitate within the polymer matrix. The key for high drug loading is attributed to precipitation of the drug before solidification of the polymer phase. The method has been validated with two different polymers and three different drugs. The microparticles loaded with methylprednisolone were evaluated via intrathecal injection in rat spinal cord injury model. The results also show high and sustained drug levels in cerebrospinal fluid with microparticles as compared to that with drug alone, improvement in functional recovery and reduced edema and cavitation as compared to controls. Below are few major issues:

Answer: We thank the reviewer for the time spent in reviewing our work.

The response to the reviewer's comments is addressed below point-by-point. All the modifications/changes were also introduced in the revised version of the manuscript and highlighted in yellow.

· Solvent-diffusion method to achieve high drug loaded microparticles is not new. For examples, Richey et al. (US patent 9,293,211 B2, July 9, 2016) have used similar co-solvent system to precipitate drug into the polymer phase to achieve 50% drug load and 97% encapsulation efficiency which is closed to what is described in this paper. In Richey et al. study, buprenorphine was used in PLGA polymer phase with co-solvent benzyl alcohol (also described the use of DMSO, acetonitrile, etc.). Co-solvents selected are such that they increase the solubility of the drug in the polymer phase, but when it diffuses out, drug precipitates out in the polymer phase. In this study, the same basic principle was used — the only difference appears is the use of microfluidics whereas the patent describes homogenization and dilution to facilitate the diffusion of the co-solvent. Indeed, the method described in the patent produced microparticles similar to those described in this paper in terms of particle size, drug load, release profiles and encapsulation efficiency. Therefore, the authors' claim that somehow this is a new method or resolved a major issue of drug encapsulation is not quite convincing.

Answer: We thank the reviewer for providing Richey *et al.* patent, which we did not notice before. Now, we have carefully checked this patent.

If we follow the experimental methods described in Richey *et al.* patent and use homogenization to prepare our high drug (ATV, MPS or HCT) loaded polymer (AcDX or PLGA) microspheres, we will fail. As shown in Fig. S10 and Fig. S11, the bulk emulsion approach (by magnetic stirring) failed to prepare high drug loaded microspheres, as illustrated by the formation of many micro-sized free drug crystals outside microspheres. Homogenization has an even higher energy input than magnetic stirring. The strong and inhomogeneous turbulent mixing is expected to cause uncontrolled solvent diffusion and drug precipitation, and breakage of emulsion droplets containing drug precipitates/crystals. In contrast, our microfluidics-based controlled in-droplet precipitation method can successfully prepare high drug (ATV, MPS or HCT) loaded polymer (AcDX or PLGA) microspheres. The novelty of our method is that it is a delicate combination of microfluidics, emulsification and nanoprecipitation, as acknowledged by Reviewer #2. Our method enables controlled precipitation of drug molecules and sequential solidification of drug and polymer molecules in the droplets.

According to Richey *et al.* patent, “*the use of a single solvent or co-solvent had little effect on the drug load of the prepared microspheres*” (Table 2 of Richey *et al.* patent). In our study, we proved that the co-solvent has a significant effect on the drug loading of the prepared microspheres. The drug loading degree of microspheres prepared with co-solvent (Fig. 2d) is about 40–450 times higher than those prepared without co-solvent (Fig. S1b). In our study, co-solvent is a must/prerequisite for achieving high drug loading in the microspheres. We proposed the “in-droplet precipitation” and proved it by experiments and numeral simulation. Richey *et al.* patent does not describe anything about solvent diffusion and drug precipitation. Based on these facts, we would say the concept “co-solvent system to precipitate drug into the polymer phase” does not exist in Richey *et al.* patent.

· The claim that this method forms nanoparticles inside the microspheres does not seem correct—these are just drug precipitates/crystals that are in the nanometer is size range.

Answer: We thank the reviewer for this comment. Nanoparticles are particles that exist on a nanometre scale, which is a quite broad definition. In 2008, the International Organization for Standardization defined a nanoparticle as a discrete nano-object where all three Cartesian

dimensions are less than 100 nm. Three years later, the Commission of the European Union endorsed a wider-ranging definition: *a natural, incidental or manufactured material containing particles, in an unbound state or as an aggregate or as an agglomerate and where, for 50% or more of the particles in the number size distribution, one or more external dimensions is in the size range 1 nm–100 nm*. Here, we would like to point out that in our manuscript the nanoparticles are not defined by their shape/morphology. If the drug precipitates/crystals are in the nanometer size, then it would be reasonable to consider them as nanoparticles.

· In terms of experimental design in this study, for example the effect of co-solvent amount on encapsulation is similar to what is described in the above patent.

Answer: We thank the reviewer for this comment. Regarding our method for preparing high drug loaded microspheres, co-solvent is believed to have a significant effect on the drug encapsulation, it stands to the reason that we designed an experiment to investigate the effect of co-solvent amount on drug encapsulation. Specifically, in this work, we studied the effect of several different co-solvent amounts on drug encapsulation (Fig. 3). The conclusion is that a higher ratio of co-solvent leads to lower drug loading and encapsulation efficiency. This effect seems to be similar to what is described in Table 2 of Richey *et al.* patent, but we noticed that Richey *et al.* patent compared the effect of co-solvent with a single solvent, but it did not compare the effect of several different co-solvent amounts. In addition, Richey *et al.* patent states, “*The use of a single solvent or co-solvent had little effect on the drug load of the prepared microspheres*”. We would say this is a quite careful statement, as there were multiple variables for the microsphere preparation (Table 2 of Richey *et al.* patent). The lower drug loading and encapsulation efficiency of the microspheres prepared by the co-solvent method might also partly caused by the used higher mixing speed. Higher mixing speed is expected to generate smaller emulsion droplets and enhance the diffusion of drug molecules from emulsions droplets to the outer aqueous phase, resulting in lower drug loading and encapsulation efficiency. Therefore, from Richey *et al.* patent, it is not clear how the co-solvent amount would affect the encapsulation.

· In spinal cord injury model, the treatment was given within 5 min after inducing the injury. Although the data look quite good, giving treatment right after the injury is clinically irrelevant. The victims of spinal cord injury are not going to get treatment right after the injury. The treatment during the acute phase considered at least 3 hrs after the injury. The biology of injury response changes significantly with time, it is the secondary injury, and hence what is effective right after the injury may not work when given several hrs after the injury. In fact, this is the limitations of methylprednisolone. It was not effective when given several hrs after the injury.

Answer: We thank the reviewer for this comment. MPS gradually released from microspheres, therefore, MPS concentration in cerebrospinal fluid is quite low after the administration of HMPS@AcDX. HMPS@AcDX group only reached the peak concentration (around 56.7 µg/mL) at day-3 after injury (**Fig. 5b**). In contrast, MPS concentration was about 554.1 µg/mL at 0.5 h after the administration of MPS. The *in vivo* data clearly demonstrated that the locally sustained release of MPS from high drug loading HMPS@AcDX conforms to the therapeutic need for SCI therapy. Benefiting from the high drug loading degree and controlled release of MPS, HMPS@AcDX inhibited gliosis, attenuated the activation of microglia, suppressed CSBP production, and protected axon and myelin sheath in the injured spinal cord. Ultimately, HMPS@AcDX microspheres effectively improved motor function after SCI. By comparing the therapeutic outcomes of HMPS@AcDX with all the other groups, and by considering the sustained release of MPS from microspheres, the administration time seems to have little impact on the final therapeutic outcomes.

In addition, this study mainly discusses the therapeutic effects of different MPS formulations on spinal cord injury. From the safety and operability of animal experiments, especially for intrathecal administration, the risk of postoperative infection and death in rats was increased when the animals are anesthetized for a longer time and the incision is exposed for a longer time. Therefore, in previous studies on MPS for spinal cord injury therapy in rats, MPS was usually immediately administrated after the injury^{1, 2, 3}. Immediate postoperative administration has also been applied in the treatment of spinal cord injury in rats for other drugs, such as an antagonist of ATP-sensitive receptor P2X₇⁴, ferulic acid (FA)-glycol chitosan nanoparticles⁵, hydrogel containing lentiviral vectors encoding Lingo-1 short harpin interfering RNA⁶, and PLGA nanoparticles coated with GDNF⁷.

In clinical practice, the damage immediately after dosing is questionable. In 2013, the

latest evidence-based medical guidelines for acute cervical and spinal cord injuries were published by the Congress of Neurological Surgeons (CNS) and the American Association of Neurological Surgeons (AANS) for the first time. There is no Class I or Class II medical evidence supporting intravenous administration of MPS within 3 h and 3–8 h⁸. The main reason is the presence of the blood-brain barrier, which blocks the transport of MPS from the circulation system to the central nervous system, so that MPS cannot reach the effective treatment concentration in the cerebrospinal fluid⁹.

· Drug levels in cerebrospinal fluid were analyzed over 7 days after the treatment. There are total 9 data points and for each time point analysis, 20 μl of cerebrospinal fluid (CSF) was used. Assuming that minimum 30 μl of CSF withdrawn for each time point—so for 9 time point, it would 270 μl . Total volume of CSF in 3-month old rat is 156 μl . Is it possible to withdraw such a large volume of CSF at such a high frequency within a one week? Can such data be reliable? More importantly, is it ethical?

Answer: We thank the reviewer for this comment. A misunderstanding may be caused by our previous description in the methods part, which missed some details. The production rate of cerebrospinal fluid in rats is estimated to be 2.2 $\mu\text{l}/\text{min}$ and the total volume of cerebrospinal fluid in rats is estimated to be 400–550 μl ^{10, 11}. Usually, less than 190 μl of cerebrospinal fluid can be collected from rats. In this study, different rats were used to extract cerebrospinal fluid at different time points in each group, so cerebrospinal fluid was extracted from each rat only once. To minimize the time consuming and expensive animal studies, as well as the application of the animal 3Rs (Replacement, Reduction, and Refinement) rule, some rats after cerebrospinal fluid withdraw were used for the evaluation of pathological changes towards the injured spinal cord.

The following sentences were added to “*Methods-Pharmacokinetics of MPS in cerebrospinal fluid*” of the revised manuscript:

“The cerebrospinal fluid was collected by a 1 mL syringe equipped with a 25G disposable needle. For every sampling time point, the volume of withdrawn cerebrospinal fluid ranged between 100 and 150 μL for each rat. Each rat was only sampled once.”

· There are no details of animal age/weight. Only male animals were used but one should use equal number of M/F since the response in male differs than in female.

Answer: We thank the reviewer for this comment. According to our literature review, gender does not have a clear influence on the prognosis of spinal cord injury. In 2019, a retrospective cohort study on patients with traumatic spinal cord injury in Western Denmark from 2010 to 2017 showed that gender has no significant effect on the prognosis of neurological function after early decompression in patients¹². Similarly, in 2017, Park *et al.*¹³ retrospectively analyzed the prognostic factors affecting the outcomes of decompression surgery in patients with spinal cord injury. They classified all patients based on their interval to decompression, gender, age, surgical level, presence of high signal intensity, American Spinal Injury Association Impairment scale (AIS) before surgery, blood pressure at admission, the amount of cord compression, surgical time, estimated blood loss during surgery, and steroid use. They found that gender had no significant influence on the prognosis of neurological function. Many other clinical studies^{14, 15, 16, 17, 18, 19, 20, 21, 22} also confirm that gender did not affect the neurological function recovery after spinal cord injury. In the rat spinal cord injury model, Fee *et al.*²³ found that gender has no significant influence on the prognosis of rat neurological function. Therefore, when studying spinal cord injury in rats, many researchers did not use the same number of male and female rats, and they only selected male or female rats^{1, 2, 3, 4, 5, 6, 7, 24}.

· The BBB score looks quite “clean” which is rare in SCI model, as it is very well known that some animals without any treatment spontaneously recover whereas there are few which do not respond to the treatment.

Answer: Our group at Nanjing Medical University has been engaged in the research of spinal cord injury for more than 20 years. During this period, we have a set of very consistent experimental operating standard procedures for experimental verification and exploration, which ensures the consistency of the spinal cord injury model. Clinically, for patients with complete spinal cord injury, there is almost no functional recovery in the early stage after injury. Therefore, we continued to observe the motor function recovery of the rats after anesthesia recovery, 1 d, and 2 d after spinal cord injury, and the BBB score of the rats could not be enrolled until 0 points, which ensured the completeness and consistency of spinal cord injury. Before the spinal cord injury model was established, the vertebral plate was fully bitten to expose the spinal cord tissue as much as possible, so as to reduce the dispersion of

the strike force caused by the surrounding soft tissue coverage during the strike. Adequate positioning ensured that the strike position was in the midline of the spinal cord during each modeling. During the strike, the spine was fixed to avoid the downward depression of the spine and ensured the consistency of the stroke height.

After spinal cord surgery, the recovery of locomotor capacity was measured by the Basso, Beattie and Bresnahan (BBB) open field locomotor test procedure²⁵. Rats were placed in an open field and were allowed to explore freely for 4 min. Two examiners, who were blinded to the treatment regimen, participated in all open field tests and were positioned across from each other to observe both sides of the rats^{25, 26}.

· Few other important details are missing—e.g., conditions used for the release study? What buffer? Whether the study was done under sink conditions? These details are necessary as it could affect drug release profile.

Answer: We agree that the conditions used for the release study could affect the drug release profile. In our study, PBS (pH 7.4) was used for drug release study, and the release studies were performed under the sink condition for all the drug molecules. This information was previously already included in the “*Methods-Loading degree, encapsulation efficiency and drug release studies*”.

· As pointed out by Reviewer #1, there are several unnecessary/sensational sentences/words. e.g., “superior” therapeutic efficacy

Answer: We made necessary changes in the revised manuscript. For example, we removed “superior”, or changed “superior” to “improved”.

References:

1. Kim YT, Caldwell JM, Bellamkonda RV. Nanoparticle-mediated local delivery of Methylprednisolone after spinal cord injury. *Biomaterials* **30**, 2582-2590 (2009).
2. Chvatal SA, Kim YT, Bratt-Leal AM, Lee H, Bellamkonda RV. Spatial distribution and acute anti-inflammatory effects of methylprednisolone after sustained local delivery to the contused spinal cord. *Biomaterials* **29**, 1967-1975 (2008).
3. Cerqueira SR, *et al.* Microglia response and in vivo therapeutic potential of methylprednisolone-loaded dendrimer nanoparticles in spinal cord injury. *Small* **12**, 972 (2016).
4. Peng W, *et al.* Systemic administration of an antagonist of the ATP-sensitive receptor P2X7 improves recovery after spinal cord injury. *Proc. Natl. Acad. Sci. U.S.A.* **106**, 12489-12493 (2009).

5. Wu W, *et al.* Neuroprotective ferulic acid (FA)-glycol chitosan (GC) nanoparticles for functional restoration of traumatically injured spinal cord. *Biomaterials* **35**, 2355-2364 (2014).
6. Wu HF, *et al.* The promotion of functional recovery and nerve regeneration after spinal cord injury by lentiviral vectors encoding Lingo-1 shRNA delivered by Pluronic F-127. *Biomaterials* **34**, 1686-1700 (2013).
7. Wang YC, *et al.* Sustained intraspinal delivery of neurotrophic factor encapsulated in biodegradable nanoparticles following contusive spinal cord injury. *Biomaterials* **29**, 4546-4553 (2008).
8. Hadley MN, Walters BC. Introduction to the guidelines for the management of acute cervical spine and spinal cord injuries. *Neurosurgery* **72 Suppl 2**, 5-16 (2013).
9. Pardridge WM. CSF, blood-brain barrier, and brain drug delivery. *Expert Opin. Drug Deliv.* **13**, 963-975 (2016).
10. Chiu C, *et al.* Temporal course of cerebrospinal fluid dynamics and amyloid accumulation in the aging rat brain from three to thirty months. *Fluids Barriers CNS* **9**, 3 (2012).
11. Lai YL, Smith PM, Lamm WJ, Hildebrandt J. Sampling and analysis of cerebrospinal fluid for chronic studies in awake rats. *J. Appl. Physiol. Respir. Environ. Exerc. Physiol.* **54**, 1754-1757 (1983).
12. Haldrup M, Schwartz OS, Kasch H, Rasmussen MM. Early decompressive surgery in patients with traumatic spinal cord injury improves neurological outcome. *Acta Neurochir.* **161**, 2223-2228 (2019).
13. Park JH, Kim JH, Roh SW, Rhim SC, Jeon SR. Prognostic factor analysis after surgical decompression and stabilization for cervical spinal-cord injury. *Br. J. Neurosurg.* **31**, 194-198 (2017).
14. Equebal A, Anwer S, Kumar R. The prevalence and impact of age and gender on rehabilitation outcomes in spinal cord injury in India: a retrospective pilot study. *Spinal Cord* **51**, 409-412 (2013).
15. New PW, Epi MC. Influence of age and gender on rehabilitation outcomes in nontraumatic spinal cord injury. *J. Spinal Cord Med.* **30**, 225-237 (2007).
16. Lynch AC, Wong C, Anthony A, Dobbs BR, Frizelle FA. Bowel dysfunction following spinal cord injury: a description of bowel function in a spinal cord-injured population and comparison with age and gender matched controls. *Spinal Cord* **38**, 717-723 (2000).
17. Kirshblum S, Johnston MV, Brown J, O'Connor KC, Jarosz P. Predictors of dysphagia after spinal cord injury. *Arch. Phys. Med. Rehabil.* **80**, 1101-1105 (1999).
18. Jang HJ, Park J, Shin HI. Length of hospital stay in patients with spinal cord injury. *Ann. Rehabil. Med.* **35**, 798-806 (2011).
19. Franke AC, Snoek GJ, de Groot S, Nene AV, Spooren AI, Post MW. Arm hand skilled performance in persons with a cervical spinal cord injury--long-term follow-up. *Spinal Cord* **51**, 161-164 (2013).
20. Harrop JS, *et al.* Neurologic improvement after thoracic, thoracolumbar, and lumbar spinal cord (conus medullaris) injuries. *Spine* **36**, 21-25 (2011).
21. Scivoletto G, Morganti B, Molinari M. Sex-related differences of rehabilitation outcomes of spinal cord lesion patients. *Clin. Rehabil.* **18**, 709-713 (2004).
22. Greenwald BD, Seel RT, Cifu DX, Shah AN. Gender-related differences in acute rehabilitation lengths of stay, charges, and functional outcomes for a matched sample

- with spinal cord injury: a multicenter investigation. *Arch. Phys. Med. Rehabil.* **82**, 1181-1187 (2001).
23. Fee DB, Swartz KR, Joy KM, Roberts KN, Scheff NN, Scheff SW. Effects of progesterone on experimental spinal cord injury. *Brain Res.* **1137**, 146-152 (2007).
 24. Hellal F, *et al.* Microtubule stabilization reduces scarring and causes axon regeneration after spinal cord injury. *Science* **331**, 928-931 (2011).
 25. Basso DM, Beattie MS, Bresnahan JC. A sensitive and reliable locomotor rating-scale for open-field testing in rats. *J. Neurotrauma* **12**, 1-21 (1995).
 26. Liu D, *et al.* An in situ gelling drug delivery system for improved recovery after spinal cord injury. *Adv. Healthc. Mater.* **5**, 1513-1521 (2016).

REVIEWER COMMENTS

Reviewer #2 (Remarks to the Author):

I have no further comments on this manuscript but it does appear that Reviewer 3 has some pertinent issues that need to be addressed in full

Reviewer #3 (Remarks to the Author):

Overall, authors' response to the previous critique is not satisfactory. Further careful review of the manuscript raised additional questions.

- In general, the method described for high load encapsulation protocol is not new. Others have used similar co-solvent diffusion method—the mode of diffusion could be different than using microfluidic method described in this paper. As mentioned, Richey et al. (US patent 9,293,211 B2, July 9, 2016) have used similar solvent, co-solvent system to precipitate drug into the polymer phase to achieve 50% drug load and 97% encapsulation efficiency. In addition, co-authors have published extensively microfluidic method to fabricate microparticles. It seems that the original method is picked from the laboratory of David A. Weitz (Monodisperse Double Emulsions Generated from a Microcapillary Device, *Science* 22 Apr 2005:Vol. 308, Issue 5721, pp. 537-541). There are other methods (supercritical fluid, microfluidizers) that can produce microparticles with high drug load.
- Authors did provide data in Fig S10 that bulk method leads to precipitation of drug outside but did not describe in detail how these were prepared. It appears from the wording “bulk method” the polymer phase was dumped into the aqueous phase which is going to cause separation of drug crystals. Did author try to control the process of emulsification and solvent removal (DMSO) as Richey et al. have done?
- Authors indicated in response, according to Richey et al. patent, “the use of a single solvent or co-solvent had little on the drug load of the prepared microspheres effect on the drug load of the prepared microspheres”. The point is that they have fabricated microparticles with high drug load with and without co-solvent.
- On many occasions, to justify that the method developed is unique to achieve high drug load, comparison was not like apple to apple. For example, the reference cited to support that the other method lead to low drug loaded is for peptide and not for hydrophobic drugs, like methylprednisolone (Ref 22 is for peptide). Similarly, several references are cited that are for micelles which show low drug loading, but it is a different kind of system and the main purpose of

micelles more as solubilizing agent. There are several such misleading examples throughout the manuscript.

- Clinical significance of injecting microparticles within 5 min after the injury is minimum. For immediate injection after the injury, microparticles can directly be injected at the lesion site. The efficacy of such an approach has been demonstrated by other group using methylprednisolone-loaded PLGA microparticles (Biomaterials 30, 2582-2590,2009). It is not clear why intrathecal injection of microparticles is needed if the formulation is to be used immediately after the injury such as during spinal surgery. In real life, no patient can show-up within 5 min after the injury. The justification that there is high mortality several hrs after the injury is not convincing as there are several published studies where the treatment was given several hrs after the injury.
- Authors previously have published a paper (Adv. Mater. 2018, 30, 1706032) where they have shown AcDX microparticles (without drug) when delivered intrathecally were shown to improve functional outcome and promote healing of the spinal cord, which has been attributed to scavenging of glutamate and calcium ion. However, in the study when AcDX microparticles were used as a control, similar effect was seen. For example, the data in Fig 5i do not show healing with control AcDX microparticles but only when these microparticles are loaded with the drug. Perhaps, authors need to explain the discrepancy between the two studies.
- The question was raised about the frequency of withdrawal of CSF for drug assays. Based on the previous method, it was not possible to withdraw 9 times CSF. Now the narrative has been changed that different animals were used. Still, the description in figure legend states n=7 which does not make sense as the last two data point should not have any animal left as total number of data points are 9 and CSF was withdrawn only once from the animal. This is not just a simple error. It would be difficult to reproduce data without complete details.
- During the previous review, detail conditions for the release study were requested. That information is not provided (how much microparticles, how much buffer, etc.). It is just stated that sink condition was maintained. Without the detail information, one may not be able to reproduce the data. The information is critical considering methylprednisolone is not soluble in aqueous buffer.
- This reviewer still remains surprised with tight error bars for the BBB score. Weight drop method is known to cause variation in injury and also animal response varies to the same level injury. It has been well documented (e.g., Nature 2015 Feb 19;518(7539): 404-8).

Comments on abstract:

The following statement in abstract, "Systems with high content of cargo can minimize excipients administration, reduce side effects, improve therapeutic efficacy and promote patient compliance. However, engineering such systems is extremely challenging, as their loading capacity is inherently limited by the compatibility between cargo molecules and carrier materials", requires revision/editing. As it is written carries no meaning. For example, what Systems mean? Drug Delivery Systems or Microparticle-based drug delivery systems? The limitation of drug cargo content that the

authors are implying are relevant to microparticle-based system because matrix type drug delivery systems can be prepared with high drug content. Also, it is clear how authors can claim that systems with high content can reduce side effect, improve therapeutic efficacy and promote patient compliance because the desired properties of drug delivery systems depend on the disease conditions. For example, dexamethasone eluting pacing electrode requires very low drug content in the matrix but for a longer duration to reduce scar tissue formation at the contact point. Drug eluting stents is another example where drug load is kept low so that the released is sustained. The point is that one cannot generalize a statement that high drug content in drug delivery systems can achieve all of the stated objectives. Also, authors need to specific to refer “cargo” as “drug cargo”.

The following statement in abstract, “To mitigate the cargo-carrier compatibility limitation towards therapeutics encapsulation, we developed a sequential solidification strategy” also requires editing to be scientifically correct. Incompatibility of cargo-carrier system could be of different types such as drug interacting ionically or chemically to carrier system. The authors are specifically referring to “miscibility” of drug cargo-polymer and not general incompatibility.

The amount of acetylated dextran used was 1/433 of that of conventional microspheres. Which are these conventional microspheres. There is no reference found in the entire manuscript that describes “conventional microspheres” to compare with.

The statement that “the controlled release of methylprednisolone--- and reduced side effects than conventional formulations in spinal cord injury therapy. Which formulation is considered as conventional formulations? There is no toxicity study carried out to claim that the formulation developed in this study result in reduced side effects.

RESPONSE TO THE REVIEWER'S COMMENTS

Reviewer #3:

Overall, authors' response to the previous critique is not satisfactory. Further careful review of the manuscript raised additional questions.

Answer: We thank the reviewer for the time spent in reviewing our work and for the suggestions to improve our paper.

The response to the reviewer's comments is addressed below point-by-point. All the modifications/changes were also introduced in the revised version of the manuscript.

• In general, the method described for high load encapsulation protocol is not new. Others have used similar co-solvent diffusion method—the mode of diffusion could be different than using microfluidic method described in this paper. As mentioned, Richey et al. (US patent 9,293,211 B2, July 9, 2016) have used similar solvent, co-solvent system to precipitate drug into the polymer phase to achieve 50% drug load and 97% encapsulation efficiency.

Answer: Thank you for the comment. We believe we answered this question in the previous round of review but would highlight that Richey *et al.* did not anticipate to utilize the “*co-solvent system to precipitate drug into the polymer phase*”. The reason for Richey *et al.* to use the co-solvent system is that “*the incorporation of a co-solvent reduces the amount of continuous phase needed by about 6-fold and allows for larger batch sizes to be produced while keeping the volumes of the continuous and dispersed phases lower*”. **In other words, the aim of Richey et al. using the co-solvent system is to improve the manufacturing efficiency/production rate of drug-loaded microspheres.** In Richey *et al.* patent, they stated that “*the use of a single solvent or co-solvent had little effect on the drug load of the prepared microspheres*” (Table 2 in Richey et al. patent). In their system, co-solvent is not a prerequisite for achieving high drug loading for microspheres. Moreover, there is no evidence about drug precipitation in Richey *et al.* patent.

In our study, we are aiming to engineer microspheres with high drug loading. We proved that the precipitation of drug molecules prior to the solidification of polymers in droplets can

enable the production of microspheres with high content of drug nanoparticles. In our system, the co-solvent has a significant effect on the drug loading of the prepared microspheres. The drug loading degree of microspheres prepared with co-solvent (Fig. 2d) is about 40–450 times higher than those prepared without co-solvent (Fig. S1b). For our study, co-solvent is a prerequisite for engineering high drug-loaded microspheres.

Our controlled in-droplet precipitation strategy is a delicate combination of droplet microfluidics, emulsification, controlled diffusion of solvents, and soft matter nanoprecipitation, as acknowledged by Reviewer #2 from last round of comments and replies. Our strategy enables controlled in-droplet precipitation of drug molecules and sequential solidification of drug and polymer molecules. Benefiting from the excellent capability in manipulating nanoliter volume of liquid, we controlled the solvent removal process from droplets by a droplet-based microfluidic device. This strategy is succeeded in preparing high drug (ATV, MPS or HCT) loaded polymer (AcDX or PLGA) microspheres. Regarding the bulk magnetic stirring approach, many micro-sized free drug crystals formed outside microspheres and failed to prepare high drug loaded microspheres (Fig. S10 and S11). Homogenization has an even higher energy input than magnetic stirring. The strong and inhomogeneous turbulent mixing is expected to cause uncontrolled solvent diffusion and drug precipitation, and breakage of emulsion droplets containing drug precipitates/crystals. Like the magnetic stirring approach, the homogenization approach used in Richey *et al.* patent is not suitable to prepare high drug (ATV, MPS or HCT) loaded polymer (AcDX or PLGA) microspheres.

In this study, we proposed the “in-droplet precipitation” strategy and proved it by experiments (a variety combinations of drugs and polymers) and numerical simulation. Richey *et al.* patent does not describe solvent diffusion and drug precipitation. Without using the controlled in-droplet precipitation, the co-solvent system used in Richey *et al.* patent is not suitable to prepare high drug loaded microspheres in our study. **Overall, the motivation, detailed intermediate process, formation mechanism and outcome of using co-solvent are clearly different between Richey *et al.* patent and our study.**

In addition, co-authors have published extensively microfluidic method to fabricate microparticles.

Answer: Thank you for the comment. The reviewer is correct that we have published before using the microfluidic method, but we would like to mention that the “microfluidic method” and the “microparticles” are quite broad terms. We believe our previous work does not prevent us from further innovation and advance on microfluidic technique involved microparticles. Neither the concept of “in-droplet precipitation” nor “high drug loaded microspheres” have been reported in our previous publications.

It seems that the original method is picked from the laboratory of David A. Weitz (Monodisperse Double Emulsions Generated from a Microcapillary Device, Science 22 Apr 2005:Vol. 308, Issue 5721, pp. 537-541).

Answer: Thank you for the comment. Prof. Weitz’s paper published in Science reported a glass-capillary device for generating monodisperse double emulsions, which is an important contribution to double emulsions. We used single emulsions rather than double emulsions in our work, and therefore our device is different from the one reported in Prof. Weitz’s Science paper. None of “drug loading”, “high drug loading” “co-solvent” and “drug precipitation” is reported in Prof. Weitz’s Science paper. To the best of our knowledge, the microfluidic “in-droplet precipitation” approach for preparing high drug loaded microspheres is only highlighted in our work.

There are other methods (supercritical fluid, microfluidizers) that can produce microparticles with high drug load.

Answer: Thank you for the comment. We agree with the reviewer that there are other methods that can produce microparticles with high drug loading. We believe that our work contributes to the known methods and adds a new approach to an area with a wide range of demands and requirements.

• Authors did provide data in Fig S10 that bulk method leads to precipitation of drug outside but did not describe in detail how these were prepared. It appears from the wording “bulk method” the polymer phase was dumped into the aqueous phase which is going to cause separation of drug crystals. Did author try to control the process of emulsification and solvent removal (DMSO) as Richey et al. have done?

Answer: Thank you for the comment. Accordingly, we have added details about bulk method into the Methods part. “*For the bulk method, the oil and water phases were the same as the inner and outer fluids used for microfluidics, respectively. The oil phase (0.8 mL) was slowly added below the liquid surface of the water phase (10 mL) under magnetic stirring.*”. We did not dump the oil phase into the water phase as such approach is more likely to break the emulsions.

Thank you for the suggestion to “control the process of emulsification and solvent removal (DMSO) as Richey *et al.* have done”. Richey *et al.* used homogenizer to prepare their high buprenorphine loaded microspheres. To the best of our understanding, Richey *et al.* did not report how they optimized/controlled the emulsification process and solvent removal (DMSO).

As mentioned in our response to Reviewer #3 in the last round of revision, if we follow Richey *et al.* homogenization method to prepare our high drug (ATV, MPS or HCT) loaded polymer (AcDX or PLGA) microspheres, we will not obtain high-drug loaded microspheres. Homogenization has an even higher energy input than magnetic stirring. The strong and inhomogeneous turbulent mixing is expected to cause uncontrolled solvent diffusion and drug precipitation, and breakage of emulsion droplets containing drug precipitates/crystals.

• Authors indicated in response, according to Richey et al. patent, “the use of a single solvent or co-solvent had little on the drug load of the prepared microspheres effect on the drug load of the prepared microspheres”. The point is that they have fabricated microparticles with high drug load with and without co-solvent.

Answer: Thank you for the comment. We wrote “*the use of a single solvent or co-solvent had little effect on the drug load of the prepared microspheres*” in the response letter in the last round of revision, in order to answer to one of the comments in which the reviewer emphasized

the importance of co-solvent for achieving high drug loading – “*co-solvent system to precipitate drug into the polymer phase to achieve 50% drug load*”.

Regarding the new comment “*The point is that they have fabricated microparticles with high drug load with and without co-solvent.*”, actually it well supports our response to one of the previous comments: “*According to Richey et al. patent, co-solvent is not a prerequisite for achieving high drug loading for their specific drug loaded microspheres. Richey et al. mentioned in their patent that the reason for still using the co-solvent is that “the incorporation of a co-solvent reduces the amount of continuous phase needed by about 6-fold and allows for larger batch sized to be produced while keeping the volumes of the continuous and dispersed phases lower.” In other words, Richey et al. used the co-solvent for the purpose of improving the manufacturing efficiency/production rate of their specific high buprenorphine loaded microspheres. Therefore, the idea for using co-solvent is clearly different between Richey et al. patent and our study.*”.

We agree Richey *et al.* prepared high buprenorphine loaded microspheres with and without co-solvent. Regarding co-solvent, the idea/concept for using co-solvent is clearly different between Richey *et al.* patent and our study, which is already explained in detail above. Regarding the high-drug loaded microparticles, in addition to Richey *et al.* method, we also agree there are other methods that can produce microparticles with high drug loading. As stated in our response to previous comments: “*We believe that our work contributes to the known methods and adds a new approach to an area with a wide range of demands and requirements.*”.

- On many occasions, to justify that the method developed is unique to achieve high drug load, comparison was not like apple to apple. For example, the reference cited to support that the other method lead to low drug loaded is for peptide and not for hydrophobic drugs, like methylprednisolone (Ref 22 is for peptide). Similarly, several references are cited that are for micelles which show low drug loading, but it is a different kind of system and the main purpose of micelles more as solubilizing agent. There are several such misleading examples throughout the manuscript.

Answer: Thank you for the comment. We apologize if our previous responses or text was not

clear and led the reviewer to misunderstand the cited references.

Regarding “*the reference cited to support that the other method lead to low drug loaded is for peptide and not for hydrophobic drugs, like methylprednisolone (Ref 22 is for peptide)*”, we would like to show the context where we cited the reference #22: “*This so-called “initial burst” phenomenon poses a serious toxicity threat and is a major hurdle for the development of microsphere products²².*”. We were discussing the general initial burst phenomenon of microspheres when we cited reference #22 rather than discussing anything about other methods for achieving high drug loading. In fact, we did not compare the drug loading of our microspheres with that in reference #22.

We indeed cited references about polymeric micelles, but we did not compare the drug loading of our polymer microspheres with the drug loading of those polymer micelles. From the materials point of view, the carrier materials in polymer micelles are polymer molecules. We cited references about polymeric micelles when we discussed the relationship between drug loading capacity and the compatibility between drug and polymer molecules. Those references about polymeric micelles investigated the compatibility between drug and polymer molecules. The results from those references showed that loading capacity is inherently limited by the compatibility between drug and polymer molecules.

We would also like to mention that we did not just cite references about low drug loaded polymeric micelles. We also cited references about high-drug loaded polymeric micelles: “*Several approaches have been developed to enhance the drug-polymer compatibility, such as forming polymer-drug conjugates, and utilizing electrostatic attraction and donor–receptor coordination between drug and carrier molecules⁹⁻¹¹.*”, when we discussed the general relationship between drug loading capacity and drug-polymer compatibility.

• Clinical significance of injecting microparticles within 5 min after the injection is minimum. For immediate injection after the injury, microparticles can directly be injected at the lesion site. The efficacy of such an approach has been demonstrated by other groups using methylprednisolone-loaded PLGA microparticles (Biomaterials 30, 2582-2590, 2009). It is not clear why intrathecal injection of microparticles is needed if the formulation is to be used immediately after the injury such as during spinal surgery. In real life, no patient can show up within 5 min after the injury. The justification that there is high mortality several hours after the injury is not convincing as there are several published studies where the treatment was given several hours after the injury.

Answer: Thank you for the comment. We acknowledge that injection within 5 minutes after injury is not representative of real-world conditions. However, as a common method used in a range of tests, we believe the immediate injection after injury is suitable for a proof-of-concept study aimed at the development of a new strategy for preparing high-drug loaded microspheres.

To demonstrate the superior therapeutic efficacy contributed by high drug loading, in principle any diseases that require the administration of high-dose of drug within limited injection volume could be selected as the proof-of-concept study. MPS is potentially beneficial in traumatic SCI therapy¹. However, high dose intrathecal administration of MPS (1 mg/kg bolus dose plus 1 mg/kg/h) is required². In contrast to a high dose, the injection volume is limited (about 10 μ L per rat)^{3,4}. Therefore, AcDX microspheres with a high mass fraction (63.1 wt%) of MPS were selected as the proof in this study to demonstrate the superior therapeutic efficacy of developed high drug loaded microspheres. We believe that the administration time would not sacrifice the importance of our strategy in preparing the high drug-loaded microspheres.

Chondroitinase ABC can efficiently promote functional recovery of locomotor and proprioceptive behaviors. In both Bradbury's study (Nature 2002, 416: 636-640)⁵ and Alilain *et al.* study (Nature 2011, 475: 196-200)⁶, Chondroitinase ABC was injected immediately after the injury. A recent study in Nature (Nature 2020, 587: 613-618) showed that an injury to the spinal cord in neonatal mice leads to scar-free healing⁷. In this study, the therapeutic agent, microglia, was transplanted just one hour after spinal cord injury. Immediate postoperative

administration has also been applied in the treatment of spinal cord injury in rats for other drugs, such as P2X7 (an antagonist of ATP-sensitive receptor)⁸, squalenoyl adenosine⁹, ferulic acid (FA)-glycol chitosan nanoparticles¹⁰, hydrogel containing lentiviral vectors encoding Lingo-1 short harpin interfering RNA¹¹, and PLGA nanoparticles coated with GDNF¹². In general, the immediate injection after injury is quite common in the experimental study for spinal cord injury therapy.

In addition, MPS gradually released from HMPS@AcDX for 7 days. Therefore, the MPS concentration in cerebrospinal fluid is quite low after administration. HMPS@AcDX sustained the release of MPS for 7 days, and the MPS peak concentration (around 56.7 $\mu\text{g}/\text{mL}$) was achieved at day-3 after injury. For the MPS solution group, drug concentration in cerebrospinal fluid was about 554.1 $\mu\text{g}/\text{mL}$ at 0.5 h after administration. Benefiting from the high drug loading degree and controlled release of MPS, HMPS@AcDX inhibited gliosis, attenuated the activation of microglia, suppressed CSBP production, and protected axon and myelin sheath in the injured spinal cord. Ultimately, the HMPS@AcDX efficiently improved motor function after SCI. By comparing the therapeutic outcomes of HMPS@AcDX with all the other groups and by considering the sustained release of MPS from HMPS@AcDX, we would say the administration time seems to have only little impact on the final therapeutic outcome for HMPS@AcDX.

Once again, we would like to mention again that we acknowledge that injection within 5 minutes after injury is not representative of real-world conditions but as a common method used in a range of tests we believe this is suitable for a proof-of-concept study aimed at the development of a new strategy for preparing high-drug loaded microspheres.

• Authors previously have published a paper (Adv. Mater. 2018, 30, 1706032) where they have shown AcDX microparticles (without drug) when delivered intrathecally were shown to improve functional outcome and promote healing of the spinal cord, which has been attributed to scavenging of glutamate and calcium ion. However, in the study when AcDX microparticles were used as a control, similar effect was seen. For example, the data in Fig 5i do not show healing with control AcDX microparticles but only when these microparticles are loaded with the drug. Perhaps, authors need to explain the discrepancy between the two studies.

Answer: Thank you for the comment. In our previous paper (Adv. Mater. 2018, 30, 1706032), we did not perform the magnetic resonance imaging study for the rats. Only in this current manuscript, we used a 7.0T magnetic resonance imaging unit to evaluate the pathological change of the spinal cord at day-1 and day-28 post injury. Spinal cord edema increased significantly during the early time period after injury, showing hyperintensity on T2-weighted magnetic resonance. The regression of spinal cord edema took time. Therefore, the neuroprotection effect of HMPS@AcDX was observed on magnetic resonance imaging at day-1 after injury, not to mention the bare microspheres.

In this study, the amount of bare AcDX microspheres for HAcDX and LAcDX group was fixed at 4000 $\mu\text{g}/10 \mu\text{L}$ and 630 $\mu\text{g}/10 \mu\text{L}$, respectively. In our previous paper (Adv. Mater. 2018, 30, 1706032), the dose for AcDX microsphere was set at 300 $\mu\text{g}/5 \mu\text{L}/\text{injection}$ (10 μL for each rat). Therefore, only the LAcDX group is comparable with our previous study. In this current study, LAcDX significantly reduced the extent of cell apoptosis in the peritraumatic zone of spinal cord (**Fig. 5j and k**), which is similar to our previous paper (Adv. Mater. 2018, 30, 1706032). Moreover, the LAcDX group also significantly inhibited the activation of microglia near the lesion site and protected the injured neurons. Additionally, the CSPGs deposition was most serious in lesion area of rats in the HAcDX group (**Fig. 6e**), suggesting the adverse effect of high dose bare AcDX microspheres on the regeneration of axons. Our results suggested that the dosage of AcDX microspheres has an important impact on the therapeutic outcome after spinal cord injury.

• The question was raised about the frequency of withdrawal of CSF for drug assays. Based on the previous method, it was not possible to withdraw 9 times CSF. Now the narrative has been changed that different animals were used. Still, the description in figure legend states n=7 which does not make sense as the last two data point should not have any animal left as total number of data points are 9 and CSF was withdrawn only once from the animal. This is not just a simple error. It would be difficult to reproduce data without complete details.

Answer: Thank you for the comment. We apologize for the lack of clarity. We have updated our methods and checked our reporting to rectify this situation. We would like to mention that we always withdraw CSF samples from different animals. For example, in our previous paper (Adv. Mater. 2018, 30, 1706032), we allocated ≥ 3 rats in each time point of each group. Just one simple experiment for measuring the concentration of glutamate and calcium ion in cerebrospinal fluids, we needed more than 60 rats for each group.

We have updated our figure caption to rectify this situation:

Fig. 5. HMPS@AcDX microspheres improve motor function and reduce spinal cord edema. (a) Microspheres were intrathecally injected within 5 min after weight drop at T10 level. (b-d) MPS concentration in cerebral spinal fluid as a function of time (b), and the corresponding area under curve (AUC; c) and mean residence time (MRT; d). **For each group, $n \geq 5$ rats per time point; every rat was only sampled once.** (e) The rats were functionally graded up to 28 days post-injury by BBB grading scale, **$n \geq 7$ rats per group.** (f) Sagittal spinal cord sections stained with Nissl display the injury area of spinal cord at day-28 post trauma. (g) Comparison of the lesion volume after treatment (**$n = 5$ rats per group**). (h) Sagittal and axial spinal cord T2 weighted images at day-28 after injury. (1) & (1') are axial images in injury epicenter, (2) & (2') are axial images in far-injury area, (1') & (2') are axial color maps transformed from (1) and (2) normal axial images and the blue color represents the edema signal. (i) Intensity ratio for the signal of lesion to normal spinal cord and cord volume determined for consecutive slices at day-1 and day-28 post injury (**$n = 5$ rats for each group**). (j) TUNEL-positive apoptotic cells (in green) in sagittal spinal cord sections at day-1 post trauma. The nuclei of all cells were stained with DAPI (in blue). (k) Comparison of the number of TUNEL-positive cells after treatment (**$n = 6$ rats per group**).

• During the previous review, detail conditions for the release study were requested. That information is not provided (how much microparticles, how much buffer, etc.). It is just stated that sink condition was maintained. Without the detail information, one may not be able to reproduce the data. The information is critical considering methylprednisolone is not soluble in aqueous buffer.

Answer: Thank you for the comment. We apologize if we did not provide enough detailed information about drug release study in the last round of revision. We thank the review for further clarifying that the detail information about drug release study should include “*how much microparticles, how much buffer, etc.*”. We provide now these details as required.

Before the drug release studies, we measured the solubility of ATV, MPS and HCT in the media for drug release studies, *i.e.*, PBS (pH 7.4), which was determined to be $261 \pm 2 \mu\text{g/mL}$, $112 \pm 7 \mu\text{g/mL}$ and $726 \pm 25 \mu\text{g/mL}$, respectively. According to European Pharmacopeia (V8.0; 5.17.1. Recommendations on dissolution testing), “*Sink conditions normally occur in a volume of dissolution medium that is at least 3–10 times the saturation volume*”. We applied this principle to our drug release studies. The mass of samples and volume of medium are listed in Table S2 in the supplementary material.

Table S2. The mass of samples and volume of release media for drug release studies.

Sample name	Mass of sample (mg)	Volume of release media (mL)
ATV	1.7	50
ATV@AcDX	3.7	50
ATV@PLGA	3.0	50
MPS	1.2	50
MPS@AcDX	1.8	50
MPS@PLGA	5.0	50
HCT	2.7	50
HCT@AcDX	12.7	50
HCT@PLGA	5.7	50

• This reviewer still remains surprised with tight error bars for the BBB score. Weight drop method is known to cause variation in injury and also animal response varies to the same level injury. It has been well documented (e.g., Nature 2015 Feb 19;518(7539): 404-8).

Answer: Thank you for the comment. As mentioned in our response letter in the last round of revisions, a set of standard operating procedures was established to ensure the consistency of the weight-drop spinal cord injury model. For example, the vertebral plate was fully bitten to ensure the sufficient expose of the spinal cord tissue, and therefore, to minimize the dispersion of the strike force caused by the surrounding soft tissue coverage. Moreover, our standard operating procedure ensures the consistency of the strike position (in the midline of the spinal cord). Until 2 d after spinal cord injury, any rats with a BBB score ≥ 1 was removed from the study, ensuring the consistent injury level of spinal cord.

In our BBB score figure (Figure 2 below), only the plus direction of the error bar is presented, making the variation level looks smaller. Moreover, the scale of our system is larger than that presented in Lang *et al.* study. By changing our scale of BBB score to the same level, it can be easily observed that the SD values in our group are larger than those in Lang *et al.* study (Figure 2 below). We have carefully checked Lang *et al.* paper (Nature 2015, 518: 404-408) mentioned by the reviewer, and in this study, they found out that the recovery of injured spinal cord was accelerated by modulating the proteoglycan receptor $PTP\sigma$. We would like to mention that there is no description of the variation in injury for the weight drop method, not to mention the variation of animal response to the same level injury.

Figure 2. BBB scores of our study (a) and Lang's study (b).

Comments on abstract

The following statement in abstract, “Systems with high content of cargo can minimize excipients administration, reduce side effects, improve therapeutic efficacy and promote patient compliance. However, engineering such systems is extremely challenging, as their loading capacity is inherently limited by the compatibility between cargo molecules and carrier materials”, requires revision/editing. As it is written carries no meaning. For example, what Systems mean? Drug Delivery Systems or Microparticle-based drug delivery systems? The limitation of drug cargo content that the authors are implying are relevant to microparticle-based system because matrix type drug delivery systems can be prepared with high drug content. Also, it is clear how authors can claim that systems with high content can reduce side effect, improve therapeutic efficacy and promote patient compliance because the desired properties of drug delivery systems depend on the disease conditions. For example, dexamethasone eluting pacing electrode requires very low drug content in the matrix but for a longer duration to reduce scar tissue formation at the contact point. Drug eluting stents is another example where drug load is kept low so that the released is sustained. The point is that one cannot generalize a statement that high drug content in drug delivery systems can achieve all of the stated objectives. Also, authors need to specific to refer “cargo” as “drug cargo”.

Answer: Thank you for the suggestions. We made the changes in the revised abstract, accordingly.

We changed “Systems” to “Drug delivery systems”

We changed “cargo” to “drug”.

We changed “...and promote patient compliance” to “...and/or promote patient compliance”.

In our work, we prepared high TPE, ATV, MPS or HCT loaded microspheres. Drugs include ATV, MPS and HCT but TPE. Cargo could include not only drugs (ATV, MPS and HCT), but also TPE. Therefore, we would still need to use the word “cargo” in a few sentences of the manuscript when we talk about TPE, ATV, MPS or HCT loaded microspheres in general.

The following statement in abstract, “To mitigate the cargo-carrier compatibility limitation towards therapeutics encapsulation, we developed a sequential solidification strategy” also requires editing to be scientifically correct. Incompatibility of cargo-carrier system could be of different types such as drug interacting ionically or chemically to carrier system. The authors are specifically referring to “miscibility” of drug cargo-polymer and not general incompatibility.

Answer: Thank you for the comment. Compatibility is a broad term if there is no context or no limitations on its definition. In our manuscript, we consider the compatibility between a drug (cargo) and a polymer (carrier) to refer to miscibility (as written in the Introduction and Discussion), which helps the readers to more precisely understand the meaning of compatibility in our manuscript. Compatibility has been referred to miscibility in many other¹³⁻¹⁶, in which the influence of drug-polymer (cargo-carrier) compatibility on drug loading was¹³⁻¹⁵.

Regarding “*Incompatibility of cargo-carrier system could be of different types such as drug interacting ionically or chemically to carrier system.*”, this could be true in a different context. However, from the perspective of drug loading (the focus of our study), we would not consider the interaction between drug and carrier material to be certainly incompatible. Several researchers used the interactions (such as electrostatic attraction and donor-receptor coordination¹⁷⁻¹⁹) between drug molecules and carrier materials to increase the drug-carrier material compatibility and consequently improve the drug loading capacity. Nevertheless, we agree that “*drug interacting ionically or chemically to carrier system*” could be considered to be incompatible, if incompatibility is specifically defined in a different context.

The amount of acetylated dextran used was 1/433 of that of conventional microspheres. Which are these conventional microspheres. There is no reference found in the entire manuscript that describes “conventional microspheres” to compare with.

Answer: Thank you for the comment. We changed “*conventional microspheres*” to “*low drug loaded microspheres*”.

The statement that “the controlled release of methylprednisolone--- and reduced side effects than conventional formulations in spinal cord injury therapy. Which formulation is considered as conventional formulations? There is no toxicity study carried out to claim that the formulation developed in this study result in reduced side effects.

Answer: Thank you for the comment. We changed “*conventional formulations*” to “*low drug loaded microspheres and free drug*”.

HAcDX (4000 $\mu\text{g}/10 \mu\text{L}$) and LAcDX (630 $\mu\text{g}/10 \mu\text{L}$) equals the corresponding amount of bare AcDX microspheres for LMPS@AcDX and HMPS@AcDX, respectively. The neuroprotection effect was only observed in the LAcDX group (630 $\mu\text{g}/10 \mu\text{L}$). By increasing the dosage of AcDX to 4000 $\mu\text{g}/10 \mu\text{L}$, no protective effects could be observed in HAcDX group. In lesion area of rats, the most serious CSPGs deposition was even observed in HAcDX group (**Fig. 6e**). All these results suggested the adverse effect of high-dose bare AcDX microspheres on the recovery progress of injured spinal cord. When the MPS dose was fixed at 400 μg , the amount of AcDX used in HMPS@AcDX was 432-time less than that of LMPS@AcDX. Therefore, the high-drug loaded microspheres greatly reduced the amount of used carrier materials and consequently reduced the side effects caused by high-dose AcDX microspheres.

References:

- 1 Bracken, M. B. *et al.* A randomized, controlled trial of methylprednisolone or naloxone in the treatment of acute spinal-cord injury - results of the 2nd national acute spinal-cord injury study. *N. Engl. J. Med.* **322**, 1405-1411 (1990).
- 2 Koszdin, K. L., Shen, D. D. & Bernards, C. M. Spinal cord bioavailability of methylprednisolone after intravenous and intrathecal administration - The role of P-glycoprotein. *Anesthesiology* **92**, 156-163 (2000).
- 3 Liu, D. *et al.* Biodegradable spheres protect traumatically injured spinal cord by alleviating the glutamate-induced excitotoxicity. *Adv. Mater.* **30**, 1706032 (2018).
- 4 Liu, D. *et al.* An in situ gelling drug delivery system for improved recovery after spinal cord injury. *Adv. Healthc. Mater.* **5**, 1513-1521 (2016).
- 5 Bradbury, E. J. *et al.* Chondroitinase ABC promotes functional recovery after spinal cord injury. *Nature* **416**, 636-640 (2002).
- 6 Alilain, W. J., Horn, K. P., Hu, H., Dick, T. E. & Silver, J. Functional regeneration of respiratory pathways after spinal cord injury. *Nature* **475**, 196-200 (2011).
- 7 Li, Y. *et al.* Microglia-organized scar-free spinal cord repair in neonatal mice. *Nature* (2020). DOI: 10.1038/s41586-020-2795-6.
- 8 Peng, W. *et al.* Systemic administration of an antagonist of the ATP-sensitive receptor P2X7 improves recovery after spinal cord injury. *Proc. Natl. Acad. Sci. USA* **106**, 12489-12493 (2009).
- 9 Gaudin, A. *et al.* Squalenoyl adenosine nanoparticles provide neuroprotection after stroke and spinal cord injury. *Nat. Nanotechnol.* **9**, 1054-1062 (2014).
- 10 Wu, W. *et al.* Neuroprotective ferulic acid (FA)-glycol chitosan (GC) nanoparticles for functional restoration of traumatically injured spinal cord. *Biomaterials* **35**, 2355-2364 (2014).
- 11 Wu, H. F. *et al.* The promotion of functional recovery and nerve regeneration after spinal cord injury by lentiviral vectors encoding Lingo-1 shRNA delivered by Pluronic F-127. *Biomaterials* **34**, 1686-1700 (2013).
- 12 Wang, Y. C. *et al.* Sustained intraspinal delivery of neurotrophic factor encapsulated in

- biodegradable nanoparticles following contusive spinal cord injury. *Biomaterials* **29**, 4546-4553 (2008).
- 13 Liu, J., Xiao, Y. & Allen, C. Polymer–drug compatibility: A guide to the development of delivery systems for the anticancer agent, ellipticine. *J. Pharm. Sci.* **93**, 132-143 (2004).
- 14 Lübtow, M. M., Haider, M. S., Kirsch, M., Klisch, S. & Luxenhofer, R. Like Dissolves Like? A comprehensive evaluation of partial solubility parameters to predict polymer–drug compatibility in ultrahigh drug-loaded polymer micelles. *Biomacromolecules* **20**, 3041-3056 (2019).
- 15 Zhang, Y. *et al.* Strategies for improving the payload of small molecular drugs in polymeric micelles. *J. Control. Release* **261**, 352-366 (2017).
- 16 Turpin, E. R. *et al.* In silico screening for solid dispersions: the trouble with solubility parameters and χ_{FH} . *Mol. Pharm.* **15**, 4654-4667 (2018).
- 17 Peng, L. *et al.* Monolayer nanosheets with an extremely high drug loading toward controlled delivery and cancer theranostics. *Adv. Mater.* **30**, 1707389 (2018).
- 18 Liu, C. *et al.* Novel Inhalable ciprofloxacin dry powders for bronchiectasis therapy: mannitol–silk fibroin binary microparticles with high-payload and improved aerosolized properties. *AAPS PharmSciTech* **20**, 85 (2019).
- 19 Lv, S. *et al.* High drug loading and sub-quantitative loading efficiency of polymeric micelles driven by donor–receptor coordination interactions. *J. Am. Chem. Soc.* **140**, 1235-1238 (2018).

Reviewers' comments:

Reviewer #3 (Remarks to the Author):

The revised version reads better due to corrections and clarifications, but the core concerns remain which are summarized below.

- In response to the previous critique, authors now agree that the treatment given immediately after the injury has not clinically relevant but used as a proof-of-concept study to evaluate the efficacy of high-loaded microspheres. There are multiple preclinical studies in Spinal Cord Injury which did not translate to human spinal cord injury for the reason that those were not carried out under clinically relevant conditions, and this study will be one of those. With spinal cord injury, it is most critical to test the treatment several hrs after the injury (at least 3 to 4 hrs- considered as acute phase) as the pathophysiology of the injury response dramatically changes with time. Further, the effect of methylprednisolone depends on when the treatment is given. Therefore, it is strongly recommended that authors carry out the study several hours after the injury.
- It is indicated the number of animals used for functional recovery study (Figure legend 5e) to $n \geq 7$ rats per group. However, the data set (excel file) shows the number of animals $n=10$ per group. The data set shows that there is the mortality of animals treated with intrathecal injection of microparticles—up to 30%. However, this mortality is not seen in control animals (SCI that received saline or methylprednisolone in solution). So, the mortality is not due to SCI but due to intrathecal injection of microparticles. These microparticles are quite large (25 μm) and they could aggregate in the presence of CSF proteins, potentially blocking the flow of CSF. Interestingly, mortality is seen on the day of microparticles injection. There is complete omission of these results and discussion around the risk associated with injecting large-sized microparticles via intrathecal route. Others have shown significantly better efficacy and safety via direct injection of methylprednisolone-loaded microparticles at the lesion site (Biomaterials, 2009 May;30(13):2582-9). Therefore, the rationale for the risky intrathecal injection as it causes mortality is not clear.
- This concern is related to the data shown in Fig 5. Based on the description on page 16, LAcDX (control without MP) is the same as high dose methylprednisolone loaded microparticles (HMPs@AcDX) in terms of the amount of the delivery vehicle used (630 μg). LAcDX i.e., control microparticles show significant protective efficacy by themselves as compared to saline control (Fig 5 data), and the results are similar to HMPs@AcDX i.e., the delivery system with MP. The point is the control vehicle has its significant effect (also per a previously published study in Advanced Materials) and the improvement with the addition of drug into the formulation seems to be marginal in many parameters. There appears no statistical analysis of the data between the above two groups to demonstrate that claimed high dose MP microparticles indeed makes any difference.
- There is no clarification as to why delivery of high doses of LAcDX (control vehicle) has adverse or no effect than the low dose of the vehicle. It looks like there is dose-dependent toxicity.

RESPONSE TO THE REVIEWER'S COMMENTS

Reviewer #3:

The revised version reads better due to corrections and clarifications, but the core concerns remain which are summarized below.

Answer: We thank the reviewer for the time spent in reviewing our work and for the suggestions to improve our paper.

The response to the reviewer's comments is addressed below point-by-point. All the modifications/changes were also introduced in the revised version of the manuscript.

• In response to the previous critique, authors now agree that the treatment given immediately after the injury has not clinically relevant but used as a proof-of-concept study to evaluate the efficacy of high-loaded microspheres. There are multiple preclinical studies in Spinal Cord Injury which did not translate to human spinal cord injury for the reason that those were not carried out under clinically relevant conditions, and this study will be one of those. With spinal cord injury, it is most critical to test the treatment several hrs after the injury (at least 3 to 4 hrs- considered as acute phase) as the pathophysiology of the injury response dramatically changes with time. Further, the effect of methylprednisolone depends on when the treatment is given. Therefore, it is strongly recommended that authors carry out the study several hours after the injury.

Answer: Thank you for the comment. As answered in the last round of review, we acknowledge the injection time after injury of this study is not representative of real-world conditions. Also, there are differences in physiological and pathological characters between human and animal. There is always room to improve the animal mode. However, as a common method used in a range of tests, we believe the immediate injection after injury is suitable for a proof-of-concept study aimed at the development of a new strategy for preparing high-drug loaded microspheres.

We added the following text to the Discussion part: "*In addition, although this in vivo proof-of-concept study confirmed the advantages of high drug loaded microspheres, before*

clinical translation, the therapeutic outcomes of these microspheres still need to be verified in spinal cord injury animal models in which the treatment is given at least a few hours after the injury?

• It is indicated the number of animals used for functional recovery study (Figure legend 5e) to $n \geq 7$ rats per group. However, the data set (excel file) shows the number of animals $n=10$ per group. The data set shows that there is the mortality of animals treated with intrathecal injection of microparticles—up to 30%. However, this mortality is not seen in control animals (SCI that received saline or methylprednisolone in solution). So, the mortality is not due to SCI but due to intrathecal injection of microparticles. These microparticles are quite large (25 μm) and they could aggregate in the presence of CSF proteins, potentially blocking the flow of CSF. Interestingly, mortality is seen on the day of microparticles injection. There is complete omission of these results and discussion around the risk associated with injecting large-sized microparticles via intrathecal route. Others have shown significantly better efficacy and safety via direct injection of methylprednisolone-loaded microparticles at the lesion site (Biomaterials, 2009 May;30(13):2582-9). Therefore, the rationale for the risky intrathecal injection as it causes mortality is not clear.

Answer: Thank you for the comment. Indeed, the raw data (Source data, Fig. 5e) shows the number of animals is $n = 10$ per group. There was death of animals in some groups. Therefore, in the figure legend of Fig. 5e it is not $n = 10$ per group anymore. We had provided the raw data and marked the n values as well as death of animals during initial submission according to the journal requirements.

We would like to point out the exact mortality of animals treated with intrathecal injection of microspheres in different amount.

Actually, only one animal was dead in the high drug loaded microspheres (HMPS@AcDX) group (low amount of microspheres), and no animal was dead in LAcDX group (no drug, low amount of microspheres). Therefore, the mortality of animals treated with intrathecal injection of microspheres in low amount (630 μg) is $1/20=5\%$.

Two animals were dead in the low drug loaded microspheres (LMPS@AcDX) group

(high amount of microspheres), and three animals were dead in HAcDX group (no drug, high amount of microspheres). Therefore, the mortality of animals treated with intrathecal injection of microspheres in high amount (4000 µg) is 5/20=25%.

Only one out of 20 animals was dead when injected with low amount of microspheres. We do not know the exact reason for the death of this animal yet, which needs further careful investigation, however one out of 20 is far away from a high mortality rate. The 25% mortality rate in groups injected with high amount of microspheres can prove the advantages of our high drug loaded microspheres, which can greatly reduce the use of materials/excipients and associated side effects/safety risk.

We would like to point out that the reference (Biomaterials, 2009, 30(13): 2582-9) mentioned by Reviewer #3 actually reported methylprednisolone-loaded nanoparticles (200–700 nm) rather than microparticles. This reference did not provide the raw data and did not report the mortality of animals. Therefore, it is not clear whether it has significantly better safety.

Usually, the level of total proteins in CSF is < 50 µg/mL (J. Neurotrauma, 2015, 32: 1658-1665.)¹, which is quite low in comparison with those in blood. Moreover, high MPS-loading degree has greatly reduced the number of administrated microspheres, which can minimize the risk of microsphere aggregation in CSF. We have checked the stability of microspheres in CSF, which showed no significant aggregation for 1 week. Overall, the impact of CSF total protein on the dispersity of microspheres is negligible.

• This concern is related to the data shown in Fig 5. Based on the description on page 16, LAcDX (control without MP) is the same as high dose methylprednisolone loaded microparticles (HMPs@AcDX) in terms of the amount of the delivery vehicle used (630 µg). LAcDX i.e., control microparticles show significant protective efficacy by themselves as compared to saline control (Fig 5 data), and the results are similar to HMPs@AcDX i.e., the delivery system with MP. The point is the control vehicle has its significant effect (also per a previously published study in Advanced Materials) and the improvement with the addition of drug into the formulation seems to be marginal in many parameters. There appears no

statistical analysis of the data between the above two groups to demonstrate that claimed high dose MP microparticles indeed makes any difference.

Answer: Thank you for the comment. We performed statistical analysis of the Fig. 5 data between LAcDX and HMPS@AcDX, to clarify whether the delivered MPS contributed to the therapeutic effects of HMPS@AcDX.

Fig. 5e (LAcDX compared with HMPS@AcDX): $P < 0.001$ for 28 d; $P < 0.001$ for 21 d; $P < 0.01$ for 14 d; there is no statistical difference for 7 d.

Fig. 5g (LAcDX compared with HMPS@AcDX): $P < 0.01$.

Fig. 5i (LAcDX compared with HMPS@AcDX): $P < 0.01$ for relative intensity at 28d; $P < 0.01$ for lesion volume at 28d.

Fig. 5k (LAcDX compared with HMPS@AcDX): $P < 0.05$.

The results of these statistical analysis had been included in the updated Fig. 5, as follows.

Fig. 5. HMPs@AcDX microspheres improve motor function and reduce spinal cord edema. (a) Microspheres were intrathecally injected within 5 min after weight drop at T10 level. (b-d) MPS concentration in cerebral spinal fluid as a function of time (b), and the corresponding area under curve (AUC; c) and mean residence time (MRT; d). For each group, $n \geq 5$ rats per time point; every rat was only sampled once. (e) The rats were functionally graded up to 28 days post-injury by BBB grading scale, $n \geq 7$ rats per group. (f) Sagittal spinal cord sections stained with Nissl display the injury area of spinal cord at day-28 post trauma. (g) Comparison of the lesion volume after treatment ($n = 5$ rats per group). (h) Sagittal and axial spinal cord T2 weighted images at day-28 after injury. (1) & (1') are axial images in injury epicenter, (2) & (2') are axial images in far-injury area, (1') & (2') are axial

color maps transformed from (1) and (2) normal axial images and the blue color represents the edema signal. (i) Intensity ratio for the signal of lesion to normal spinal cord and cord volume determined for consecutive slices at day-1 and day-28 post injury ($n = 5$ rats for each group). (j) TUNEL-positive apoptotic cells (in green) in sagittal spinal cord sections at day-1 post trauma. The nuclei of all cells were stained with DAPI (in blue). (k) Comparison of the number of TUNEL-positive cells after treatment ($n = 6$ rats per group). The intervention groups were compared with the SCI group (*); HMPS@AcDX group was compared with the LMPS@AcDX group (#) and LAcDX group (\perp); the levels of significance were set at probabilities of *, #, \perp $P < 0.05$, **, ##, $\perp\perp$ $P < 0.01$, ***, ###, $\perp\perp\perp$ $P < 0.001$.

In addition, we also performed statistical analysis of the Fig. 6 data between LAcDX and HMPS@AcDX, to clarify whether the delivered MPS contributed to the therapeutic effects of HMPS@AcDX.

Fig. 6b (LAcDX compared with HMPS@AcDX): $P < 0.05$ for GFAP; $P < 0.001$ for CD68.

Fig. 6e (LAcDX compared with HMPS@AcDX): $P < 0.05$.

Fig. 6h (LAcDX compared with HMPS@AcDX): $P < 0.05$ for NF200; $P < 0.05$ for MBP.

The results of these statistical analysis had been included in the updated Fig. 6, as follows.

Fig. 6. HMPS@AcDX inhibits gliosis, attenuates the activation of microglia, suppresses CS56 production, and protects axon and myelin sheath. (a) Representative immunohistochemical staining of GFAP (in green) and CD68 (in red) in longitudinal sections of injured spinal cord at day-28 post injury. The nuclei of all cells were stained with DAPI (in blue). (b) Semi-quantification of GFAP intensity and density of microglia in the injured spinal cord. For GFAP intensity, the data are plotted as the relative ratio of the immunoreactivity near the injury site compared with that in distant area. ($n = 6$ per group). (c) Representative immunohistochemical staining of GFAP (in green) and CD68 (in red) adjacent to the lesion area. (d) The deposition of chondroitin sulfate proteoglycans (CSPGs) determined by CS56 antibody (in red) at day-28 after injury. (e) Semi-quantification of CS56 intensity increase in

the traumatic lesion area. The data are plotted as the relative ratio of the immunoreactivity near the injury site compared with that in the distant area. (f) Enlarged immunohistochemical staining of GFAP (in green) and CS56 (in red) adjacent to the lesion area. (g) Immunohistochemical staining of MBP (in green) and NF200 (in red) in injured spinal cord at day-28 post trauma. (h) Semi-quantification of NF200 and MBP intensity after spinal cord injury therapy. The data are plotted as the relative ratio of the immunoreactivity near the injury site compared with that in distant area. ($n = 6$ per group). (i) Representative immunohistochemical staining of MBP (in green) and NF200 (in red) adjacent to the lesion area. The intervention groups were compared with the SCI group (*); HMPS@AcDX group was compared with the LMPS@AcDX group (#) and LAcDX group (\perp); the levels of significance were set at probabilities of *, #, \perp $P < 0.05$, **, ##, $\perp\perp$ $P < 0.01$, ***, ###, $\perp\perp\perp$ $P < 0.001$.

In summary, HMPS@AcDX indeed had significantly better therapeutic effects than LAcDX, and the MPS delivered by HMPS@AcDX indeed made significance difference in therapeutic effects.

• There is no clarification as to why delivery of high doses of LAcDX (control vehicle) has adverse or no effect than the low dose of the vehicle. It looks like there is dose-dependent toxicity.

Answer: Thank you for the comment. AcDX microspheres show the capability of buffering out excess glutamate and calcium ions. Benefiting from the fast sequestering of glutamate and calcium ions by AcDX microspheres, significantly lower level of glutamate and calcium ions in CSF has been achieved at the first evaluation time point (8 h post-injury). The higher the dose of AcDX, the lower the level of glutamate and calcium ions. With the high dose of AcDX (HAcDX), the calcium concentration in CSF was 0.6 mM, which is even lower than the normal level in CSF. Too low concentration of calcium ions may adversely affect the neuronal excitability and the function of spinal cord (J. Neurochem., 2019, 149: 452-470)².

In the previous version of our manuscript, we had already mentioned this phenomenon

and explained it in the Results–“High MPS-loaded microspheres improve motor function and reduce spinal cord edema.” part: “*In comparison to the SCI group, LAcDX exhibited a faster rising trend in the BBB scores, but an inverse trend for HAcDX. This phenomenon was in consist with our previous study, in which AcDX microspheres with a concentration of 1 mg/mL protected neurons from glutamate-induced excitotoxicity, and therefore, repaired the injured spinal cord* (Reference 25 of manuscript file: Adv. Mater. 30, 1706032 (2018)).”. We had highlighted this sentence in the revised manuscript.

References:

- 1 Anderson, K. M. et al. Acute phase proteins in cerebrospinal fluid from dogs with naturally-occurring spinal cord injury. *J. Neurotrauma* **32**, 1658-1665 (2015).
- 2 Forsberg, M. et al. Ionized calcium in human cerebrospinal fluid and its influence on intrinsic and synaptic excitability of hippocampal pyramidal neurons in the rat. *J. Neurochem.* **149**, 452-470 (2019).

REVIEWER COMMENTS

Reviewer #3 (Remarks to the Author):

The reviewer appreciates the author's further clarification that

1. there is significant mortality in the animals that were treated with microparticles via intrathecal injection;
2. there is some aggregation of microparticles in the cerebrospinal fluid; and
3. the study is a "proof of concept" rather than the treatment for Spinal Cord Injury.

All of the above points significantly diminish the impact of the study and rationale for the use of high-dose methylprednisolone microparticles for the treatment of spinal cord injury. It is also noticed that authors have changed the title of the manuscript but still refers to the treatment for spinal cord injury throughout the manuscript. Thus, there is a disconnect between the title and the manuscript content.

The reviewer would like to propose that

Major Points

- Authors thoroughly investigate why there is high mortality in the microparticle treated group and design an appropriate delivery system. The data from this set of experiments will be helpful so that others do not attempt to develop similar systems for intrathecal injection. It could be the size, charge, high drug load, the polymer used, aggregation, or combination of factors that could be the cause of high mortality.
- Conduct the study under clinically relevant conditions, i.e., giving treatment several hours after the injury. It would be very useful comparative data. What happens when the treatment is given immediately and few hours after the injury?
- Authors did not discuss the mortality issue in the main manuscript, which is a critical point to mention and did not correct n=10 in the figure legend.

Minor points

- Authors did not provide apple to apple comparison while justifying the method developed. There are formulations with high drug dose-load.

While advocating the proposed microfluidic method for formulating high-dose microparticles, the authors used the magnetic stirring method to suggest that the conventional method does not work. So, using the worst condition for comparison to make a claim. Perhaps, high-speed homogenization

is needed, or ethyl acetate is not the right solvent for making microparticles by the conventional method.

- It is a weak argument that the Biomaterial study where nanoparticles vs. microparticles were used, and that study did not report mortality (an assumption that there was mortality and authors did not report), it is acceptable to have 25% mortality via intrathecal injection as seen in this study.

Reviewer #4 (Remarks to the Author):

The authors responses to the concerns raised by Reviewer 3 are solid and well justified, particularly their responses to concerns over potential safety, which were robustly addressed.

Reviewer 3's concerns with safety seem to be over the safety of the microspheres themselves. However, I was surprised that no comment has been made regarding the selection of drug to be administered by microspheres – methylprednisolone.

Methylprednisolone (MPS) has a controversial history for the treatment of spinal cord injury. In the paper, MPS is merely referenced as a beneficial anti-inflammatory drug, and the paper cited (Bracken et al 1990) was indeed the paper that stimulated MPS to be approved for clinical use for spinal cord injury. However, the opinion of MPS as a standard of care for SCI has changed dramatically. There is now a large body of evidence indicating that MPS either has no beneficial effects, or has modest effects but only when delivered in a strict regimen within the first 24 hours, as well as numerous reports stating that it is actually detrimental to recovery and should not be used clinically as the risks outweigh the benefits e.g.:

<https://pubmed.ncbi.nlm.nih.gov/28447605/>

<https://pubmed.ncbi.nlm.nih.gov/33684579/>

<https://pubmed.ncbi.nlm.nih.gov/10879751/>

<https://pubmed.ncbi.nlm.nih.gov/29164025/>

The authors seem completely unaware of these controversies. In relation to these issues with MPS, the CSF data seemed to indicate that free MPS delivery yielded a much higher concentration over the first 24 hours in vivo than the encapsulated MPS, which perhaps explains this group having the worst functional outcome, given the evidence that high dose MPS may be detrimental to tissue protection and recovery (as discussed above, but not by the authors). It is possible that the lower

concentration of MPS evident in the first 24 hours and maintained at a stable level for the next 7 days with the novel encapsulated MPS system is the reason for improved recovery and tissue preservation. Perhaps the authors have found an optimal method of delivering this controversial drug, which (at this stable dose and delivery method) can enable the putative benefits of MPS, without the associated risks. This indeed would be a major advance, and addresses what is considered to be an important goal by many in the field, to find "...novel drug delivery systems that have demonstrated the potential to improve MP's bioavailability at the site of injury while minimizing systemic side effects" (Canseco et al 2021).

In this regard, this paper could be extremely influential, but the authors would need to address and discuss these implications.

For other aspects, I cannot comment extensively on the on the engineering aspects, although the nano-in-micro structured microsphere configuration seems an elegant approach to achieve highly efficient drug encapsulation, so I do not have a problem with the approach.

I also have no issue with the authors applying the treatment immediately after injury (another concern of reviewer 3) – this is standard in our field, where new therapeutics are typically administered immediately after injury to obtain proof of principle data of effectiveness, with subsequent follow up studies investigating the therapeutic time window. This has also been clearly stated by the authors.

As stated above, my main concern for this paper is the drug choice, the lack of awareness of the controversies of using this drug, the implications of what their drug delivery system may reveal, and what this may mean in terms of impact of the paper. These can, however, be addressed in discussion points and in the introduction.

Minor:

Line 357-368: The BBB locomotor scoring data (Fig. 5e) should be described in more detail, including the performance of the free MPS group, in comparison to SCI only and encapsulated MPS. The free MPS group appears to be one of the worst performing groups (as discussed above) but this is not described in the text.

Line 368: the phrase "...and therefore repaired the injured spinal cord 25." Should be removed or modified: any beneficial effects shown here were neuroprotective (i.e. stopped some of the wave of secondary injury, resulting in tissue preservation rather than repair), which is far different from repairing an injured spinal cord, which suggests axon regeneration, reconnection and restoration of injured tissue.

Line 444: "...the CSPGs deposition was most serious in lesion area of rats in HAcDX group..." could be rephrased e.g. "...CSPG deposition was most highly expressed in lesioned tissue of the HAcDX group..." .

Line 458-459: "As shown in the enlarged immunohistochemical staining images, MBP continuously wrapped around the axon (NF200 positive)..." . The images in Fig. 6i are not high resolution enough to see MBP "wrapping around an axon" – rephrase.

RESPONSE TO THE REVIEWER'S COMMENTS

Reviewer #3:

The reviewer appreciates the author's further clarification that

1. there is significant mortality in the animals that were treated with microparticles via intrathecal injection;
2. there is some aggregation of microparticles in the cerebrospinal fluid; and
3. the study is a "proof of concept" rather than the treatment for Spinal Cord Injury.

All of the above points significantly diminish the impact of the study and rationale for the use of high-dose methylprednisolone microparticles for the treatment of spinal cord injury. It is also noticed that authors have changed the title of the manuscript but still refers to the treatment for spinal cord injury throughout the manuscript. Thus, there is a disconnect between the title and the manuscript content.

Answer: We thank the reviewer for the time spent in reviewing our work and for the suggestions to improve our paper. The response to the reviewer's comments is addressed below point-by-point. All the modifications/changes were also introduced in the revised version of the main manuscript and supplementary material.

Regarding the mortality of animals treated with microspheres via intrathecal injection, as answered in the last round of revisions, the mortality of animals treated with microspheres in low dose (630 µg) and high dose (4000 µg) is 5% (1/20) and 25% (5/20), respectively. Indeed, there was a significant mortality (25%) in the groups injected with high dose microspheres, but these groups are control groups that correspond to the microspheres with low drug loading. Only one out of 20 animals died in the groups injected with low dose microspheres, and these groups are treatment groups that correspond to the developed high drug loaded microspheres. Furthermore, we performed the suggested *in vivo* studies to compare the therapeutic effects of treatments given at different time points after injury, and no animal died in the groups injected with low dose microspheres (10 animals per group, 2 groups, and 2 time points, in total 40 animals). Therefore, there was no significant mortality in the animals that were treated with high drug loaded microspheres via intrathecal injection.

Regarding the aggregation of microparticles in the cerebrospinal fluid, as answered in the last

round of revisions, there was no significant aggregation of microspheres in CSF for 1 week.

Since more *in vivo* results about SCI therapy were included in the revised manuscript, we changed the title back to “*A high drug loaded microsphere enabled by controlled in-droplet precipitation promotes functional recovery after spinal cord injury*”.

The reviewer would like to propose that

Major Points

- Authors thoroughly investigate why there is high mortality in the microparticle treated group and design an appropriate delivery system. The data from this set of experiments will be helpful so that others do not attempt to develop similar systems for intrathecal injection. It could be the size, charge, high drug load, the polymer used, aggregation, or combination of factors that could be the cause of high mortality.

Answer: Thank you for the comment. As answered to the above question, there was significant mortality of animals in LMPS@AcDX and HAcDX groups treated with high dose microspheres, but not in HMPS@AcDX and LAcDX groups treated with low dose microspheres. Since HAcDX and LAcDX are the same AcDX microspheres, the mortality shall not be related to the size of the microspheres. Since HMPS@AcDX did not lead to significant mortality, the mortality shall not be related to the high drug loading. Also, there was no significant aggregation of microspheres in CSF. These results suggest that the mortality is more likely to be dependent on the dose of the microspheres.

We evaluated the capture capability of AcDX microspheres for glutamate and calcium ions by incubating AcDX microspheres with glutamate and calcium ions solutions. After 30 minutes incubation, each sample was centrifuged at 1000 ×g for 10 min to remove AcDX microspheres. The free glutamate in the supernatant was quantified by glutamate assay kit (Abcam, UK). For studying the effect of AcDX concentration on the capture of glutamate, the glutamate concentration was fixed at 50 μM. With a fixed AcDX microsphere concentration (1 and 8 mg/mL), the effect of glutamate concentrations (25, 50 and 100 μM) on the capture efficiency of glutamate was studied. We also studied the calcium ion sequestering capability of AcDX microspheres. The calcium ion content in the supernatant was measured using an automatic analyzer (AU5800; Beckman Coulter, Inc., USA). We also evaluated the effect of calcium ion concentrations on the sequestering efficiency of calcium ions

by AcDX microspheres.

As shown in Fig. R1, the prepared AcDX microspheres have the capability to adsorb glutamate and calcium ions. In our previous study¹, benefiting from the fast sequestering of glutamate and calcium ions by AcDX microspheres, significantly lower level of glutamate and calcium ions in CSF has been achieved at the first evaluation time point (8 h post-injury) after intrathecal administration of AcDX microspheres (600 μg)¹. The higher the dose of AcDX, the lower the level of glutamate and calcium ions (Fig. R1). Therefore, with the administration of HAcDX and LMPS@AcDX (4000 μg AcDX microspheres), the calcium ion concentration in CSF is expected to be rapidly decreased to a low level within few hours after administration. Too low concentration of calcium ions may have adverse influence on the neuronal excitability and consequently adversely affect the function of spinal cord² and the central respiratory control system³.

Fig. R1: (a) The adsorption of glutamate (50 μM) by AcDX microspheres in terms of the microsphere concentrations ($n = 5$). (b) The impact of glutamate concentrations on the adsorption of glutamate by AcDX microspheres ($n = 5$). (c) The adsorption of calcium ions (0.5 mM) by AcDX microspheres in terms of the microsphere concentrations ($n = 5$). (d) The impact of calcium ion concentrations on the adsorption of calcium ions by AcDX microspheres ($n = 5$).

The death of animals treated with high dose microspheres might also be related to other factors. It

should be noted that many materials/excipients, considered “inactive ingredients” and ubiquitous in drug formulations, actually have direct activities against biologically relevant enzymes, receptors, ion channels, or transporters, leading to potential dose-dependent side effects⁴. Comprehensive studies would be needed to find the exact reason for side effects including mortality caused by a particular material/excipient. Nevertheless, it is possible to significantly reduce or even avoid the side effects including mortality by lowering the dose of administered materials/excipients. As shown in this study, 5 out of 20 animals died in groups treated with high dose microspheres, but only 1 out of 60 animals died in groups treated with low dose microspheres. The developed high drug loaded microspheres have sufficient safety in the treatment of SCI.

- Conduct the study under clinically relevant conditions, i.e., giving treatment several hours after the injury. It would be very useful comparative data. What happens when the treatment is given immediately and few hours after the injury?

Answer: Thank you for the suggestion. As suggested, we performed additional *in vivo* studies to compare the therapeutic effects of immediate treatment (within 5 min after the injury) and delayed treatment (4 hours after the injury). Since control groups HAcDX and LMPS@AcDX did not show significant therapeutic effects in previous studies, they were not included in the additional *in vivo* studies to reduce the number of used animals. HMPS@AcDX and control groups (SCI, MPS and LAcDX) were evaluated.

After traumatic SCI, motor behavior was assessed in open-field by the 21-point Basso, Beattie, and Bresnahan (BBB) locomotor rating scale (Supplementary **Fig. S15**). For both immediate treatment and delayed treatment, throughout the entire assessment period, HMPS@AcDX recovered motor function significantly faster than the other groups including LAcDX. In addition, for HMPS@AcDX, there was no significant difference in the BBB score between immediate treatment and delayed treatment.

Fig. S15: The rats were functionally graded up to 28 days post-injury by BBB grading scale, $n = 10$ rats per group. The intervention groups were compared with the SCI group (*); HMPS@AcDX group was compared with LAcDX group (†); delayed treatment group was compared with immediate treatment group (#); **, †† $P < 0.01$, ***, ††† $P < 0.001$.

The development of a fluid-filled cystic cavity is one of the prominent pathological features for the injured spinal cord. Therefore, we examined the pathological change of the injured spinal cord by Nissl staining (Supplementary Fig. S16). Four weeks after injury, the lesion volume in the groups treated by LAcDX ($P < 0.001$) and HMPS@AcDX ($P < 0.001$) was notably smaller than that of the SCI group for both immediate treatment and delayed treatment. In comparison with LAcDX, HMPS@AcDX significantly reduced the loss of post-traumatic spinal cord tissue ($P < 0.001$ for immediate treatment and $P < 0.01$ for delayed treatment). In addition, for HMPS@AcDX, there was no significant difference in the lesion volume between immediate treatment and delayed treatment.

Fig. S16: (a) Sagittal spinal cord sections stained with Nissl display the injury area of spinal cord at day-28 post trauma. (b) Comparison of the lesion volume after treatment ($n = 5$ rats per group). The intervention groups were compared with the SCI group (*); HMPS@AcDX group was compared with LAcDX group (†); delayed treatment group was compared with immediate treatment group (#); †† $P < 0.01$, ***, ††† $P < 0.001$.

We also examined the density or status of astrocytes, microglia, neurons, and axons. All these cells play roles in the injury of spinal cord after the mechanical disruption. The activation of astrocytes and microglia near the lesion site was evaluated by immunostaining of glial fibrillary acidic protein (GFAP; in green) and CD68 (in red), respectively (Supplementary **Fig. S17**). As compared to SCI group, the treatment with LAcDX ($P < 0.01$) and HMPS@AcDX ($P < 0.001$) significantly inhibited the increase of GFAP immunoreactivity for both immediate treatment and delayed treatment. In comparison with LAcDX, HMPS@AcDX significantly inhibited the increase of GFAP immunoreactivity ($P < 0.01$ for immediate treatment and $P < 0.05$ for delayed treatment). In addition, for HMPS@AcDX, there was no significant difference in the GFAP immunoreactivity between immediate treatment and delayed treatment. The density of CD68-positive microglia in the lesion area for LAcDX was significantly smaller than that in SCI group ($P < 0.001$ for both immediate treatment and delayed treatment). HMPS@AcDX further significantly reduced the number of microglia in the traumatic lesion area compared to LAcDX group ($P < 0.001$ for both immediate treatment and delayed treatment). In addition, for HMPS@AcDX, there was no significant difference in the density of CD68-positive microglia between immediate treatment and delayed treatment.

Fig. S17: (a) Representative immunohistochemical staining of GFAP (in green) and CD68 (in red) in longitudinal sections of injured spinal cord at day-28 post injury. The nuclei of all cells were stained with DAPI (in blue). (b) Semi-quantification of GFAP intensity and density of microglia in the injured

spinal cord. For GFAP intensity, the data are plotted as the relative ratio of the immunoreactivity near the injury site compared with that in distant area. ($n = 6$ per group). The intervention groups were compared with the SCI group (*); HMPS@AcDX group was compared with LAcDX group (\perp); delayed treatment group was compared with immediate treatment group ($\#$); $\perp P < 0.05$, **, $\perp\perp P < 0.01$, ***, $\perp\perp\perp P < 0.001$.

Chondroitin sulfate proteoglycans (CSPGs) are considered as one of the principal inhibitors for axon regeneration. After SCI, these proteoglycans can be produced by astrocytic scars or diverse cells in SCI lesions including pericytes, fibroblast lineage cells and inflammatory cells. We examined cellular production of CSPG (CS56 antibody, in red) and GFAP (in green) (Supplementary Fig. S18). The CSPG level for LAcDX was significantly smaller than that in SCI group ($P < 0.001$ for both immediate treatment and delayed treatment). HMPS@AcDX further significantly reduced the CSPG level compared to LAcDX group ($P < 0.01$ for immediate treatment and $P < 0.001$ for delayed treatment). In addition, for HMPS@AcDX, there was no significant difference in the CSPG level between immediate treatment and delayed treatment.

Fig. S18: (a) The deposition of chondroitin sulfate proteoglycans (CSPGs) determined by CS56 antibody (in red) at day-28 after injury. (b) Semi-quantification of CS56 intensity increase in the traumatic lesion area. The data are plotted as the relative ratio of the immunoreactivity near the injury site compared with that in the distant area. ($n = 6$ per group). The intervention groups were compared with the SCI group (*); HMPS@AcDX group was compared with LAcDX group (\perp); delayed treatment group was compared with immediate treatment group ($\#$); $\perp\perp P < 0.01$, ***, $\perp\perp\perp P < 0.001$.

Myelin surrounds nerve cell axons to increase the rate at which electrical impulses is passed along the axon. Neurofilaments are qualified as potential surrogate markers of damage to neuron and axon. To identify the demyelination of residual or regenerated axons, we double-stained 200 kDa subunit of neurofilament (NF200) and the myelin basic-protein marker (MBP; Supplementary **Fig. S19**). For both immediate treatment and delayed treatment, in comparison with SCI group, a significant decrease in NF200 intensity was observed for LAcDX ($P < 0.001$) and HMPS@AcDX ($P < 0.001$) groups. HMPS@AcDX further significantly reduced the NF200 intensity compared to LAcDX group ($P < 0.01$ for both immediate treatment and delayed treatment). In addition, for HMPS@AcDX, there was no significant difference in the NF200 intensity between immediate treatment and delayed treatment. In comparison with SCI group, a significant decrease in MBP intensity was observed for LAcDX ($P < 0.001$ for both immediate treatment and delayed treatment) and HMPS@AcDX ($P < 0.001$ for both immediate treatment and delayed treatment) groups. HMPS@AcDX further significantly reduced the MBP intensity compared to LAcDX group ($P < 0.001$ for immediate treatment and $P < 0.01$ for delayed treatment). In addition, for HMPS@AcDX, there was no significant difference in the MBP intensity between immediate treatment and delayed treatment.

Fig. S19: (a) Immunohistochemical staining of MBP (in green) and NF200 (in red) in injured spinal cord at day-28 post trauma. (b) Semi-quantification of NF200 and MBP intensity after spinal cord injury therapy. The data are plotted as the relative ratio of the immunoreactivity near the injury site

compared with that in distant area. ($n = 6$ per group). The intervention groups were compared with the SCI group (*); HMPS@AcDX group was compared with LAcDX group (\dagger); delayed treatment group was compared with immediate treatment group ($\#$); $\dagger\dagger P < 0.01$, $***$, $\dagger\dagger\dagger P < 0.001$.

Overall, for both immediate treatment and delayed treatment, HMPS@AcDX had significant therapeutic efficacy in SCI therapy and its therapeutic efficacy was significantly higher than that of LAcDX. There was no significant difference in the therapeutic efficacy between immediate treatment (within 5 min after the injury) and delayed treatment (4 hours after the injury).

- Authors did not discuss the mortality issue in the main manuscript, which is a critical point to mention and did not correct $n=10$ in the figure legend.

Answer: Thank you for the comment. We added discussion about the mortality issue in the discussion part of the main manuscript and updated the n value in the figure caption.

Minor points

- Authors did not provide apple to apple comparison while justifying the method developed. There are formulations with high drug dose-load.

While advocating the proposed microfluidic method for formulating high-dose microparticles, the authors used the magnetic stirring method to suggest that the conventional method does not work. So, using the worst condition for comparison to make a claim. Perhaps, high-speed homogenization is needed, or ethyl acetate is not the right solvent for making microparticles by the conventional method.

Answer: Thank you for the suggestion. We used homogenization (5000 rpm) and solvents other than ethyl acetate (i.e., dichloromethane, dimethyl carbonate, and chloroform) to prepare the high drug loaded microspheres.

Like the magnetic stirring approach, homogenization failed to prepare high drug (ATV, MPS and HCT) loaded polymer (AcDX and PLGA) microspheres as illustrated by the formation of many micro-sized free drug crystals outside microspheres (Supplementary Fig. S13 and Fig. S14). Homogenization has a higher energy input than magnetic stirring. The strong and inhomogeneous turbulent mixing is expected to cause uncontrolled solvent diffusion and drug precipitation, and

breakage of emulsion droplets containing drug precipitates/crystals.

Fig. S13: Phase contrast microscope images of microspheres. ATV@AcDX, MPS@AcDX, HCT@AcDX, ATV@PLGA, MPS@PLGA and HCT@PLGA microspheres were prepared by homogenization (bulk method) using ethyl acetate as the primary solvent. Scale bars, 50 µm.

Fig. S14. X-ray powder diffractograms of microspheres prepared by homogenization (bulk method). Ethyl acetate was used as the primary solvent for preparing the microspheres.

Previously, high MPS loaded AcDX microspheres were prepared by magnetic stirring using ethyl acetate as the solvent, and it failed. As suggested, we changed the solvent from ethyl acetate to other solvents, i.e., dichloromethane, dimethyl carbonate, and chloroform, which are also the popular solvents used for emulsification. Both magnetic stirring and homogenization failed to prepare high MPS loaded AcDX microspheres when dichloromethane, dimethyl carbonate, and chloroform were used as the solvents, as illustrated by the formation of many micro-sized free drug crystals outside microspheres (**Fig. R2** and **Fig. R3** as shown below). Magnetic stirring and homogenization have much higher energy input than droplet microfluidics. Even though the solvent is changed, the strong and inhomogeneous turbulent mixing still exists during magnetic stirring and homogenization, and it is expected to cause uncontrolled solvent diffusion and drug precipitation, and, consequently, the

breakage of emulsion droplets containing drug precipitates/crystals.

Fig. R2: Phase contrast microscope images of MPS@AcDX microspheres. MPS@AcDX microspheres were prepared by bulk methods (homogenization and magnetic stirring) using dichloromethane, dimethyl carbonate, and chloroform as the primary solvent. Scale bars, 50 μm.

Fig. R3. X-ray powder diffractograms of MPS@AcDX microspheres prepared by bulk methods (homogenization and magnetic stirring) using dichloromethane, dimethyl carbonate, and chloroform as the primary solvent.

- It is a weak argument that the Biomaterial study where nanoparticles vs. microparticles were used, and that study did not report mortality (an assumption that there was mortality and authors did not report), it is acceptable to have 25% mortality via intrathecal injection as seen in this study.

Answer: Thank you for the comment. As discussed before, the high mortality (25%) only happened for control groups. Control groups represent microspheres with low drug loading, which leads to the use of high dose microspheres. The high mortality did not happen for the developed high drug loaded microspheres (treatment groups), which contributes to the use of low dose microspheres. These results prove the advantages of our high drug loaded microspheres, which can greatly reduce the use of

materials/excipients and associated side effects/safety risk. The developed high drug loaded microspheres have sufficient safety in the treatment of SCI.

References:

1. Dongfei L, *et al.* Biodegradable spheres protect traumatically injured spinal cord by alleviating the glutamate-induced excitotoxicity. *Adv Mater* **30**, 1706032 (2018).
2. Forsberg M, *et al.* Ionized calcium in human cerebrospinal fluid and its influence on intrinsic and synaptic excitability of hippocampal pyramidal neurons in the rat. *J Neurochem* **149**, 452-470 (2019).
3. Kuwana S-i, Okada Y, Natsui T. Effects of extracellular calcium and magnesium on central respiratory control in the brainstem–spinal cord of neonatal rat. *Brain Research* **786**, 194-204 (1998).
4. Pottel J, *et al.* The activities of drug inactive ingredients on biological targets. *Science* **369**, 403 (2020).

Reviewer #4:

The authors responses to the concerns raised by Reviewer 3 are solid and well justified, particularly their responses to concerns over potential safety, which were robustly addressed.

Answer: We thank the reviewer for the time spent in reviewing our work and for the suggestions to improve our paper.

The response to the reviewer’s comments is addressed below point-by-point. All the modifications/changes were also introduced in the revised version of the manuscript.

Reviewer 3’s concerns with safety seem to be over the safety of the microspheres themselves. However, I was surprised that no comment has been made regarding the selection of drug to be administered by microspheres – methylprednisolone.

Methylprednisolone (MPS) has a controversial history for the treatment of spinal cord injury. In the paper, MPS is merely referenced as a beneficial anti-inflammatory drug, and the paper cited (Bracken et al 1990) was indeed the paper that stimulated MPS to be approved for clinical use for spinal cord injury. However, the opinion of MPS as a standard of care for SCI has changed dramatically. There is

now a large body of evidence indicating that MPS either has no beneficial effects, or has modest effects but only when delivered in a strict regimen within the first 24 hours, as well as numerous reports stating that it is actually detrimental to recovery and should not be used clinically as the risks outweigh the benefits e.g.:

<https://pubmed.ncbi.nlm.nih.gov/28447605/>

<https://pubmed.ncbi.nlm.nih.gov/33684579/>

<https://pubmed.ncbi.nlm.nih.gov/10879751/>

<https://pubmed.ncbi.nlm.nih.gov/29164025/>

The authors seem completely unaware of these controversies. In relation to these issues with MPS, the CSF data seemed to indicate that free MPS delivery yielded a much higher concentration over the first 24 hours in vivo than the encapsulated MPS, which perhaps explains this group having the worst functional outcome, given the evidence that high dose MPS may be detrimental to tissue protection and recovery (as discussed above, but not by the authors). It is possible that the lower concentration of MPS evident in the first 24 hours and maintained at a stable level for the next 7 days with the novel encapsulated MPS system is the reason for improved recovery and tissue preservation. Perhaps the authors have found an optimal method of delivering this controversial drug, which (at this stable dose and delivery method) can enable the putative benefits of MPS, without the associated risks. This indeed would be a major advance, and addresses what is considered to be an important goal by many in the field, to find "...novel drug delivery systems that have demonstrated the potential to improve MP's bioavailability at the site of injury while minimizing systemic side effects" (Canseco et al 2021). In this regard, this paper could be extremely influential, but the authors would need to address and discuss these implications.

Answer: Thank you for the comment. Indeed, the use of systemic high dose MPS for the treatment of SCI is controversial due to modest improvements in neurological recovery and the risks of severe side effects. Most of the side effects of MPS therapy are related to the high systemic dose and associated toxicity, and the relatively modest neurological gains indicate inefficient dosing to the injured site. In comparison with free MPS, the developed high MPS loaded microspheres with controlled MPS release (*i.e.*, HMPS@AcDX) maintained a higher and more stable MPS concentration for 7 days. As expected, HMPS@AcDX significantly increased the area under curve and mean residence time of MPS in

cerebral spinal fluid (Fig. 5c–d), suggesting an improved bioavailability of MPS at the site of injury.

We added these discussions to the introduction and discussion parts of the revised version of the manuscript.

For other aspects, I cannot comment extensively on the on the engineering aspects, although the nano-in-micro structured microsphere configuration seems an elegant approach to achieve highly efficient drug encapsulation, so I do not have a problem with the approach.

I also have no issue with the authors applying the treatment immediately after injury (another concern of reviewer 3) – this is standard in our field, where new therapeutics are typically administered immediately after injury to obtain proof of principle data of effectiveness, with subsequent follow up studies investigating the therapeutic time window. This has also been clearly stated by the authors.

Answer: Thank you for the comment.

As stated above, my main concern for this paper is the drug choice, the lack of awareness of the controversies of using this drug, the implications of what their drug delivery system may reveal, and what this may mean in terms of impact of the paper. These can, however, be addressed in discussion points and in the introduction.

Answer: Thank you for the comment. We added discussion about these points in the introduction and discussion parts of the revised version of the manuscript.

Minor:

Line 357-368: The BBB locomotor scoring data (Fig. 5e) should be described in more detail, including the performance of the free MPS group, in comparison to SCI only and encapsulated MPS. The free MPS group appears to be one of the worst performing groups (as discussed above) but this is not described in the text.

Answer: Thank you for the comment. In comparison with SCI group, the administration of MPS, HAcDX, and LMPS@AcDX had no significant effects on the functional motor recovery, while LAcDX and HMPS@AcDX significantly improved the functional motor recovery from day-7 and day-3 post trauma, respectively. For HMPS@AcDX, the recovery of motor function was observed for

day-28 after injury, which was much faster than that of LMPS@AcDX (from day-7) and LAcDX (from day-14).

We added the text to the results part of the revised version of the manuscript.

Line 368: the phrase "...and therefore repaired the injured spinal cord 25." Should be removed or modified: any beneficial effects shown here were neuroprotective (i.e. stopped some of the wave of secondary injury, resulting in tissue preservation rather than repair), which is far different from repairing an injured spinal cord, which suggests axon regeneration, reconnection and restoration of injured tissue.

Answer: Thank you for the suggestion. We removed the phrase "...and therefore repaired the injured spinal cord."

Line 444: "...the CSPGs deposition was most serious in lesion area of rats in HAcDX group..." could be rephrased e.g. "...CSPG deposition was most highly expressed in lesioned tissue of the HAcDX group..."

Answer: Thank you for the suggestion. We rephrased the sentence "...the CSPGs deposition was most serious in lesion area of rats in HAcDX group..." to "...CSPG deposition was most highly expressed in lesioned tissue of the HAcDX group..." in the revised version of the manuscript.

Line 458-459: "As shown in the enlarged immunohistochemical staining images, MBP continuously wrapped around the axon (NF200 positive)...". The images in Fig. 6i are not high resolution enough to see MBP "wrapping around an axon" – rephrase.

Answer: Thank you for the comment. Indeed, the images in Fig. 6i have limited resolution. We added images with higher magnification to the supplementary material (Supplementary **Fig. S10**), which would help to see MBP wrapping around an axon.

Fig. S10: Representative immunohistochemical staining of MBP (in green) and NF200 (in red) adjacent to the lesion area.

REVIEWERS' COMMENTS

Reviewer #4 (Remarks to the Author):

Although the authors now acknowledge the controversy over the use of MPS as a treatment for spinal cord injury in the introduction, there is no mention of this at all in the discussion (despite their claim to have added discussion about this), or what it might mean in terms of clinical application of MPS. Would the authors recommend clinical use of MPS if given in this formulation, do they think they have found a solution for long-standing issue in the field, are there still concerns over safety, side-effects, morbidity and mortality of this therapy? And how do their anatomical data compare with the historical MPS studies? The authors need to comment and discuss these important points and the literature on this topic. Since my only main concern for this paper was "...the drug choice, the lack of awareness of the controversies of using this drug, the implications of what their drug delivery system may reveal, and what this may mean in terms of impact of the paper", it is disappointing that the authors chose not to address any of these points in the discussion.

RESPONSE TO THE REVIEWER'S COMMENTS

Reviewer #4:

Although the authors now acknowledge the controversy over the use of MPS as a treatment for spinal cord injury in the introduction, there is no mention of this at all in the discussion (despite their claim to have added discussion about this), or what it might mean in terms of clinical application of MPS. Would the authors recommend clinical use of MPS if given in this formulation, do they think they have found a solution for long-standing issue in the field, are there still concerns over safety, side-effects, morbidity and mortality of this therapy? And how do their anatomical data compare with the historical MPS studies? The authors need to comment and discuss these important points and the literature on this topic. Since my only main concern for this paper was "...the drug choice, the lack of awareness of the controversies of using this drug, the implications of what their drug delivery system may reveal, and what this may mean in terms of impact of the paper", it is disappointing that the authors chose not to address any of these points in the discussion.

Answer: Thank you for the comment. We apologize if we did not discuss these points more in detail in the Discussion section of the previous version of the manuscript.

High dose intrathecal administration of MPS has been clinically used for acute SCI therapy ¹. This treatment is controversial due to no or modest improvements in neurological recovery and the risks of severe side effects ^{2,3}, although the 2017 AOSpine guideline continues to recommend high dose MPS treatment within 8 hours post injury ^{4,5}. Most of the side effects of MPS therapy are related to the high dose MPS which may be detrimental to tissue protection and recovery, and the relatively modest neurological gains indicate inefficient dosing to the injured site ². One single injection of about 630 µg HMPS@AcDX in the limited volume (10 µL) can efficiently deliver the required amount of MPS (400 µg). With such high mass fraction of MPS, HMPS@AcDX reduced the MPS concentration in the first ~12–24 h and maintained a higher and more stable MPS concentration for the next 6 days compared to free MPS. The lower MPS concentration provided by HMPS@AcDX in the first ~12–24 h may avoid the detrimental effects of high dose MPS on tissue protection and recovery. Moreover, HMPS@AcDX significantly increased the area under curve and mean residence time of MPS in cerebral spinal fluid, suggesting an improved bioavailability of MPS at the site of injury compared to

free MPS. By comparing the therapeutic outcomes of HMPS@AcDX with all the other groups, we can conclude that the high loading and controlled release of MPS are indispensable for SCI therapy. HMPS@AcDX has demonstrated the potential to improve the bioavailability of MPS at the injured site while minimizing side effects and promote the clinical application of MPS.

The findings in our work highlighted the clinical potentials and translational possibilities of HMPS@AcDX. However, in order to clinically use HMPS@AcDX, more studies need to be performed, especially about the safety. AcDX has been shown to be biocompatible and biodegradable in many studies. However, it has not been approved by regulatory agencies for clinical use yet. The *in vivo* biocompatible and biodegradable of AcDX would need further investigations. In addition, 5 out of 20 animals died after the treatment with microspheres in high dose (4000 μg ; HAcDX and LMPS@AcDX groups), although only 1 out of 40 animals died after the treatment with microspheres in low dose (630 μg ; LAcDX and HMPS@AcDX groups). The exact reason for the death of animals treated with microspheres in high dose would need further investigations. Nevertheless, we recommend our drug delivery strategy of MPS, a controversial drug, by a formulation with both high MPS loading and controlled MPS release as such formulation has the potential to improve MPS's bioavailability at the site of injury while minimizing side effects of both drug and materials/excipients.

Furthermore, the therapeutic efficacy of polymer microspheres is mainly determined by the pharmacokinetics of delivered drugs. Human beings and small animals are different in the volume and replenishment rate of cerebrospinal fluid, which are expected to affect the pharmacokinetics of MPS released from the microspheres. Therefore, before clinical use, the drug release kinetics of HMPS@AcDX microspheres need to be further optimized to achieve desired pharmacokinetics of MPS and consequently satisfied clinical therapeutic efficacy in humans.

Taking into account Nature Communications' format requirement about the last paragraph of the Introduction section and to avoid repetition, we moved the discussion about MPS from the Introduction section to the Discussion section and combined it with the newly added discussion in the Discussion section of the revised version of the manuscript.

References:

1. Koszdin Kari L, Shen Danny D, Bernards Christopher M. Spinal cord bioavailability of methylprednisolone after intravenous and intrathecal administration: the role of

P-glycoprotein. *Anesthesiology* **92**, 156-156 (2000).

2. Canseco JA, *et al.* Updated review: the steroid controversy for management of spinal cord injury. *World Neurosurgery* **150**, 1-8 (2021).
3. Ahuja CS, *et al.* Traumatic spinal cord injury. *Nature Reviews Disease Primers* **3**, 17018 (2017).
4. Fehlings MG, *et al.* A clinical practice guideline for the management of patients with acute spinal cord injury: recommendations on the use of methylprednisolone sodium succinate. *Global Spine Journal* **7**, 203S-211S (2017).
5. Fehlings MG, *et al.* Efficacy and safety of methylprednisolone sodium succinate in acute spinal cord injury: a systematic review. *Global Spine Journal* **7**, 116S-137S (2017).